# Nucleophosmin 1 promotes mucosal immunity by supporting mitochondrial oxidative phosphorylation and ILC3 activity

Rongchuan Zhao[1,2,9], Jiao Yang [3,9], Yunjiao Zhai[4], Hong Zhang[1], Yuanshuai Zhou[1,2], Lei Hong[1,2], Detian Yuan[5], Ruilong Xia[6], Yanxiang Liu[3], Jinlin Pan[1,2], Shaheryar Shafi [1,2], Guohua Shi[1,2], Ruobing Zhang[1,2], Dingsan Luo[1], Jinyun Yuan[1], Dejing Pan[7], Changgeng Peng [6,8] ✉, Shiyang Li [4] ✉ & Minxuan Sun [1,2] ✉

Nucleophosmin 1 (NPM1) is commonly mutated in myelodysplastic syndrome (MDS) and acute myeloid leukemia. Concurrent inflammatory bowel diseases (IBD) and MDS are common, indicating a close relationship between IBD and MDS. Here we examined the function of NPM1 in IBD and colitis-associated colorectal cancer (CAC). NPM1 expression was reduced in patients with IBD. *Npm1*[+/−] mice were more susceptible to acute colitis and experimentally induced CAC than littermate controls. *Npm1* deficiency impaired the function of interleukin-22 (IL-22)-producing group three innate lymphoid cells (ILC3s). Mice lacking *Npm1* in ILC3s exhibited decreased IL-22 production and accelerated development of colitis. NPM1 was important for mitochondrial biogenesis and metabolism by oxidative phosphorylation in ILC3s. Further experiments revealed that NPM1 cooperates with p65 to promote mitochondrial transcription factor A (TFAM) transcription in ILC3s. Overexpression of *Npm1* in mice enhanced ILC3 function and reduced the severity of dextran sulfate sodium-induced colitis. Thus, our findings indicate that NPM1 in ILC3s protects against IBD by regulating mitochondrial metabolism through a p65-TFAM axis.

Inflammatory bowel diseases (IBD), including Crohn's disease (CD) and ulcerative colitis (UC), are characterized as chronic and recurring ailments of the gastrointestinal tract[1], which is considered a high risk of colitis-associated colorectal cancer (CAC)[2–4]. The precise pathogenesis of IBD remains unknown, but hypotheses include immune response disorders, alterations in intestinal microbiota, genetic susceptibility and environmental factors[2,5].

Myelodysplastic syndrome (MDS) is a hematopoietic stem cell disorder characterized by deficient hematopoiesis, cytopenia of peripheral blood and a predisposition to acute myeloid leukemia (AML)[6–8]. The cause of MDS is linked to the presence of acquired chromosomal abnormalities and genetic mutations that alter oncogene and tumor

suppressor gene function[9]. Since the first report of seven patients with both IBD and MDS in 1997, numerous cases of concurrent IBD and MDS have been documented[9–16]. Case studies of patients with IBD indicate a high incidence of AML/MDS in patients with IBD[16]. A high prevalence of IBD was also found in a large cohort of patients with MDS, suggesting a close association between IBD and MDS[15].

Mutations in nucleophosmin 1 (NPM1, also known as B23, numatrin 1 or NO38) are associated with a high risk of MDS and AML[6,17]. NPM1 was identified as a nucleolar phosphoprotein with multiple functions and binding partners[18]. NPM1 interacts with many partners in distinct cellular compartments, including nucleolar factors, transcription factors and histones. *NPM1* is the most frequently mutated gene in

patients with AML[19,20], accounting for ~60% of patients with a normal karyotype and 35% of total cases[21,22]. However, whether NPM1 regulates IBDs remains unknown.

Innate lymphoid cells (ILCs) were characterized as a family of heterogeneous lymphocytes that originate from common lymphoid progenitors in the bone marrow but with the absence of variable antigen receptors[23]. Group three ILCs (ILC3s) are the most abundant subgroup of ILCs in the gut and are the primary source of interleukin-22 (IL-22). ILC3s expressing the transcription factors retinoid-related orphan receptor gamma t (RORγt)[24] and aryl hydrocarbon receptor[25,26] produce IL-22, which triggers the synthesis of antimicrobial peptides, such as RegIIIβ and RegIIIγ, by epithelial cells[27,28]. Thus, ILC3s are at the beginning of a pathway that promotes immunity to infection. In a colon cancer model, *Il22*[−/−] mice were observed to undergo accelerated tumorigenesis compared to wild-type (WT) mice[29], suggesting a potential protective role for ILC3s in gut homeostasis.

In this study, we investigate the protective role of NPM1 in gut homeostasis and in the prevention of colitis. Using *Npm1*-haploinsufficient (*Npm1*[+/−]) mice, we observed increased susceptibility to colitis and colitis-associated colorectal cancer. NPM1 was abundant in ILC3s and was essential for IL-22 production in response to dextran sulfate sodium (DSS)-induced colitis. Conditional deletion of *Npm1* in the ILC3 lineage exacerbated colitis and decreased protective IL-22 secretion. Additionally, heterozygous deletion of *Npm1* in ILC3 dysregulated mitochondrial homeostasis, including decreased mitochondrial biogenesis and oxidative phosphorylation (OXPHOS). Mechanistically, we found that NPM1 acted as a transcription cofactor that bound p65 and stimulated mitochondrial transcription factor A (*Tfam*) transcription in DSS-induced colitis. Thus, our findings demonstrated that NPM1 regulates mitochondrial function and IL-22 production in ILC3s through the p65-TFAM axis, promoting gut homeostasis and protection against IBD.

## Results

### NPM1 deficiency leads to increased susceptibility to colitis
Patients with UC exhibited a decreased abundance of NPM1 in the colon compared to controls (Fig. 1a and Supplementary Table 1). We also observed a trend in reduced NPM1 in patients with CD; however, the reduction compared to controls was not significant (Fig. 1a). Further single-cell RNA-sequencing (scRNA-seq) analysis (GSE182270) on colonic biopsies of patients with UC and healthy control (HC)[30] indicated that the expression of *NPM1* decreased mainly in ILC3s, macrophages, natural killer T cells (NKT), cytotoxic T cells, regulatory T cells (T_reg) and Paneth cells in patients with UC (Extended Data Fig. 1a,b). *NPM1* mRNA abundance was also significantly reduced in patients with high-grade colon adenocarcinoma (COAD; stages III and IV), compared to those with low-grade COAD (stages I and II; Extended Data Fig. 1c). Analysis of The Cancer Genome Atlas database revealed that lower *NPM1* mRNA correlated with worse overall survival in patients with either COAD or rectum adenocarcinoma (READ; Extended Data Fig. 1d,e). These findings suggested that NPM1 may be involved in the pathology of IBD, especially UC, and may contribute to tumorigenesis.

To explore the putative contribution of NPM1 in gastrointestinal homeostasis and inflammation, we generated *Npm1*-haploinsufficient (*Npm1*[+/−]) mice (Supplementary Fig. 1a–c) and confirmed reduced abundance of NPM1 in the colon (Extended Data Fig. 1f–h). Note that homozygous knockout was lethal. In the absence of injury, colon length and histology were similar between WT mice and *Npm1*[+/−] mice (Extended Data Fig. 1i,j). Concurrently, the organogenesis of secondary lymphoid structures, including Peyer's patches (PP) and mesenteric lymph nodes (MLN), as well as solitary intestinal lymphoid tissue, was unaffected by *Npm1* haploinsufficiency (Extended Data Fig. 1k–n). Given the critical role of NPM1 in MDS and AML, we also examined the change of bone marrow (BM) cells in *Npm1*[+/−] mice (Supplementary Fig. 2a). Results indicated that the ratio of Lin−c-Kit+ (LK) cells, Lin−Sca1+c-Kit+ (LSK) cells and LS^low K^low cells in the BM of *Npm1*[+/−] mice

was elevated compared with that of *Npm1*[+/+] mice both in steady state and DSS-induced colitis conditions (Extended Data Fig. 1o,p), which is a characteristic phenotype of MDS[21,31,32]. Meanwhile, within the LK cell population, the proportion of granulocyte–macrophage progenitors (GMP) increased in *Npm1*[+/−] mice, especially in a steady state (Extended Data Fig. 1o,p). Using a DSS colonic injury model (2.5% wt/vol for 7 days), we found that *Npm1*[+/−] mice had greater body weight loss and a greater increase in disease activity index (DAI), a marker of inflammation, compared to littermate controls (Fig. 1b,c). On day 10 (3 days into the recovery period), NPM1 deficiency exacerbated inflammation as indicated by reduced colon length and increased epithelial injury, submucosal edema and leukocyte infiltration in the colon (Fig. 1d–f). Additionally, at day 5 of DSS exposure, expression of genes encoding antimicrobial peptides (*Reg3b* and *Reg3g*) was reduced and calprotectin (*S100a8* and *S100a9*), a marker of inflammation, was altered in colons of *Npm1*[+/−] mice (Fig. 1g,h). We also established a trinitrobenzene sulfonic acid (TNBS)-induced colitis model and evaluated the progress of colitis in WT and *Npm1*[+/−] mice. As anticipated, *Npm1*[+/−] mice also exhibited reduced colon length and enhanced inflammation, together with greater body weight loss and increased DAI (Extended Data Fig. 1q–u). Collectively, these data indicated that NPM1 has a protective role in the mouse colitis model.

### NPM1 inhibits colitis-associated colon tumorigenesis
Patients with IBD have a high risk of developing CAC[33,34]. To investigate the role of NPM1 in CAC development, we subjected WT mice and *Npm1*[+/−] mice to an azoxymethane (AOM)/DSS colon tumor model (Fig. 1i). By the end of the third cycle of DSS treatment, *Npm1*[+/−] mice failed to recover body weight and exhibited increased DAI (Extended Data Fig. 1v,w). Compared to WT mice, *Npm1*[+/−] mice developed more tumors and a greater number of larger tumors, indicative of a higher tumor burden (Fig. 1j–l). In addition to CAC, which is preceded by chronic inflammation, sporadic colorectal cancer (CRC) is a form of CRC that is often caused by mutations in the gene *APC*[35]. To examine the role of NPM1 in sporadic CRC, we crossed *Npm1*[+/−] mice with *Apc*[min/+] mice and fed them a Western diet to accelerate tumorigenesis. Results showed that there were no significant differences in colonic tumor load between *Apc*[min/+] mice and *Npm1*[+/−]*Apc*[min/+] mice (Extended Data Fig. 1x), suggesting that sporadic CRC arising from *APC* mutations does not involve NPM1. Collectively, these findings indicated that NPM1 has a pivotal role in impeding colitis-associated colon tumorigenesis by restricting tumor development and growth.

### Protection against colitis requires NPM1 in hematopoietic cells
Given that *Npm1* is expressed by many types of cells and decreased under pathological conditions (Extended Data Figs. 1b and 2a), it is unclear whether the exacerbated colitis in *Npm1*[+/−] mice is due to defects in hematopoietic or nonhematopoietic cells, particularly colonic epithelial cells. Thus, we established BM chimeras with *Npm1* deficiency in these distinct cellular populations (Extended Data Fig. 2b). After a 7-day DSS treatment, mice receiving *Npm1*-haploinsufficient BM exhibited more severe colitis compared to mice receiving *Npm1* WT BM cells. However, when the same donor BM was used regardless of the genotype of the host mice, there were no significant differences in body weight, colon length or histological features (Extended Data Fig. 2c–f), suggesting that the hematopoietic compartment is the main functional compartment for NPM1. We also detected the expression of tight junction genes (including *Tjp1*, *Tjp2*, *Cldn2* and *Cldn3*) in epithelial cells, which are pivotal for the maintenance of intestinal barrier function[36]. With the exception of *Cldn3*, which is diminished in *Npm1*-haploinsufficient mice under physiological conditions, the expression of other tight junction genes remains relatively unchanged between two groups of mice in both physiological and pathological conditions (Extended Data Fig. 2g–j). We also generated *Npm1*[flox/flox] mice (Supplementary Fig. 1d–f) and crossed them with *Villin*[cre/+] mice to directly assess a role in protection

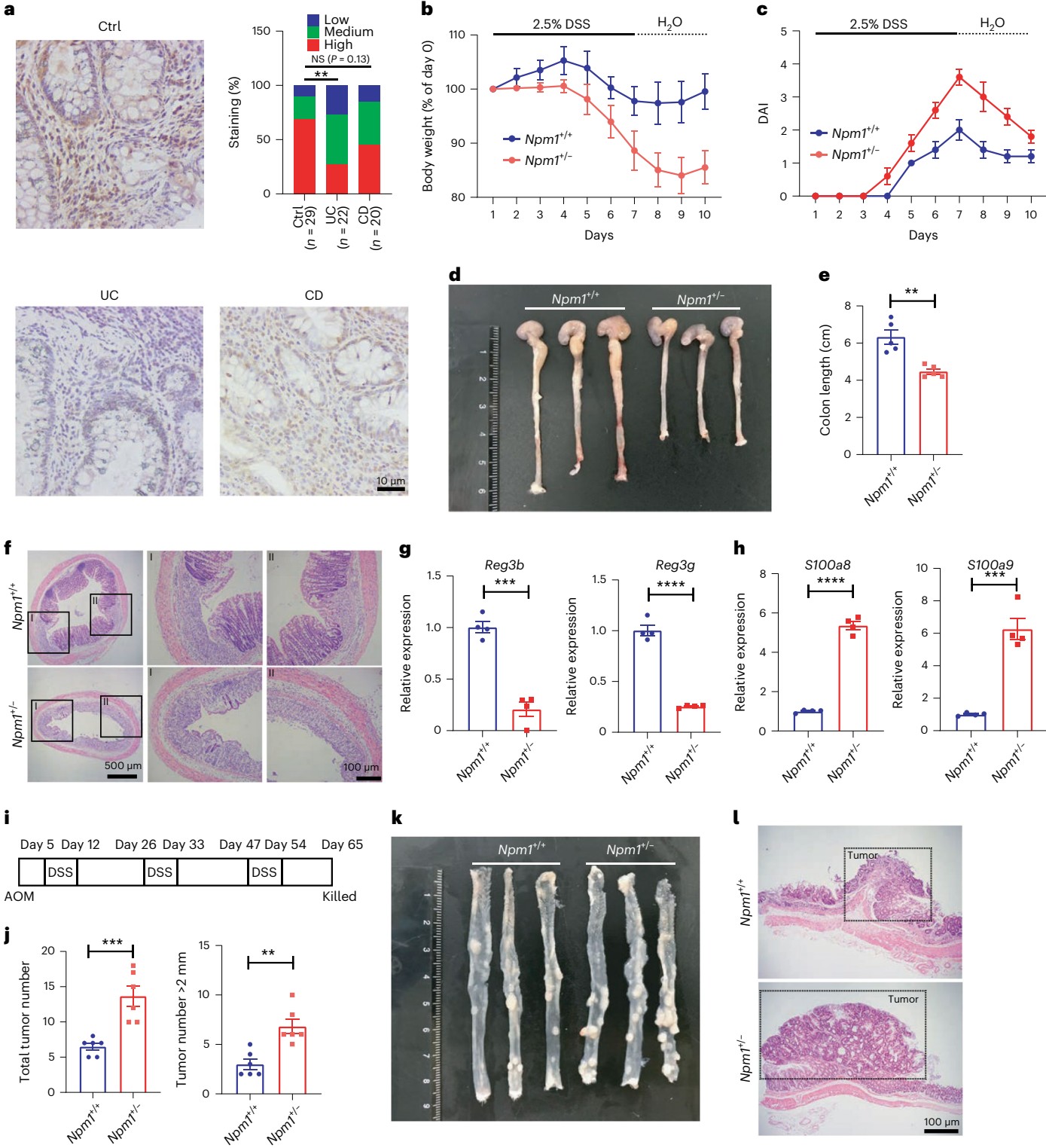

**Fig. 1 | NPM1 deficiency increases susceptibility to colitis and colonic adenocarcinoma. a**, Immunohistochemistry of NPM1 in colon tissue from patients with IBD (UC, $n = 22$ individual patients and CD, $n = 20$ individual patients) and non-IBD ($n = 29$ individual patients) controls. Scale bars = 10 μm. Immunohistochemistry score of NPM1. Statistical differences were determined by the Mann–Whitney test (**$P < 0.01$). **b**–**f**, $Npm1^{+/-}$ and control $Npm1^{+/+}$ mice were administered 2.5% DSS for 7 days, followed by 3 days of recovery ($H_2O$). Body weight (**b**), DAI (a score of inflammation) in the colon (**c**), colon length on day 10 (**d,e**) and colon histopathology on day 10 (**f**) were analyzed ($n = 5$ individual mice). Scale bars = 500 μm (left) and 100 μm (right). (**g,h**) RT–PCR analysis of mRNA abundance of *Reg3b* and *Reg3g* (**g**) and *S100a8* and *S100a9* (**h**)

in the whole colon of mice at day 5 of administration of 2.5% DSS ($n = 4$ individual mice). **i**, Diagram of AOM/DSS CAC model. **j**, Total number of tumors and number of tumors larger than 2 mm in $Npm1^{+/+}$ and $Npm1^{+/-}$ mice ($n = 5$ individual mice). **k**, Representative images of colons with tumors from $Npm1^{+/+}$ and $Npm1^{+/-}$ mice on day 65 of the AOM/DSS CAC model. **l**, Histopathology of representative colon tumors from $Npm1^{+/+}$ and $Npm1^{+/-}$ mice on day 65 of the AOM/DSS CAC model. Scale bars = 100 μm. Data in **e**, **g**, **h** and **j** are representative of two independent experiments, shown as the means ± s.e.m., and statistical significance was determined by two-tailed unpaired Student's *t* test (**$P < 0.01$, ***$P < 0.001$ and ****$P < 0.0001$), unless otherwise indicated. Ctrl, control; NS, not significant; CAC, colitis-associated colorectal cancer.

against colitis for NPM1 in colonic epithelial cells. However, there was no obvious alteration in colon length and histological features between *Villin*^cre/+*Npm1*^flox/flox mice and control mice (Extended Data Fig. 2k–m). Taken together, these data showed that impaired gut homeostasis and exacerbated inflammation in *Npm1*^+/− mice are mainly caused by the heterozygous deletion of *Npm1* in the hematopoietic compartment.

## NPM1 is critical for maintaining IL-22-producing ILC3s

Subsequently, we investigated the type of gut immune cells involved in limiting gut inflammation by NPM1. The ratio of macrophages, neutrophils, eosinophils and dendritic cells (DCs) infiltrated in intestinal lamina propria leukocytes (LPLs) exhibited few changes between WT and *Npm1*^+/− mice in steady state (Supplementary Fig. 2b and Extended Data Fig. 3a–d). However, in DSS-induced colitis, an elevation of these cells was observed in *Npm1*^+/− mice compared to WT mice (Extended Data Fig. 3e–h). It's known that infiltration of myeloid cells into the intestinal lamina propria is considered a common cause of progressive colitis[37]. Furthermore, clearance of CD11b^+ myeloid cells failed to rescue the exacerbated enteritis in *Npm1*^+/− mice, suggesting that NPM1 in myeloid cells was insufficient to regulate intestinal inflammation (Extended Data Fig. 3i–m). Likewise, evaluation of T cells (T_H17, T_reg and γδT cells) coupled with comparable colitis in two genotype mice after deletion of CD3^+ T cells indicated that exacerbated colitis in *Npm1*-haploinsufficient mice was not attributed to T cells (Supplementary Fig. 2c and Extended Data Fig. 3n–x).

We then investigated the effect of *Npm1* haploinsufficiency on colonic ILC3s (Supplementary Fig. 2d). The population of colonic ILC3s and IL-22^+ ILC3s decreased in *Npm1*^+/− mice compared to *Npm1*^+/+ mice after DSS administration, suggesting that haploinsufficient of *Npm1* affects ILC3 expansion and function (Fig. 2a–e). Additionally, *Npm1*^+/− ILC3 exhibited similar alterations in TNBS-induced colitis (Extended Data Fig. 4a–d). However, these changes were not observed under physiological conditions (Extended Data Fig. 4e–h). Further analysis revealed that there were no evident alterations in proportions of NCR^+ ILC3 and CCR6^+ ILC3 between WT and *Npm1*^+/− mice under physiological or pathological conditions (Extended Data Fig. 4i,j). Moreover, isolated ILC3s from *Npm1*^+/− mice produced less IL-22 compared with ILC3s from WT mice after DSS administration (Fig. 2f). In addition, the expression of *Il22* was also decreased in isolated ILC3s from *Npm1*^+/− mice exposed to DSS, but the expression of *Il22* was similar in both genotypes under steady state (Fig. 2g). The decreased production of IL-22 in ILC3s may contribute to the observed dysregulation of *Reg3b* and *Reg3g* in *Npm1*^+/− mice in DSS-induced colitis (Fig. 1g), and thus impaired intestinal microbiota homeostasis. There was a rapid decrease in observed operational taxonomic unit, Chao1 index and Shannon index in *Npm1*^+/− mice (Extended Data Fig. 4k–m), indicating that microbiota diversity

was repressed by *Npm1* heterozygote deletion. Moreover, feces from *Npm1*^+/− mice and WT mice showed a remarkable change in bacterial composition (Extended Data Fig. 4n–q). However, cohousing littermate *Npm1*^+/− mice still exhibited more pronounced exacerbation of enteritis compared to WT mice (Extended Data Fig. 4r–u), indicating that changes in the gut microbiota are not the priori drivers of the exacerbated inflammation in *Npm1*^+/− mice but may instead contribute to a certain extent to the exacerbation of enteritis. Collectively, our results indicated that NPM1 is important for the protective function of ILC3s in the gut immune microenvironment.

To specifically decipher the cell-intrinsic role of *Npm1* in colonic ILC3s, we generated *Rorc*^cre/+ *Npm1*^flox/flox mice that lack *Npm1* on ILC3s and subjected the mice to DSS-induced colitis. The development of PP and MLN was unimpaired in *Rorc*^cre/+*Npm1*^flox/flox mice (Extended Data Fig. 5a–c). Frequencies of intestinal ILC3 and IL-22^+ ILC3 in *Rorc*^cre/+*Npm1*^flox/flox mice were also comparable with those of the control group (Extended Data Fig. 5d–g). However, compared to *Npm1*^flox/flox mice, *Rorc*^cre/+*Npm1*^flox/flox mice exhibited greater loss of body weight and increased DAI (Fig. 2h–i), indicating exacerbated inflammation following DSS administration. When killed on day 10 (3 days after recovery), *Rorc*^cre/+*Npm1*^flox/flox mice exhibited decreased colon length and greater features of colon injury (Fig. 2j–l). The frequencies of colonic ILC3s and IL-22^+ ILC3s were also decreased in *Rorc*^cre/+*Npm1*^flox/flox mice after DSS administration (Fig. 2m–p). Without development defects, heterozygous deletion of *Npm1* in ILC3 also contributed to exacerbated enteritis and reduction of ILC3, which appears to be in a dose-dependent manner (Extended Data Fig. 5h–q). Furthermore, the percentage of apoptotic ILC3s was increased in *Rorc*^cre/+*Npm1*^flox/flox mice in DSS-induced colitis (Supplementary Fig. 2e and Extended Data Fig. 5r,s). The proportion of CCR6^+ ILC3 in total ILC3s was higher in *Rorc*^cre/+*Npm1*^flox/flox mice than that in *Npm1*^flox/flox mice under pathological conditions, which was opposite in steady state (Extended Data Fig. 5t,u). The proportion of interferon-γ (IFNγ)-producing ex-ILC3 was also unchanged between these two groups of mice with or without DSS administration (Supplementary Fig. 2f and Extended Data Fig. 5v–y). Additionally, consistent with changes observed in *Npm1*^+/− mice, the increased infiltration of myeloid cells also existed in *Rorc*^cre/+*Npm1*^flox/flox mice under pathological conditions (Extended Data Fig. 6a–h). Because RORc-Cre will also delete *Npm1* in conventional T cells and γδT cells, we examined the function of various T cell subsets and excluded their contributions to exacerbated colitis in *Rorc*^cre/+*Npm1*^flox/flox mice by depleting T cells using a CD3 antibody (Extended Data Fig. 6i–q). Moreover, isolated ILC3s from *Rorc*^cre/+*Npm1*^flox/flox mice showed less *Il22* expression and IL-22 production compared with *Npm1*^flox/flox ILC3s (Fig. 2q,r). We also confirmed our findings in vitro using the ILC3 cell line, MNK3. MNK3 cells retain phenotypic and functional features

**Fig. 2 | NPM1 is required for maintaining the frequency and function of colonic ILC3s. a**, Colon LPLs were isolated from *Npm1*^+/+ and *Npm1*^+/− mice at day 5 of administration of 2.5% DSS. Analysis of ILC3s (live CD45^+Lin^−RORγt^+ cells) and IL-22-producing ILC3s (live CD45^+Lin^−RORγt^+IL-22^+ cells) by flow cytometry. Numbers indicate percentages of cells in each outlined region. **b,c**, The proportion of CD45^+ cells that are ILC3s (**b**; *n* = 6 individual mice) and the proportion of IL-22^+ ILC3s in the total ILC3 population (**c**; *n* = 5 individual mice) in LPLs of *Npm1*^+/+ and *Npm1*^+/− mice after DSS administration are shown. **d,e**, Number of ILC3s (**d**) and IL-22^+ ILC3s (**e**) in LPLs of *Npm1*^+/+ and *Npm1*^+/− mice after DSS administration are depicted (*n* = 5 individual mice). **f**, ILC3s, isolated by cell sorting from LPLs of *Npm1*^+/+ and *Npm1*^+/− mice after DSS administration, were analyzed by ELISA for IL-22 (*n* = 3 individual mice). **g**, Relative mRNA abundance of *Il22* in ILC3s, isolated by cell sorting from the LPL of *Npm1*^+/+ and *Npm1*^+/− mice exposed to 2.5% DSS or water (steady state), was analyzed. The results are shown relative to the amount in cells from *Npm1*^+/+ mice exposed to water (steady state; *n* = 6 individual mice). **h–l**, *Npm1*^flox/flox and *Rorc*^cre/+*Npm1*^flox/flox mice were administered 2.5% DSS for 7 d followed by 3 d of recovery. Body weight (**h**), DAI (**i**), colon length (**j,k**) and colon histopathology on day 10 (**l**)

were analyzed (*n* = 5 individual mice). Scale bars = 500 μm (left) and 100 μm (right). **m,n**, LPLs were isolated from *Npm1*^flox/flox and *Rorc*^cre/+*Npm1*^flox/flox mice after 5 days of administration of 2.5% DSS (*n* = 5 individual mice). The proportion of CD45^+ cells that are ILC3s (**m**) and the proportion of IL-22^+ ILC3s in the total ILC3 population (**n**) are shown. **o,p**, The number of ILC3s (**o**) and IL-22^+ ILC3s (**p**) in LPLs of *Npm1*^flox/flox and *Rorc*^cre/+*Npm1*^flox/flox mice after DSS administration are depicted (*n* = 5 individual mice). **q**, Relative mRNA abundance of *Il22* in ILC3s, isolated by cell sorting from the LPL of *Npm1*^flox/flox and *Rorc*^cre/+*Npm1*^flox/flox mice at day 5 of administration of 2.5% DSS was analyzed (*n* = 4 individual mice). **r**, ILC3s, isolated by cell sorting from LPLs of *Npm1*^flox/flox and *Rorc*^cre/+*Npm1*^flox/flox, were analyzed by ELISA for IL-22 (*n* = 3 individual mice). **s,t**, IL-22 production by MNK3 cells after stimulation with IL-1β and IL-23 in vitro by flow cytometry (**s**; *n* = 5 individual mice) and by ELISA (**t**; *n* = 3 individual mice). Data in **b–g**, **k** and **m–t** are representative of two independent experiments, shown as the means ± s.e.m., and statistical significance was determined by two-tailed unpaired Student's *t* test (*$P < 0.05$, **$P < 0.01$, ***$P < 0.001$ and ****$P < 0.0001$). LPLs, lamina propria leukocytes.

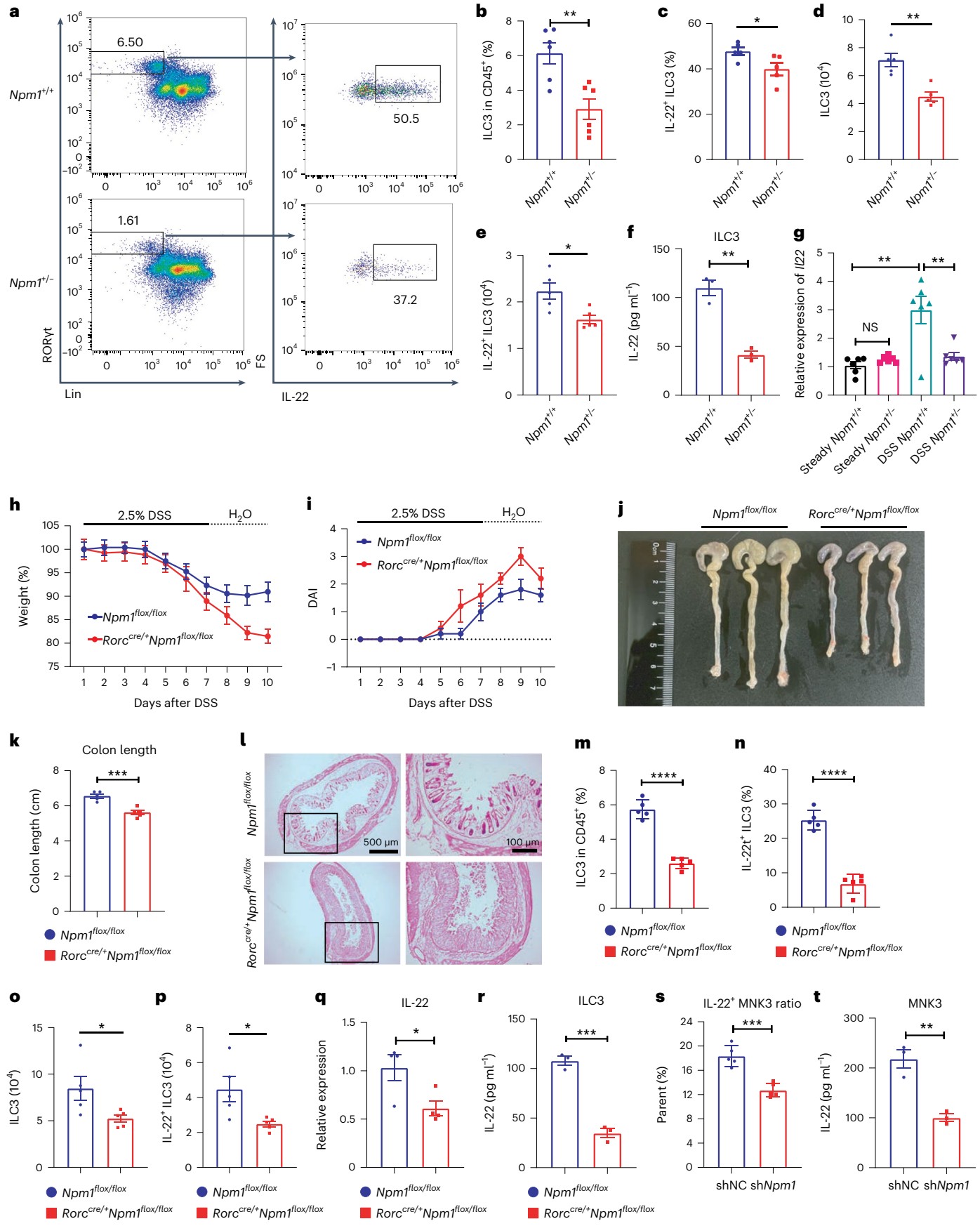

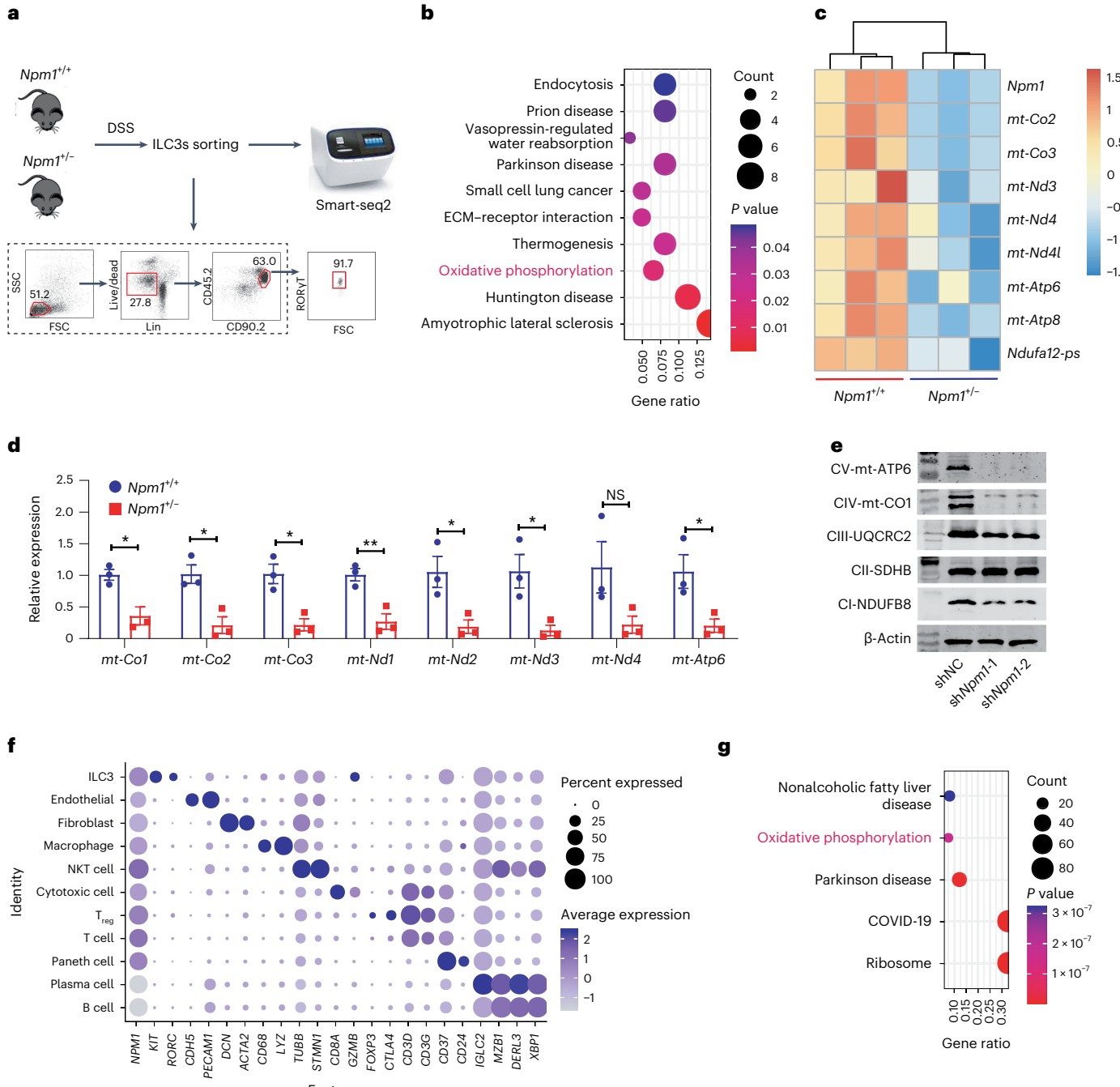

**Fig. 3 | NPM1 regulates the OXPHOS pathway in ILC3s. a**, RNA-seq analysis of colonic ILC3 isolated from $Npm1^{+/+}$ and $Npm1^{+/-}$ mice at day 5 of administration of 2.5% DSS ($n$ = 3). **b**, KEGG pathway enrichment analysis of downregulated genes in $Npm1^{+/-}$ mice ($n$ = 3 individual mice). **c**, Heatmap of selected DEGs encoding proteins involved in OXPHOS in ILC3s between $Npm1^{+/+}$ and $Npm1^{+/-}$ mice ($n$ = 3 individual mice). **d**, RT–PCR analysis of mRNA abundance of the indicated genes in ILC3s, isolated by cell sorting from the LPL of $Npm1^{+/+}$ and $Npm1^{+/-}$ mice at day 5 of administration of 2.5% DSS ($n$ = 3 individual mice). **e**, Western blot showing the abundance of selected mitochondrial complex components in MNK3 cells. The samples were derived from the same experiment, and the blots were

processed in parallel. **f**, Single-cell analysis of colonic samples of patients with UC from the GEO database (GSE182270). Representative DEGs ($x$ axis) by cluster ($y$ axis) with dot size representing the fraction of cells within the cluster that express each gene and colors indicating the $z$-scaled expression of genes in cells within each cluster. **g**, KEGG pathway enrichment analysis of upregulated genes in $NPM1^{high}$ ILC3s compared to $NPM1^{low}$ ILC3s from the data of patients with UC. Data in **d** is representative of two independent experiments, shown as the mean ± s.e.m., and statistical significance was determined by two-way ANOVA (*$P$ < 0.05 and **$P$ < 0.01).

characteristic of mouse primary ILC3s, including the production of IL-17A and IL-22 when stimulated with IL-23 and IL-1β[38,39]. Knockdown of $Npm1$ in MNK3 significantly suppressed the secretion of IL-22 upon stimulation (Fig. 2s,t). Furthermore, $Rorc^{cre/+}Npm1^{flox/flox}$ mice developed more tumors compared with $Npm1^{flox/flox}$ mice when subjected to AOM/DSS (Extended Data Fig. 6r–t). Collectively, these data supported

that NPM1 in ILC3s is critical for gut homeostasis under injury conditions and limiting inflammation.

### NPM1 promotes mitochondrial gene expression in ILC3s
To uncover mechanisms by which NPM1 regulates ILC3 expansion and function, we performed RNA-seq (smart-seq2) of Live⁺Lin⁻CD45^{low}CD

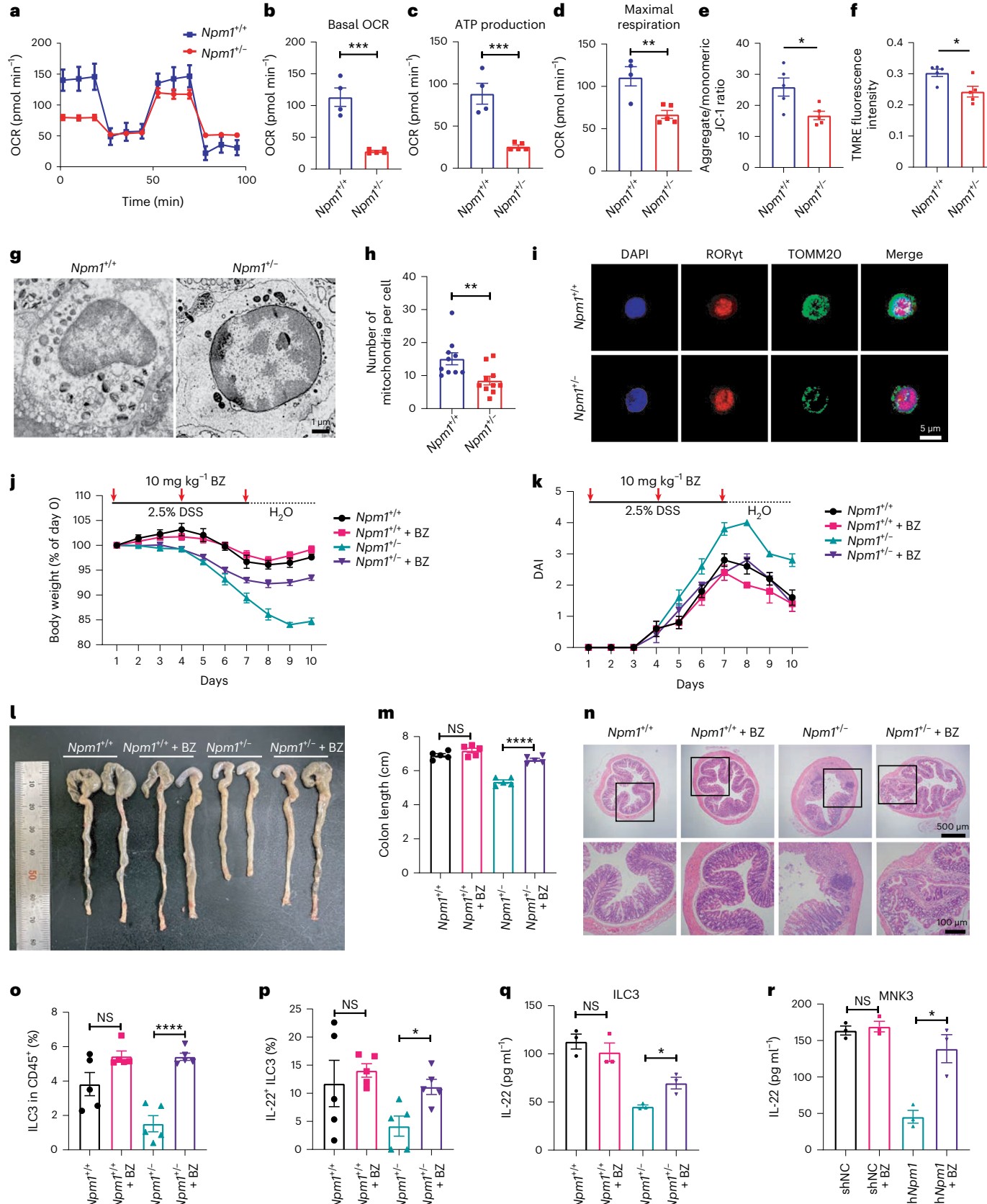

90$^{high}$ LPLs[27,40] from colon of WT and Npm1$^{+/-}$ mice with colitis induced by DSS treatment (Fig. 3a). Kyoto Encyclopedia of Genes and Genomes (KEGG) pathway analysis revealed that the OXPHOS pathway is a top differentially modulated pathway in Npm1-haploinsufficient mice

compared to WT mice (Fig. 3b). We also observed decreased expression of several genes encoding mitochondrial complex subunits of OXPHOS in Npm1$^{+/-}$ mice, specifically those for electron transport chain (ETC) complex I (mt-Nd3, mt-Nd4, mt-Nd4l and Ndufa12-ps), complex IV

**Fig. 4 | *Npm1* is essential for maintaining mitochondrial OXPHOS and biogenesis in ILC3s. a–d,** Cell mito stress test was performed with isolated colonic ILC3s from *Npm1*[+/+] (*n* = 4 individual mice) and *Npm1*[+/−] (*n* = 5 individual mice) mice. Representative OCR profile (**a**), basal OCR (**b**), ATP production (**c**) and maximal respiration (**d**) are shown. **e,f,** Mitochondrial membrane potential was assessed with the indicator 5,5′,6,6′-tetrachloro-1,1′,3,3′-tetraethylbenzimid azolylcarbocyanine iodide (JC-1) (**e**) and TMRE (**f**) in isolated colonic ILC3s from *Npm1*[+/+] and *Npm1*[+/−] mice under DSS (*n* = 5 individual mice). **g,** Ultrastructural analysis of mitochondria by SEM of isolated colonic ILC3s from *Npm1*[+/+] and *Npm1*[+/−] mice. Scale bar = 1 μm. **h,** The number of mitochondria per cell was counted in SEM images (*n* = 10 fields per group). **i,** TOMM20 in ILC3s from *Npm1*[+/+] and *Npm1*[+/−] mice was detected by immunofluorescence staining. Scale bar = 5 μm. **j–n,** *Npm1*[+/+] and littermate control *Npm1*[+/−] mice were treated

with bezafibrate (i.g., oral gavage) and administered 2.5% DSS for 7 days followed by 3 days of recovery. Body weight (**j**), DAI (**k**), representative colon images (**l**), colon length (**m**) and colon histopathology (**n**) are shown (*n* = 5 individual mice). Scale bars = 500 μm (up) and 100 μm (down). **o,p,** The proportion of CD45[+] cells that are ILC3s (**o**) and the proportion of IL-22[+] ILC3s in the total ILC3 population (**p**) in LPLs from mice of the indicated genotypes with or without bezafibrate treatment (*n* = 5 individual mice). **q,r,** Analysis of IL-22 production by isolated colonic ILC3s from mice treated with or without bezafibrate (10 mg kg[−1], i.g.; **q**) and MNK3 (**r**) cells with or without bezafibrate (200 μM) addition by ELISA (*n* = 3 individual mice). Data in **b–f, h, m, o–r** are representative of two independent experiments, shown as the means ± s.e.m., and statistical significance was determined by two-tailed unpaired Student's *t* test (\**P* < 0.05, \*\**P* < 0.01, \*\*\**P* < 0.001 and \*\*\*\**P* < 0.0001). BZ, bezafibrate.

(*mt-Co2* and *mt-Co3*) and complex V (*mt-Atp6* and *mt-Atp8*; Fig. 3c) and confirmed these findings by RT–PCR (Fig. 3d). However, the universal decrease of mtDNAs was not observed in epithelial cells, macrophages or T cells of *Npm1*[+/−] mice (Extended Data Fig. 7a–c). The abundance of NDUFB8 (complex I), mt-CO1 (complex IV) and mt-ATP6 (complex V) was also notably reduced due to *Npm1*-knockdown in MNK3 cells (Fig. 3e). These results indicated that NPM1 has a role in regulating the OXPHOS pathway in ILC3s.

To validate that NPM1 regulates OXPHOS in ILC3s in humans as well, scRNA-seq data of human colonic biopsies (GSE182270)[30] was analyzed, and the ILC3 cluster was identified based on higher expression of *KIT*, *RORC* but lower expression of *CTLA4*, *CD3D* and *CD3G* (markers of T cells; Fig. 3f). Although *NPM1* is broadly expressed across all clusters, ILC3s were among those with comparatively high expression (Fig. 3f). Similar to our mouse data, KEGG pathway enrichment analysis of differentially expressed genes (DEGs) between *NPM1*[high] and *NPM1*[low] ILC3s in patients with UC identified the OXPHOS pathway among the top five pathways regulated by NPM1 (Fig. 3g). These findings suggested that altered cellular metabolism through the OXPHOS pathway in ILC3s represents a potential mechanism by which NPM1 activity influences UC.

## Lack of NPM1 impairs mito-OXPHOS and biogenesis in ILC3s

According to the abovementioned results, mitochondrial OXPHOS is probably impaired in *Npm1*-haploinsufficient ILC3s (Fig. 3c–e). Therefore, we evaluated OXPHOS in isolated ILC3s from DSS-induced *Npm1*[+/−] mice and WT mice. Heterozygous deletion of *Npm1* in ILC3s reduced oxygen consumption rate (OCR) in response to DSS (Fig. 4a). Compared to WT ILC3s, *Npm1*[+/−] ILC3s exhibited a marked reduction in basal OCR, ATP production and maximal respiration (Fig. 4b–d), indicating that mitochondrial OXPHOS in ILC3s was impaired by insufficient NPM1. However, such impaired mitochondrial function in *Npm1*[+/−] ILC3s was not observed under physiological conditions (Extended Data Fig. 7d–g). In the DSS model, mouse intestinal ILC3s exhibited a dramatic mitochondrial activation in the acute tissue damage phase (day 5) and then partially restored to a normal state in the repair phase (day 10; Extended

Data Fig. 7h–k). The inadequate mitochondrial activation of ILC3 in the acute phase caused by heterozygous deletion of *Npm1* could lead to exacerbated colitis (Fig. 4b–d). Besides, epithelial cells, macrophages and T cells in *Npm1*[+/−] mice exhibited few differences in OXPHOS compared to those in WT mice in both steady state and DSS-induced colitis conditions (Extended Data Fig. 7l–t). Moreover, the mitochondrial membrane potential of *Npm1*[+/−] ILC3s was also reduced significantly compared with that of WT ILC3s only under pathological conditions (Fig. 4e,f and Extended Data Fig. 7u,v). These results showed the importance of NPM1 in maintaining mitochondrial OXPHOS in ILC3s.

To determine whether reduced transcription of OXPHOS genes in *Npm1*-haploinsufficient ILC3s (Fig. 3c–e) is associated with decreased mitochondrial biogenesis, we quantified the number of mitochondria using scanning electron microscopy (SEM; Fig. 4g). We found fewer mitochondria per cell in *Npm1*[+/−] ILC3s (Fig. 4h). ILC3s from *Npm1*[+/−] mice also had reduced staining of TOMM20, a mitochondrial protein (Fig. 4i). Collectively, these data indicated that NPM1 has a critical function in maintaining mitochondria numbers and mitochondrial metabolism in ILC3s and that impairment of such metabolism represents a mechanism by which heterozygous deletion of *Npm1* exacerbates DSS-induced colitis.

To confirm that mitochondrial biogenesis and function were impaired by *Npm1* heterozygous deletion and that such impairment contributed to colitis severity, we used bezafibrate, an agonist of the transcription factors peroxisome proliferator-activated receptor (PPAR) γ coactivator 1α (PGC1α) that stimulates mitochondrial OXPHOS and biogenesis[41,42]. Compared with *Npm1*[+/−] mice without bezafibrate treatment during the course of DSS administration, *Npm1*[+/−] mice receiving bezabrifate exhibited greater recovery of body weight, greater reduction in DAI, longer colons and reduced inflammation (Fig. 4j–n), suggesting that maintaining mitochondrial function through bezafibrate limited colitis severity in *Npm1*[+/−] mice. However, bezafibrate had minimal impact on mice with sufficient NPM1 function, suggesting that both NPM1 and bezafibrate maintain mitochondrial function to limit colitis (Fig. 4j–n). Similarly, bezafibrate

**Fig. 5 | The role of NPM1 in p65 signaling. a,** Schematic diagram of protein–protein interaction analysis with NPM1. **b,** Silver-stained gel showing proteins that were immunoprecipitated with NPM1 and exhibited higher intensity in stimulated than unstimulated cells. Red rectangle shows bands that were excised for MS analysis. **c,** Top five candidate NPM1-interacting proteins identified by MS. **d,** IP and IB of the interaction between NPM1 and p65 in MNK3 cells. The samples were derived from the same experiment, and the blots were processed in parallel. **e,** Immunofluorescence staining of the subcellular location of p65 in ILC3s isolated from mice at day 5 of administration of 2.5% DSS or under the steady state (water). Scale bar = 5 μm. **f,** Unstimulated or stimulated MNK3 cells were subjected to cellular fractionation into cyto and nuc fractions followed by western blotting for p65. Glyceraldehyde-3-phosphate dehydrogenase (GAPDH) and histone H3 were used as markers for cytosolic and nuclear proteins, respectively. **g,** RT–PCR analysis of mRNA abundance of p65 target genes in isolated colonic ILC3s from *Npm1*[+/+] and *Npm1*[+/−] mice at day 5 of administration

of 2.5% DSS (*n* = 5 individual mice). **h,** KEGG pathway enrichment analysis of upregulated transcription factor-related pathways in ILC3s from patients with UC compared with ILC3s from healthy participants. **i,** Gene Ontology (GO) analysis of downregulated pathways in *NPM1*[low] ILC3s compared with *NPM1*[high] ILC3s from patients with UC, which was identified using a median expression cutoff for *NPM1* in ILC3 of patients with UC. **j–m,** Cell mito stress test was performed with stimulated MNK3 cell line with or without *p65* knockdown. Representative OCR (**j**), basal OCR (**k**), ATP production (**l**) and maximal respiration (**m**) are shown (*n* = 3 biological samples). **n,** Expression of *Il22* in unstimulated and stimulated MNK3 cell line (shNC and sh*p65*; *n* = 3 biological samples). Data in **g** and **k–n** are representative of two independent experiments, shown as the means ± s.e.m., and statistical significance was determined by two-tailed unpaired Student's *t* test (\**P* < 0.05, \*\**P* < 0.01, \*\*\**P* < 0.001 and \*\*\*\**P* < 0.0001). IB, immunoblot; cyto, cytosol; nuc, nuclear.

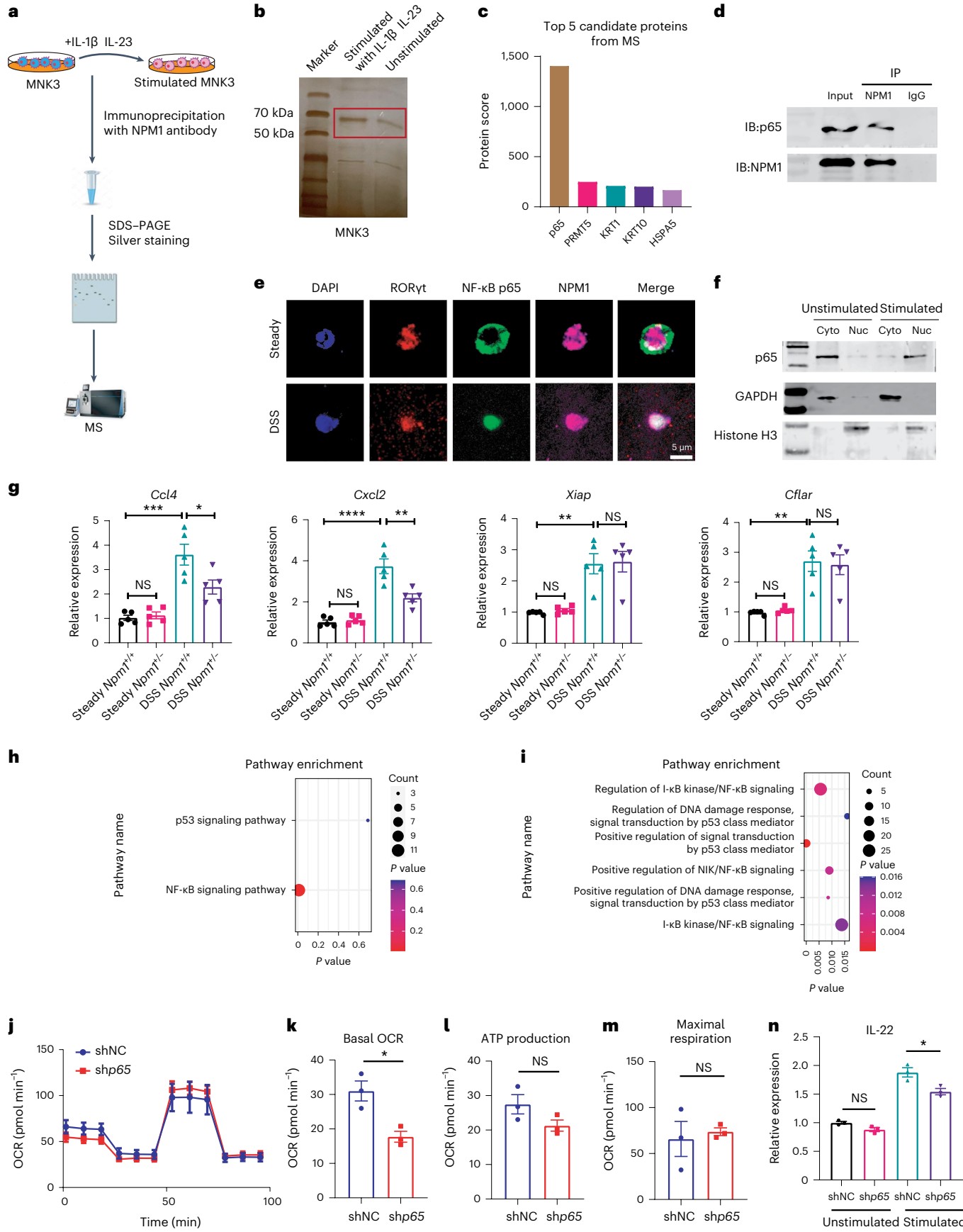

succeeded in reversing the colitis in *Rorc*^cre/+*Npm1*^flox/flox mice (Extended Data Fig. 7w–y). Moreover, in *Npm1*^+/− mice exposed to DSS, bezafibrate resulted in increased percentages of total colonic ILC3s, IL-22^+ ILC3s and IL-22 production by ILC3s (Fig. 4o–q), suggesting that sufficient mitochondrial OXPHOS and biogenesis are required for ILC3 activity in DSS-induced colitis.

MNK3 also exhibited mitochondrial activation after IL-1β/IL-23 stimulation, which was regulated by NPM1 (Extended Data Fig. 8a–d). Knockdown of *Npm1* in MNK3 suppressed the secretion of IL-22 in response to IL-23 and IL-1β (Fig. 4r). However, bezafibrate rescued ILC3 function in terms of IL-22 secretion in *Npm1*-knockdown cells (Fig. 4r and Extended Data Fig. 8e,f). In contrast, OXPHOS inhibitors oligomycin and rotenone suppressed the activation of MNK3 (Extended Data Fig. 8g,h). However, the difference in *Il22* expression between shNC and sh*Npm1* MNK3 after OXPHOS inhibitor administration indicated that NPM1 may participate in other biological processes to sustain ILC3 activation (Extended Data Fig. 8g,h). The tricarboxylic acid (TCA) cycle, a crucial component of mitochondrial metabolism, is known to participate in the activation of immune cells[43]. Because succinate is a substrate for the TCA cycle, its addition partially rescued the impaired ILC3 activation resulting from *Npm1* heterozygous deletion (Extended Data Fig. 8i). These results revealed that the defect in mitochondrial function resulting from *Npm1* deficiency accounts for the impairment of ILC3 activation and function, leading to exacerbated colitis.

## NPM1 regulates *TFAM* transcription by binding to p65

To uncover the molecular mechanism by which NPM1 regulates mitochondrial homeostasis of ILC3s, we immunoprecipitated MNK3 cells with or without stimulation (Fig. 5a). Proteins associated with NPM1 were separated by SDS–PAGE and then silver stained. A band of ~70 kDa was enriched in stimulated MNK3 cells compared to unstimulated cells (Fig. 5b). Mass spectrometry (MS) revealed that p65, a component of the nuclear factor kappa B (NF-κB) transcription factor, is the top candidate for ~70 kDa protein that co-immunoprecipitated with NPM1 (Fig. 5c), which is consistent with a previous study reporting an interaction between NPM1 and p65 (also known as RelA), RelB and p50 (ref. 44). The two proteins, NPM1 and p65, were co-immunoprecipitated from stimulated MNK3 cells, confirming the MS findings and suggesting that the proteins interacted (Fig. 5d). Immunofluorescence analysis of ILC3s revealed that p65 was localized in the cytoplasm and NPM1 was localized in the nucleus in the noninflammatory steady state, whereas p65 accumulated in the nucleus after DSS-induced colitis and colocalized with NPM1 (Fig. 5e). Stimulation of MNK3 cells with IL-1β and IL-23 also promoted the accumulation of p65 in the nucleus (Fig. 5f). These results indicated that inflammatory stimulation induces subcellular translocation of p65 and promotes the interaction between p65 and NPM1 in the nucleus of ILC3s.

To investigate whether NPM1 functions as a transcriptional cofactor that binds to p65 and influences the transcription of p65 target genes in activated ILC3s, we thus monitored the expression of four p65-regulated genes (*Cxcl2*, *Ccl4*, *Xiap* and *Cflar*) and found that they were dramatically induced in the colitis condition compared with

the steady-state condition. However, only the expression of *Cxcl2* and *Ccl4* was markedly decreased in *Npm1*^+/− ILC3s compared to WT ILC3s from mice with DSS-induced colitis (Fig. 5g). These transcriptional results indicated that the NF-κB pathway in ILC3s was activated by DSS-induced colitis and that NPM1 contributes to the regulation of a subset of NF-κB target genes. We also observed that the NF-κB signaling pathway in ILC3s from patients with UC was significantly upregulated compared to HCs in GSE182270 (Fig. 5h). Additionally, several NF-κB-related signaling pathways were also enriched when comparing *NPM1*^high ILC3s and *NPM1*^low ILC3s from patients with UC (Fig. 5i). In vitro tests of MNK3 cells with *p65* knockdown exhibited a decrease in OCR, especially basal OCR (Fig. 5j–m). More notably, the knockdown of *p65* resulted in the downregulation of *Il22* expression after stimulation (Fig. 5n). Hence, our findings showed that p65 signaling was critical for the activation of ILC3. Meanwhile, NPM1 bound to p65 and participated in downstream transcriptional regulation in ILC3s in colitis.

To investigate a transcriptional regulatory role for NPM1 in mitochondrial OXPHOS and biogenesis, we examined the expression of the following three mitochondrial transcription factors in ILC3s: *Tfam*, mitochondrial transcription factor B1 (*Tfb1m*) and mitochondrial transcription factor B2 (*Tfb2m*). These transcription factors participate in mtDNA transcription and are stimulated by PGC1α[45]. The expression levels of the three mitochondrial transcription factors in ILC3s showed no differences between *Npm1*^+/+ and *Npm1*^+/− mice in steady state (Fig. 6a–c). However, under pathological conditions, a remarkable decrease in *Tfam* expression was only observed in ILC3s, not macrophages, T cells and epithelial cells, of *Npm1*^+/− mice when compared to *Npm1*^+/+ mice (Fig. 6a–c), suggesting that NPM1 has an indispensable role in upregulation of *Tfam* in ILC3s upon DSS treatment. However, *Tfb1m* and *Tfb2m* were significantly increased in macrophages, T cells and epithelial cells, but not in ILC3s after DSS treatment (Fig. 6a–c). These data suggested that mitochondrial activation in ILC3s is primarily dependent on TFAM rather than on TFB1M or TFB2M. Overexpression of *Tfam* in MNK3 markedly enhanced the expression of mtDNAs, including *mt-Nd1*, *mt-Nd2*, *mt-Nd3*, *mt-Nd4* and *mt-Atp6* (Extended Data Fig. 8j). Knockdown of *Tfam* in MNK3 notably impaired its mitochondrial function and attenuated the production of IL-22 (Fig. 6d–h). Accordingly, NPM1 is crucial for the heightened demand of TFAM to subsequently increase mitochondrial function in ILC3s, not other cell types, during DSS-induced colitis.

To determine if NPM1 and p65 regulate *TFAM* expression, we examined whether they directly bind to the *TFAM* promoter and affect its transcription. We identified four putative binding sites for p65 in the *TFAM* promoter and constructed luciferase reporter plasmids (Fig. 6i–k). Using luciferase reporter assays in HEK293T cells, we found that p65 significantly enhanced *TFAM* promoter-dependent reporter expression in plasmids with either third or fourth binding sites (Fig. 6l). However, knockdown of *NPM1* inhibited promoter activity (Fig. 6l), indicating that NPM1 contributes to *TFAM* transcription.

To validate a direct interaction between p65 and the *Tfam* promoter in ILC3s, we performed chromatin immunoprecipitation (ChIP)

**Fig. 6 | NPM1 regulates *TFAM* transcription by binding to p65. a–c**, RT–PCR analysis of mRNA expression of *Tfam* (**a**), *Tfb1m* (**b**) and *Tfb2m* (**c**) in isolated ILC3s, macrophages, T cells and epithelial cells from *Npm1*^+/+ and *Npm1*^+/− mice exposed to 2.5% DSS or water (steady state; *n* = 3 individual mice). **d–g**, Cell mito stress test was performed with stimulated MNK3 cell line with or without *Tfam* knockdown. Representative OCR (**d**), basal OCR (**e**), ATP production (**f**) and maximal respiration (**g**) are shown (*n* = 4 biological samples). **h**, Expression of *Il22* in unstimulated and stimulated MNK3 cell line (shNC and sh*Tfam*; *n* = 3 biological samples). **i**, Logo plot of the consensus binding motif of the transcription factor p65. **j**, The positions and sequences of the four predicted binding sites of p65 in the *TFAM* promoter. **k**, Diagram of the pGL3-*TFAM* promoter luciferase reporter plasmids. **l**, *TFAM* reporter activity measured in HEK293T cells

(*n* = 4 biological samples). **m**, ChIP–qPCR assays of the binding efficiency of p65 to the *Tfam* promoter in MNK3 cells with or without stimulation by IL-23 and IL-1β. IgG served as the negative control (*n* = 3 biological samples). **n,o**, Analysis of the effect of *Tfam* overexpression (*Tfam*OE) on IL-22 production in MNK3 cells by flow cytometry (**n**) and ELISA (**o**), (*n* = 3 biological samples). **p**, Model depicting transcription activity change of *Tfam* in ILC3 cells with or without NPM1. Data are representative of two independent experiments, shown as the means ± s.e.m., and statistical significance was determined by two-way ANOVA (**a–c,l,m**) and two-tailed unpaired Student's *t* test (**e–h,o**; *P < 0.05, **P < 0.01, ***P < 0.001 and ****P < 0.0001). TSS, transcription start site; shNC, nontargeted short hairpin RNA; sh*Npm1*, short hairpin RNA targeting *Npm1*.

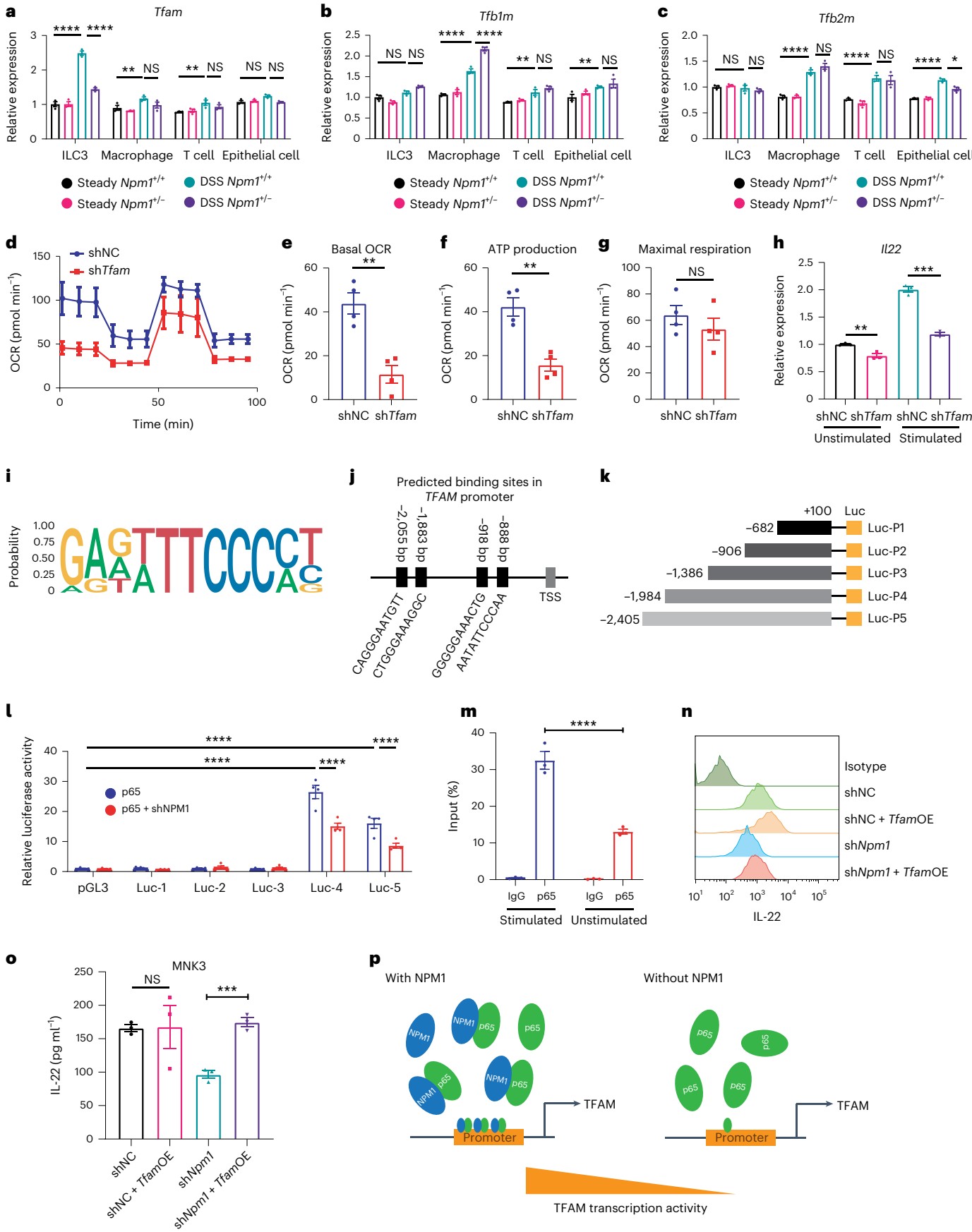

using MNK3 cells and a p65 antibody and tested for the presence of *Tfam* promoter sequences. To determine the effect of inflammatory signals on the interaction, we evaluated MNK3 cells with and without stimulation by IL-1β and IL-23. We found that p65 is bound to the *Tfam* promoter in ILC3s under both conditions, with stimulation enhancing this interaction (Fig. 6m). To confirm the importance of *Tfam* in ILC3 activation, we overexpressed *Tfam* in MNK3 cells and found that expression of *Tfam* mostly restored secretion of IL-22 in MNK3 cells in which *Npm1* was knocked down (Fig. 6n,o). Collectively, our findings indicated that NPM1 acts as a partner of p65 to promote *Tfam* transcription, thereby supporting ILC3 mitochondrial function and activation (Fig. 6p).

### *Npm1* overexpression (*Npm1*OE) protects against DSS-induced colitis

Our subsequent investigation aimed to explain why NPM1 is downregulated in UC and whether overexpression of NPM1 could ameliorate colitis. GATA binding protein 3 (GATA3), interferon regulatory factor 1 (IRF1) and signal transducer and activator of transcription 3 (STAT3), which are predicted transcriptional factors associated with NPM1, demonstrated reduced expression in ILC3s of patients with UC and enteritic mice in comparison to the control groups (Extended Data Fig. 8k,l). These findings may provide insights into the mechanisms underlying the downregulation of NPM1 in IBD. To confirm the protective function of NPM1 in colitis, we generated *Npm1*[UTR−/−] mice that have a genetic knockout of the 3′-UTR region of *Npm1* and overexpress *Npm1* (Supplementary Fig. 1g–j). Compared to control (Ctrl) mice, *Npm1*[UTR−/−] mice had less severe DSS-induced colitis, as evidenced by the increased recovery of body weight, decreased DAI, increased colon length and reduced inflammation (Fig. 7a–e). Although overexpression of *Npm1* did not enhance the frequency of colonic ILC3s, a higher proportion were producing IL-22, indicating that *Npm1*OE enhanced the defense function of ILC3s against colitis (Fig. 7f–j). Expression of various mtDNA was upregulated in *Npm1*[UTR−/−] ILC3 compared to control group (Extended Data Fig. 8m). Overexpression of *Npm1* in MNK3 cells also increased IL-22 secretion (Fig. 7k). Eventually, we crossed *Npm1*[UTR−/−] with *Npm1*[+/−] mice and generated *Npm1*[UTR+/−]*Npm1*[+/−] mice. As expected, heterozygous overexpression of *Npm1* prevented the exacerbated DSS-induced colitis caused by the *Npm1* haploinsufficiency (Fig. 7l–n). ILC3s isolated from *Npm1*[UTR+/−]*Npm1*[+/−] mice also showed increased IL-22 secretion compared with ILC3s from *Npm1*[+/−] mice (Fig. 7o). Taken together, these results demonstrated that *Npm1*OE has a protective function against colitis.

### Discussion

In this study, we demonstrated that NPM1, a protein that is abundant in colonic ILC3s, is critical for the activation of IL-22 production in response to colitis. We found that NPM1 binds to p65 and regulates transcription of the mitochondrial transcription factor *TFAM*, thereby having a role in maintaining mitochondrial biogenesis and OXPHOS. Our findings revealed the protective role of NPM1 in gut homeostasis and suggested that a deficiency in the activity of NPM1 is a key factor linking IBD and MDS/AML.

The NF-κB family of transcription factors has a crucial role in responding to various stimuli by regulating the expression of genes involved in diverse biological processes such as inflammation, metabolism, cancer and development[44]. Here we identified p65 as the top interacting protein with NPM1 in ILC3s in DSS-induced colitis and observed the subcellular translocation of p65 and colocalization with NPM1 in the nucleus of ILC3s to activate downstream gene transcription after inflammatory stimulation. A previous study revealed that NPM1 interacts with the N-terminal DNA-binding domain of p65 and enhances binding to target gene promoters[44]. We also found that TFAM is a potential target of p65 in ILC3s and that p65 regulates *TFAM* transcription in a manner enhanced by NPM1. TFAM is a mitochondrial transcription factor that controls mtDNA replication and transcription[46]. *Tfam*[−/−] mice are embryonically lethal, and tissue-specific deficiency of *Tfam* leads to severe OXPHOS defects, which is the main cause of human mitochondrial diseases[47]. Furthermore, *Tfam*[ΔILC3] mice exhibit a substantial reduction of ILC3s by 6 weeks of age[48]. Here we demonstrated that TFAM is highly expressed in ILC3s and acts as a key downstream effector of NPM1 in DSS-induced colitis. Moreover, mitochondrial activation in ILC3s is primarily dependent on TFAM, rather than on TFB1M or TFB2M. In contrast to ILC3s, macrophages, T cells and epithelial cells are primarily dependent on TFB1M and/or TFB2M in DSS-treated mice. This indicates the indispensable role of NPM1 for ILC3s, not T cells, macrophages or epithelial cells. By maintaining mtDNA replication, mitochondrial number and OXPHOS levels in activated ILC3s, NPM1-stimulated TFAM expression supports the cells' high energy requirements. Although there is no direct evidence that TFAM regulates IL-22, the lack of mitochondrial-derived energy by insufficient TFAM could limit the activation of ILC3s and effector cytokine secretion in colitis. Meanwhile, although it cannot be ruled out that NPM1 affects ILC3 mitochondrial function and cell activation through interactions with other molecules that participate in mitochondria functions, such as NPM1's known partners c-Myc[49,50], SP1 (refs. [51,52]), p53 (refs. [53,54]) and IRF1 (refs. [55,56]), the effects of *p65* knockdown and *Tfam* knockdown on ILC3 function in MNK3 cells are similar to those of knocking down *Npm1*. Therefore, it is believed that the p65-TFAM axis is an important effector for NPM1 to increase the function and metabolism of ILC3.

Mitochondria have an important role in the activation of immune cells. Activation of ETC in mitochondria is essential for T cell activation, expansion and cytokine production[57]. The proliferation and cytokine secretion of ILC3s depend on glycolysis and also mitochondrial ROS following in vitro activation by IL-1β and IL-23 or in vivo during bacterial infection[39]. By analyzing mitochondria in primary ILC3s from *Npm1*[+/−] mice under DSS administration, we observed diminished mitochondrial numbers, consistent with impaired biogenesis, and reduced OXPHOS. To confirm that the deficiency in IL-22 secretion was due to mitochondrial dysfunction, bezafibrate was used to activate mitochondrial function and successfully rescued the production of IL-22 in *Npm1*-deficient ILC3s. Overexpression of *Tfam* also increased IL-22 production, providing additional support for mitochondrial biogenesis and OXPHOS were crucial for ILC3 activation.

**Fig. 7 | *Npm1*OE protects against DSS-induced colitis. a–e**, *Npm1*[UTR−/−] and littermate control mice were fed with 2.5% DSS for 7 days and allowed to recover for 3 days. Body weight (**a**), DAI (**b**), representative colon images (**c**), colon length (**d**) and colon histopathology (**e**) are shown (*n* = 4 individual mice). Scale bars = 500 μm (left) and 100 μm (right). **f,g**, LPLs were isolated from *Npm1*[UTR−/−] and control mice on day 5 of administration of 2.5% DSS (*n* = 5 individual mice). The proportion of ILC3s (live CD45[+]Lin[−]RORγt[+] cells) within the CD45[+] population (**f**) and of IL-22-producing ILC3s (live CD45[+]Lin[−]RORγt[+]IL-22[+] cells) with the ILC3 population (**g**) was determined by flow cytometry. **h,i**, Number of ILC3s (**h**) and IL-22[+] ILC3s (**i**) in LPLs of *Npm1*[UTR−/−] and control mice after DSS administration are depicted (*n* = 5 individual mice). **j,k**, IL-22 production by isolated colonic ILC3s from *Npm1*[UTR−/−] and control mice on day 5 of administration of 2.5% DSS (**j**) and MNK3 cells in response to *Npm1*OE (**k**) was determined by ELISA (*n* = 3 biological samples). **l–n**, Control, *Npm1*[UTR+/−], *Npm1*[+/−] and *Npm1*[UTR+/−]*Npm1*[+/−] mice were administered 2.5% DSS for 7 days and allowed to recover for 3 days. Representative images of the mouse colons (**l**), colon length (**m**) and colon histopathology (**n**) are shown (*n* = 5 individual mice). Scale bars = 500 μm (top) and 100 μm (bottom). **o**, IL-22 production by isolated colonic ILC3s from the indicated groups of mice was determined by ELISA (*n* = 3 individual mice). Data in **d**, **f–k**, **m** and **o** are representative of two independent experiments, shown as the means ± s.e.m., and statistical significance was determined by two-tailed unpaired Student's *t* test (**P* < 0.05, ***P* < 0.01 and ****P* < 0.001).

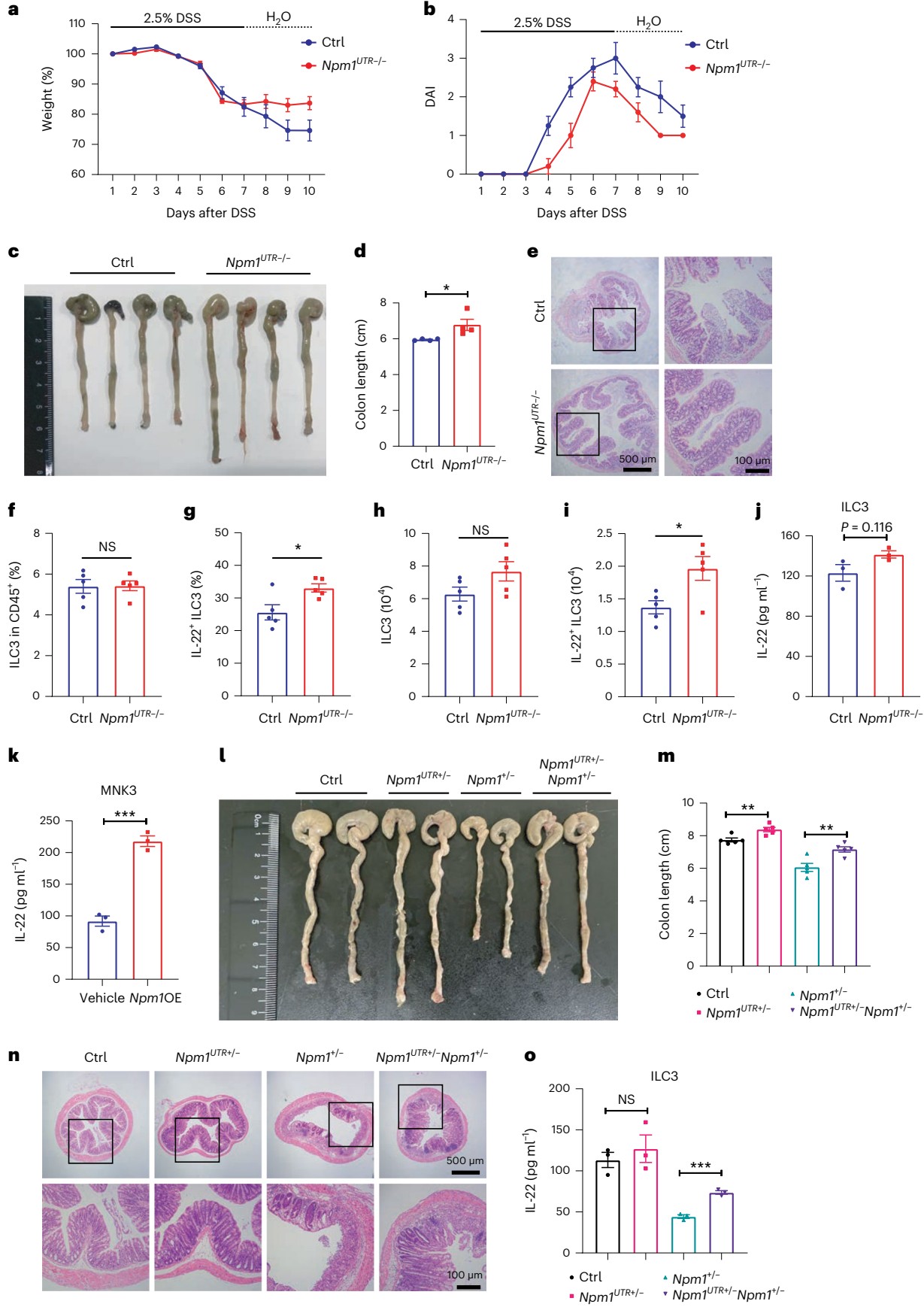

MDS is a hematopoietic disorder involving clonal abnormalities of cells caused by mutations in oncogenes and tumor suppressor genes, as well as chromosomal abnormalities[9]. The subsequent alterations in the function and properties of BM-derived immune cells can lead to the development of immune-mediated disorders including IBD. Genetic mutations provide insight into the relationship between MDS and IBD. For example, mutations in *PTPN11*, a driver gene in MDS/AML, result in exacerbation of intestinal inflammation by disrupting BM-derived macrophage responsiveness to IL-10 (ref. [58]). NPM1 acts as a top driver mutation in high-risk MDS and AML[6,17]. In our study, we demonstrated that NPM1 functions in BM-derived ILC3s to control the gut microenvironment, particularly through a protective IL-22-related immune response. Our data provide insight with potential relevance for the diagnosis and treatment of patients with concurrent IBD and MDS.

In summary, our study highlights the role of NPM1 in maintaining mitochondrial function and IL-22 production in ILC3s in the progression of colitis. Our findings suggest that NPM1 might be a therapeutic target for IBD and provide insights into a connection between MDS/AML and IBD.

## Online content

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

[1]Suzhou Institute of Biomedical Engineering and Technology, Chinese Academy of Science, Suzhou, China. [2]School of Biomedical Engineering (Suzhou), Division of Life Sciences and Medicine, University of Science and Technology of China, Hefei, China. [3]Suzhou Hospital, Affiliated Hospital of Medical School, Nanjing University, Suzhou, China. [4]Advanced Medical Research Institute, Shandong University, Jinan, China. [5]Department of Biochemistry and Molecular Biology, School of Basic Medical Sciences, Cheeloo College of Medicine, Shandong University, Jinan, China. [6]The First Rehabilitation Hospital of Shanghai, Brain and Spinal Cord Innovation Research Center, School of Medicine, Advanced Institute of Translational Medicine, Tongji University, Shanghai, China. [7]CAM-SU Genomic Resource Center, Soochow University, Suzhou, China. [8]Shanghai Key Laboratory of Anesthesiology and Brain Functional Modulation, Translational Research Institute of Brain and Brain-Like Intelligence, Shanghai Fourth People's Hospital, School of Medicine, Tongji University, Shanghai, China. [9]These authors contributed equally: Rongchuan Zhao, Jiao Yang. ✉e-mail: changgeng.peng@tongji.edu.cn; lishiyang@sdu.edu.cn; minxuan.sun@sibet.ac.cn

## Methods

### Generation of *Npm1*⁺/⁻, *Npm1*^{UTR+/−} and *Npm1*^{flox/+} mice

*Npm1*⁺/⁻ and *Npm1*^{UTR+/−} mice were generated by knocking out the DNA-binding domain (including partial exon 8, exon 9, 10 and partial exon 11) and 3′-UTR domain with the binding sites of microRNAs using CRISPR−Cas9 technology from the CRO company Shanghai Model Organisms Center. In brief, Cas9 mRNA and gRNA were synthesized in vitro and then injected into fertilized eggs of C57BL/6J mice. The resulting F0 mice were screened for *Npm1*⁺/⁻ genotype using specific PCR primers (PI, 5′-GAAAAGGTCCCAGTGAAGAAAGTGA-3′; PII, 5′-TGGCAAGTGAACCTGGACAACAT-3′; PIII, 5′-GGCTGAC CCACAGGCTGAGGAG-3′ and PIV, 5′-CCAACAGATTGGCTATCAA TAGAGGA-3′) or *Npm1*^{UTR+/−} (PI, 5′-CCACAGGCTGAGGAGGCAACAC-3′; PII, 5′-AAAAGGTTCAGGCACGAAGCAG-3′; PIII, 5′-GTCAGATGTGG AAATGGTAGGGAGA-3′ and PIV, 5′-AAAAGGTTCAGGCACGAAGCAG-3′) and crossed with WT C57BL/6J mice to get F1 heterozygous mice which were identified by genotyping PCR. F1 heterozygous mice were crossed with WT C57BL/6J mice to get F2 heterozygous mice. The third and further generations of *Npm1*⁺/⁻ and *Npm1*^{UTR+/−} mice were used in the experiments.

*Npm1*^{flox/+} mice were generated by introducing two loxp sequences into *Npm1* using CRISPR−Cas9 technology from the CRO company Cyagen Biosciences. Briefly, Cas9 mRNA and gRNA were synthesized in vitro with a homologous arms-encompassed targeting vector and injected into fertilized eggs of C57BL/6J mice. The resulting F0 mice were screened using specific PCR primers (FI, 5′-AACAGCTAGAT GGGAAGTATGGA-3′; RI, 5′-AGTTCCCAAGTTTGCTTTGAACAG-3′ and FII, 5′-ACGTTGCAGATAGCTGTACTGATG-3′; RII, 5′-GCTAAAGC GAATCTTGTCTGTTCA-3′) and crossed with WT C57BL/6J mice to get F1 heterozygous mice, which were identified by genotyping PCR with primer pairs (F2 and R2). Positive F1 *Npm1*^{flox/+} mice were crossed with WT C57BL/6J mice to get F2 heterozygous mice. The third and further generations of *Npm1*^{flox/+} mice were used in the experiments.

### Mice

All mice used in this study were bred in the animal facility of Suzhou Institute of Biomedical Engineering and Technology and Shandong University and were approved in accordance with the Institutional Animal Care and Use Committee guidelines at Suzhou Institute of Biomedical Engineering and Technology and Shandong University. Mice were housed in individually ventilated cages under a 12-h light/12-h dark cycle with normal food and water. All experiments were performed using C57BL/6J mice, which also served as controls for *Npm1*⁺/⁻, *Npm1*^{UTR−/−} and *Apc*^{min/+} mice. *Npm1*^{flox/flox} mice served as controls for *Rorc*^{cre/+}*Npm1*^{flox/flox} and *Villin*^{cre/+} *Npm1*^{flox/flox} mice. Male mice aged 6−8 weeks were used for the experiments. For the DSS model, drinking water containing 2.5% DSS was given to age-matched male mice for 7 days, followed by regular water for 3 days, with DSS water being replaced each day. For the rescue experiment, mice were treated with 10 mg kg⁻¹ bezafibrate (i.g.) every other day. Throughout the experiment, body weight was monitored. To induce colon cancer model, WT and *Npm1*⁺/⁻ mice were injected intraperitoneally with AOM (10 mg kg⁻¹). After 5 days, 2.5% DSS was added to the drinking water for seven consecutive days, followed by 14 days of regular water. This cycle was repeated three times. Mice were killed for analysis on day 65 of the experiments. In the TNBS model, mice were anesthetized and then treated with 2 mg of TNBS dissolved in 50% ethanol via rectal administration using a polyethylene catheter (2 mm in outer diameter). Following administration, the mice were maintained in an inverted position for a minimum of 1 min. Control mice were treated rectally with 50% ethanol alone. The progression of colitis was monitored daily, assessing parameters such as diarrhea, presence of blood in stools, body weight and survival rates. Note that littermate mice are generally genotyped at 3−4 weeks of age and then placed in separate cages when grouping, according to their genotype. The DAI is calculated by combining the following three parameters: the percentage weight loss of the mice, the consistency of stool and the presence of stool blood. The scoring for each parameter is as follows: (1) weight loss−0 points if weight remains stable, 1 point for a 1−5% weight loss, 2 points for a 5−10% weight loss, 3 points for a 10−15% weight loss and 4 points for a weight loss greater than 15%; (2) stool consistency−0 points for normal stool, 2 points for loose stool and 4 points for diarrhea and (3) stool blood−0 points for no blood, 2 points for occult blood positivity and 4 points for overt bleeding. The DAI is calculated as follows: DAI = (weight loss index + stool consistency + blood in stool)/3. Note that mice are generally genotyped and caged at 3−4 weeks of age and then placed in separate cages when grouping, according to their genotype.

### Generation of BM chimera

The generation of BM chimeras was achieved by collecting BM cells from both WT and *Npm1*⁺/⁻ mice and subsequently flushing them with 1× PBS. The cell suspension, comprising $1 \times 0^7$ BM cells, was then intravenously injected into lethally irradiated recipient mice of both WT and *Npm1*⁺/⁻ genotypes, with a dose of 102.2 cGy min⁻¹ for 9 min. Experiments were conducted 4 weeks following reconstitution.

### In vivo T cell and myeloid cell blocking

To deplete T cells, anti-CD3ε (Bio X Cell, BE0001-1; clone 145-2C11) was administered intravenously daily (50 µg per mouse, from day −2 to day 6), and control mice were administered an equivalent amount of IgG (Bio X Cell, BE0091). To deplete myeloid cells, antimouse/antihuman CD11b (Bio X Cell, BE0007; clone M1/70) were administered intravenously every 2 days (100 µg per mouse, from day −2 to day 6), and control mice were administered an equivalent amount of IgG (Bio X Cell, BE0091; clone LTF-2). The DSS-induced colitis model was initiated on day 0.

### Histology

We dissected the colons from the indicated mice, fixed them in 10% formalin and stained them with hematoxylin and eosin (H&E) using paraffin-embedded sections. We used the following scoring system to evaluate colon tissue histologically: 0 = no evidence of inflammation, 1 = low level of inflammation with scattered infiltrating mononuclear cells (1−2 foci), 2 = moderate inflammation with multiple foci, 3 = high level of inflammation with increased vascular density and marked wall thickening and 4 = maximal inflammation with transmural infiltration and loss of goblet cells.

### Flow cytometry and isolation of lamina propria leukocytes

To isolate leukocytes from the lamina propria, we incubated intestinal segments of approximately 0.5 cm at 37 °C for 1.5 h in complete Roswell Park Memorial Institute (RPMI) medium (Suzhou Haixing Biosciences), supplemented with DNase I (150 µg ml⁻¹; Sigma) and collagenase VIII (300 U ml⁻¹; Sigma). The digested fragments were triturated and filtered through a 100 µm cell strainer. The cells were collected from the interface of the 80% and 40% Percoll gradients after centrifugation at 660*g* for 15 min at room temperature. Before surface staining, Fc receptors were blocked using CD16/32 antibody (eBioscience; dilution 1:100). Leukocytes isolated from the intestinal lamina propria were then stained with antibodies against the following markers: CD45 eFluor 506 (dilution 1:100), RORγt PE (dilution 1:50), Ly-6G PE (dilution 1:100), CD127 Super Bright 645 (dilution 1:100), F4/80 FITC (dilution 1:100), CD3 Alexa-488 (dilution 1:100), CD34 FITC (dilution 1:100), CD117 APC (dilution 1:100), CD19 eFluor (450 dilution 1:100), IL-22 PE (dilution 1:50), CD4 APC (dilution 1:100), IL-17A BV421 (dilution 1:50), Lineage Percp-cy5.5 Cocktail (dilution 1:50), T-bet PE (dilution 1:100), IFNγ-APC (dilution 1:50), NKp46-PerCPcy5.5 (dilution 1:50), FOXP3-eFluor 450 (dilution 1:100), CCR6-BV421 (dilution 1:50), TCR γ/δ-APC (dilution 1:50) and CD127-FITC (dilution 1:100). For cytokine staining, cells were stimulated with phorbol 12-myristate 13-acetate (PMA) (50 ng ml⁻¹) and

ionomycin (500 ng ml$^{-1}$) for 2 h, along with the addition of brefeldin A (2 µg ml$^{-1}$). Live and dead cells were distinguished using the Live and Dead Violet Viability Kit (BioLegend).

## Smart-seq

Live$^+$Lin$^-$CD45$^{low}$CD90.2$^{high}$ ILC3s were sorted from colon LPLs of the indicated mice. The SMARTer cDNA synthesis protocol was used to synthesize cDNA, which was then fragmented using dsDNA Fragmentase (New England Biolabs (NEB), M0348S) and incubated at 37 °C for 30 min. Library construction commenced with fragmented cDNA, where blunt-end DNA fragments were generated through a combination of fill-in reactions and exonuclease activity. Size selection was carried out using the provided sample purification beads. An A-base was added to the blunt ends of each strand, indexed Y adapters were ligated to the fragments and the ligated products were amplified using PCR. Subsequently, paired-end sequencing was conducted on NovaSeq 6000 (Illumina), following the protocol recommended by the vendor.

## scRNA-seq data processing

scRNA-seq dataset (GSE182270) was downloaded from the Gene Expression Omnibus (GEO) database and was performed on cells extracted from colonic biopsies of inflamed mucosa (patients with UC, n = 5) and normal colonic mucosa (HCs, n = 4). Count tables were analyzed using the Seurat 4.0 package following the standard workflow with default settings. The number of principal components (PCs) was determined based on Elbow plots, PCs = 13. Next, FindNeighbors and FindClusters functions were used for cell clustering, and the UMAP method was used for visualization. Cell-type-specific markers were found by the FindMarkers function; cell-type identities were manually annotated by matching cluster-specific upregulated marker genes with cell-type markers in the CellMarker 2.0 database. *NPM1*$^{low}$ ILC3s and *NPM1*$^{high}$ ILC3s were identified using a median expression cutoff for *NPM1* in ILC3s. Note that the cell dropout of NPM1 was not included in the analysis. FindMarkers function was used to identify significantly regulated genes in *NPM1*$^{high}$ ILC3. The ClusterProfiler package was applied for functional annotation.

## Immunoprecipitation (IP) and western blot analysis

To perform IP, cells were lysed in an IP lysis buffer containing 20 mmol l$^{-1}$ Tris (pH 7.5), 150 mmol l$^{-1}$ NaCl and 1% Triton X-100, supplemented with a cocktail of protease and phosphatase inhibitors. Following lysis, the supernatants were collected after centrifugation and incubated overnight at 4 °C with constant rotation with the indicated antibodies. The antibody–antigen complexes were then precipitated using protein A/G magnetic beads (Millipore) and washed with PBS. For western blot analysis, cell lysates were prepared using radio immunoprecipitation assay (RIPA) lysis buffer (CoWin Biosciences) containing protease inhibitors and phosphatase inhibitors (CoWin Biosciences). Equal amounts of protein were loaded onto SDS–PAGE gels and transferred to nitrocellulose membranes. The membranes were blocked with 5% non-fat dried milk for 1 h at room temperature before being incubated with primary antibodies overnight at 4 °C, including SDHB (Proteintech; 1:2,000), NDUFB8 (Proteintech; 1:2,000), MT-ATP6 (Abclonal; 1:1,000), MT-CO1 (Abclonal; 1:1,000), UQCRC2 (Proteintech; 1:1,000), NPM1 (Abclonal; 1:1,000) and p65 (Cell Signaling Technology (CST); 1:1,000). After washing, the membranes were incubated with IRDye 800cw or 680cw conjugated secondary antibodies (LICORbio; 1:10,000) for 1 h. The membranes were then imaged using an Odyssey CLx Infrared Imaging System.

## ChIP

The ChIP assay was conducted using the ChIP-IT Kit (Beyotime). In brief, the cells were initially fixed with formaldehyde and subsequently lysed. To precipitate the DNA fragment, either 2 µg of anti-p65 or normal IgG were used. The DNA–protein complexes were then pulled down with magnetic beads and subjected to decross-linking. The extracted DNA samples were finally amplified using specific *Tfam* promoter primers for the sequences containing the binding site 5′-GGGAAAGGC-3′.

## Luciferase reporter assay

HEK293T cells that overexpressed p65 were transfected with the specified pGL3-luciferase reporter plasmid that contained the TFAM promoter, along with the Renilla pRL-TK plasmid as the internal control. After incubation for 24–48 h, the cell lysates were subjected to luciferase activity analysis using the Dual-Luciferase Reporter Assay kit (Promega).

## Immunofluorescence staining

Glass slides were pre-inserted into 12-well plates, and cells were seeded onto these plates. After 24 h, when the cells had reached 40–50% confluence, they were washed with PBS and subsequently fixed with 4% paraformaldehyde for 30 min. The cells were then permeabilized with a 0.5% Triton X-100 solution for an additional 20 min. A blocking buffer containing 5% bovine serum albumin was added next. Primary antibodies comprising anti-NPM1 (Proteintech; 1:200), anti-RORγt (Thermo Fisher Scientific; 1:200), anti-TOMM20 (Proteintech; 1:200) and anti-p65 (CST; 1:200) were used in this experiment. The corresponding secondary antibodies conjugated with Alexa Fluor 488, 555 and 647 were also used at a concentration of 1:2,000 (Invitrogen).

## MS

The Q Exactive Mass Spectrometer (Thermo Fisher Scientific) and Dionex Ultimate 3000 RSLCnano (Thermo Fisher Scientific) were used to analyze the affinity-purified samples according to the manufacturer's instructions. The proteins were first reduced with 0.05 M Tris (2-carboxyethyl) phosphine (TCEP) and then alkylated with 55 mM methyl methanethiosulfonate (MMTS). The sample was then centrifuged and subjected to centrifugation steps before being digested with trypsin. After digestion, the resulting peptides were loaded onto a reversed-phase analytical column and underwent high-performance liquid chromatography (HPLC)–MS analysis. The peptide detection was conducted using an Orbitrap at a resolution of 70,000, with tandem mass spectrometry (MS/MS) using normalized collision energy (NCE) setting as 27, and MASCOT software was used to identify proteins. The peptide mass tolerance was 20 ppm, while the fragment mass tolerance was 0.6 Da, and the significance threshold was 0.05.

## RNA extraction and quantitative real-time PCR analysis

The procedures were conducted as described previously[59]. Supplementary Table 2 lists all PCR primer sequences used for the detection of *mt-Co1*, *mt-Co2*, *mt-Co3*, *mt-Nd1*, *mt-Nd2*, *mt-Nd3*, *mt-Nd4*, *mt-Atp6*, *Cxcl2*, *Ccl4*, *Xiap*, *cFlip*, *Tfam*, *Tfb1m*, *Tfb2m*, *Il22*, *Npm1*, *Gata3*, *IRF1*, *Stat3*, *Tjp1*, *Tjp2*, *Cldn2*, *Cldn3* and *Gapdh*. Moreover, *Tfam* ChIP–qPCR primers are also listed in Supplementary Table 2.

## SEM

The cells were centrifuged to precipitate, and the medium was removed. Then 500 µl of 2.5% glutaraldehyde (Ted Pella) was slowly added, avoiding suspending the precipitated cells, and left at room temperature for 1 h, followed by keeping at 4 °C for 3 h. The glutaraldehyde solution was replaced with PBS, and cells were left at 4 °C overnight. The cells were stained following the reported protocol[60]. The cells were then embedded in resin (Eponate 12 Kit, Ted Pella). Ultrathin sections of 50 nm thickness were cut (UC7, Leica) and collected on carbon-coated Kapton tapes. EM images were acquired with an SEM (GeminiSEM 300, Zeiss).

## Mitochondrial membrane potential assay

Primary ILC3s were loaded with the JC-1 primer (Beyotime, C2006) and potentiometric dye TMRE (Beyotime, C2001S) at 37 °C for 20 min and washed with buffer or cell medium three times. $\Delta\psi m$ was measured

using a microplate reader. When detecting JC-1 monomers, the excitation light can be set to 490 nm and the emission light can be set to 530 nm. When detecting JC-1 polymer, the excitation light can be set to 525 nm and the emission light can be set to 590 nm. The maximum excitation wavelength of TMRE is 550 nm, and the maximum emission wavelength is 575 nm.

## Immunohistochemical staining

Immunohistochemical staining was performed as follows: after deparaffinization and hydration, paraffin slides were repaired by boiling in Tris–EDTA buffer (pH 8.0) for 10 min. Next, sections were treated with 3% $H_2O_2$ for 20 min to bleach endogenous peroxidase. After blocking with donkey serum, primary antibodies against NPM1 were diluted 1:100 and then incubated at 4 °C overnight. After three washes with PBS, the tissue slides were treated with horseradish peroxidase (HRP), conjugated donkey anti-rabbit or mouse secondary antibody (Dako) for 45 min and then stained by 3,3′-diaminobenzidine (DAB). Semi-quantitative immunohistochemistry is generally divided into the following three levels: low (+), medium (++) and high (+++). These levels are scored as follows: low (+) = 1, medium (++) = 2 and high (+++) = 3. Then calculate the value based on (+)% × 1 + (++)% × 2 + (+++)% × 3. The final score is (+) for a value less than 1.0, (++) for a value between 1.0 and 1.5 and (+++) for a value greater than 1.5.

## Seahorse metabolic analysis

Cellular OCR was quantified using the Agilent Seahorse XFe24, following the manufacturer's protocol. Primary ILC3s, macrophages, T cells, epithelial cells and MNK3 cells were plated on 24-well plates precoated with poly-D-lysine and incubated with the complete RPMI medium over night. Following the incubation period, the cells were washed and transferred into seahorse assay medium supplemented with 1 mM pyruvate, 2 mM glutamine and 10 mM glucose and cultured for an additional hour at 37 °C in a $CO_2$-free environment. To measure OCR, indicated inhibitors such as oligomycin (1.5 μM), carbonyl cyanide-4-(trifluoromethoxy) phenylhydrazone (carbonyl cyanide 4-(trifluoromethoxy) phenylhydrazone (FCCP), 1 μM), rotenone (0.5 μM) and antimycin A (0.5 μM) were introduced where specified, and the rates of OCR (pmol $O_2$ per min) were monitored in real-time.

## Ethics

Pathology sections were obtained from patients with UC, patients with CD and healthy individuals after approval was obtained from the Ethics Committee of Shandong University School of Basic Medicine (ECSBMSSDU2020-1-035). All animal experiments were approved and are in accordance with the Institutional Animal Care and Use Committee guidelines at Suzhou Institute of Biomedical Engineering and Technology (2021-C058) and Shandong University (ECSBMSSDU2020-2-057).

## Randomization and blinding

For DSS/TNBS/AOM-DSS animal studies to assess the changes in *Npm1* deficiency, no method of randomization was used. Mice were grouped according to genotype, and all experiments were performed with sex-matched littermates. For the bezafibrate experiment, mice were first grouped by genotype and then randomly assigned to two groups (bezafibrate-treated group and control group). For CD11b/CD3 antibody treatment experiments, mice were first grouped by genotype and then randomly assigned to two groups (CD11b/CD3 antibody-treated group and IgG antibody-treated group). Animal studies were not blinded (mice were named with mouse ID and genotyped within 6 weeks of birth). Group allocation was not applicable because mice were grouped based on and compared across different genotypes. Histological analyses were conducted by two independent investigators, who had limited knowledge of the group of mice and patients. DAI was analyzed by two independent investigators,

who had limited knowledge of the group of mice and patients. Data from FACS, qPCR and enzyme-linked immunosorbent assay (ELISA) were collected by an investigator with only the knowledge of mouse ID (without grouping information).

## Statistical analysis

Flow cytometry data were analyzed by FlowJo (v10), and immunofluorescence images were analyzed by Image J 64-bit Java 8. Statistical analyses were performed using GraphPad Prism (v8). Data were presented as mean ± s.e.m. Statistical significance was assessed by Student's *t* test (unpaired) or two-way analysis of variance (ANOVA) analyses. All statistical tests were two-tailed, and a *P* value <0.05 was considered statistically significant. Data distribution was assumed to be normal, but this was not formally tested. The number of patients, mice and biological repeats are indicated in the figure legends, as well as the number of independent experiments. No animal or data point was excluded from the analyses. In all studies using at least three to five animals per group, all experiments were performed at least twice to ensure reproducibility.

## Reporting summary

Further information on research design is available in the Nature Portfolio Reporting Summary linked to this article.

## Data availability

Smart-seq analysis of primary colonic ILC3 in *Npm1*[+/+] and *Npm1*[+/−] mice have been deposited in the GEO under the accession code GSE271455. scRNA-seq data from patients with UC and healthy individuals was downloaded from GEO under the accession code GSE182270. Source data are provided with this paper.

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

## Acknowledgements

We thank Z. Zhang's laboratory, W. Feng and H. Zhu for their help and suggestions. Advanced Optical Microscopy Application Open Platform in Suzhou Institute of Biomedical Engineering and Technology (SIBET) is acknowledged for professional assistance in image acquisition. We thank N.R. Gough (BioSerendipity) for professional editing services. This work was funded by the Chinese Academy of Sciences (CAS) Project for Young Scientists in Basic Research (grant YSBR-067 to G.S.), the Basic Research Pilot Program of Suzhou (SJC2022007 to M.S.), National Natural Science Foundation of China (82273336 to M.S. and 82071854 to S.L), Pre-Research and Construction Project of Major Scientific Research Facilities in Jiangsu Province (BM2022010 to M.S.), National Key R&D Program of China (2020YFA0804400 to S.L.) and Innovative Research Group Project (82321002 to S.L.).

## Author contributions

R.C.Z., J.Y., S.Y.L. and M.X.S. conceptualized the project. R.C.Z. and J.Y. developed the methodology. R.C.Z., Y.J.Z., H.Z., J.L.P., L.H., D.S.L. and J.Y.Y. performed validation. Y.S.Z. and D.T.Y. carried out a formal analysis. R.C.Z., Y.J.Z., H.Z. and L.H. conducted the investigation. M.X.S., S.Y.L., Y.X.L., D.J.P., R.L.X. and C.G.P. arranged the resources. R.C.Z., Y.S.Z. and D.T.Y. curated the data. R.C.Z. wrote the original draft. J.Y., S.S., S.Y.L., D.T.Y., C.G.P. and M.X.S. did the writing, reviewing and editing of the manuscript. M.X.S., S.Y.L., C.G.P., R.B.Z. and G.H.S. provided supervision.

## Competing interests

The authors declare no competing interests.

## Additional information

**Extended data** is available for this paper at

**Supplementary information** The online version
contains supplementary material available at

**Correspondence and requests for materials** should be addressed to
Changgeng Peng, Shiyang Li or Minxuan Sun.

**Peer review information** *Nature Immunology* thanks Matthew Hepworth
and the other, anonymous, reviewer(s) for their contribution to the peer
review of this work. Primary Handling Editor: Nick Bernard, in collaboration
with the *Nature Immunology* team. Peer reviewer reports are available.

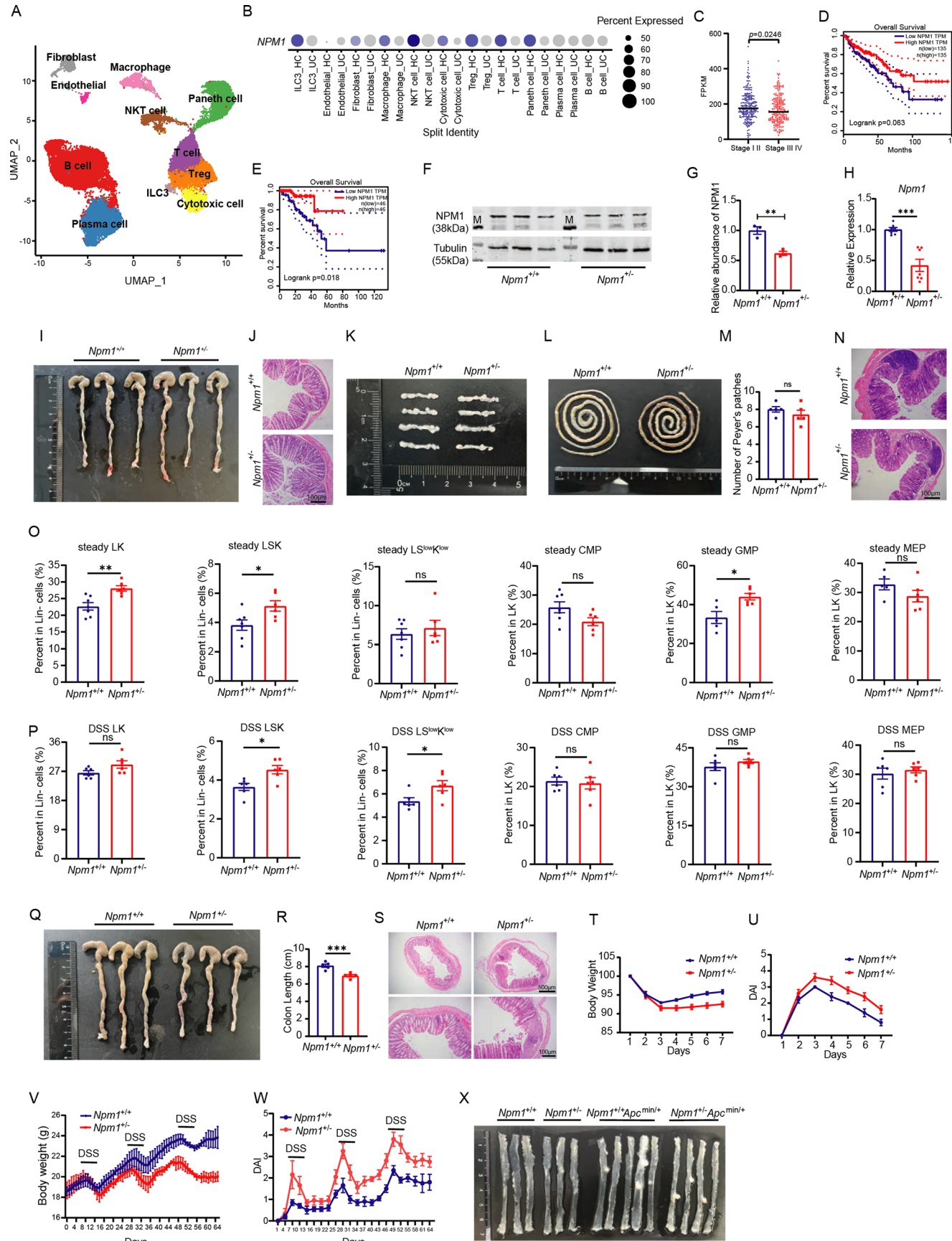

Extended Data Fig. 1 | See next page for caption.

**Extended Data Fig. 1 | NPM1 deficiency increases susceptibility to enteritis and colonic adenocarcinoma.** (**a**) UMAP analysis of GSE182270 shows different clusters of cells in human colonic biopsies. (**b**) Expression of *NPM1* in different clusters (y axis). Dot size represents the fraction of cells within the cluster that express each *NPM1*. Colors indicate the z-scaled expression of genes in cells within each cluster. (**c**) Expression of *NPM1* in different stages of 425 COAD patients in TCGA. (Stages I and II: n = 238 individual patients; stages III and IV: n = 187 individual patients). (**d**) Survival analysis of *NPM1* in 270 COAD patients in TCGA, cutoff by the median expression of *NPM1* in groups. (**e**) Survival analysis of *NPM1* in 92 READ patients in TCGA, cutoff by the median expression of *NPM1* in groups. (**f,g**) Protein expression analysis of NPM1 in the colon of *Npm1*<sup>+/+</sup> and *Npm1*<sup>+/-</sup> mice. Quantitative analysis of protein levels of NPM1, relative to tubulin (**e**) (n = 3 individual mice). (**h**) RT-PCR analysis of mRNA expression of *Npm1* in whole colon of *Npm1*<sup>+/+</sup> and *Npm1*<sup>+/-</sup> mice (n = 7 individual mice). (**i**) Representative images of colons from *Npm1*<sup>+/+</sup> and *Npm1*<sup>+/-</sup> mice in steady state. (**j**) H&E staining of colon from *Npm1*<sup>+/+</sup> and *Npm1*<sup>+/-</sup> mice in steady state. Scale bars: 100 µm. (**k**) Comparison of mesenteric lymph nodes from *Npm1*<sup>+/+</sup> and *Npm1*<sup>+/-</sup> mice in steady state (n = 4). (**l,m**) Representative images of Peyer's patches from *Npm1*<sup>+/+</sup> and *Npm1*<sup>+/-</sup> mice in steady state (**l**). Analysis of the number of Peyer's patches (**m**) was performed (n = 5 individual mice). (**n**) H&E staining of solitary intestinal lymphoid tissue from *Npm1*<sup>+/+</sup> and *Npm1*<sup>+/-</sup> mice in steady state. Scale bars: 100 µm. (**o**) Ratio of LK, LSK, Lin<sup>-</sup>Sca1<sup>low</sup>, Lin<sup>-</sup>Sca1<sup>low</sup> CD117<sup>low</sup> cells (*Npm1*<sup>+/+</sup>: n = 7 individual mice; *Npm1*<sup>+/-</sup>: n = 6 individual mice), CMP, GMP and MEP (*Npm1*<sup>+/+</sup>: n = 5 individual mice; *Npm1*<sup>+/-</sup>: n = 6 individual mice) in bone marrow from *Npm1*<sup>+/+</sup> and *Npm1*<sup>+/-</sup> mice under steady-state. (**p**) Ratio of LK, LSK, Lin<sup>-</sup>Sca1<sup>low</sup> CD117<sup>low</sup> cells, CMP, GMP and MEP in bone marrow from *Npm1*<sup>+/+</sup> and *Npm1*<sup>+/-</sup> mice under steady-state (n = 6 individual mice). (**q–u**) *Npm1*<sup>+/-</sup> and control *Npm1*<sup>+/+</sup> mice were administered trinitrobenzene sulfonic acid (TNBS). Colon length (**q,r**), colon histopathology on day 7 (**s**), body weight (**t**) and DAI (**u**) were analyzed (n = 5 individual mice). Scale bars: 500 µm (up), 100 µm (down). (**v,w**) *Npm1*<sup>+/-</sup> and control *Npm1*<sup>+/+</sup> mice were treated with AOM-DSS for 65 days, and body weight (**v**) and DAI (**w**) were analyzed (n = 5 individual mice). (**x**) Representative images of colons from CRC mouse model. Data in **c, g, h, m, o, p** and **r** are representative of two independent experiments, shown as the means ± s.e.m., and statistical significance was determined two-tailed unpaired Student's *t*-test (*$p < 0.05$, **$p < 0.01$ and ***$p < 0.001$).

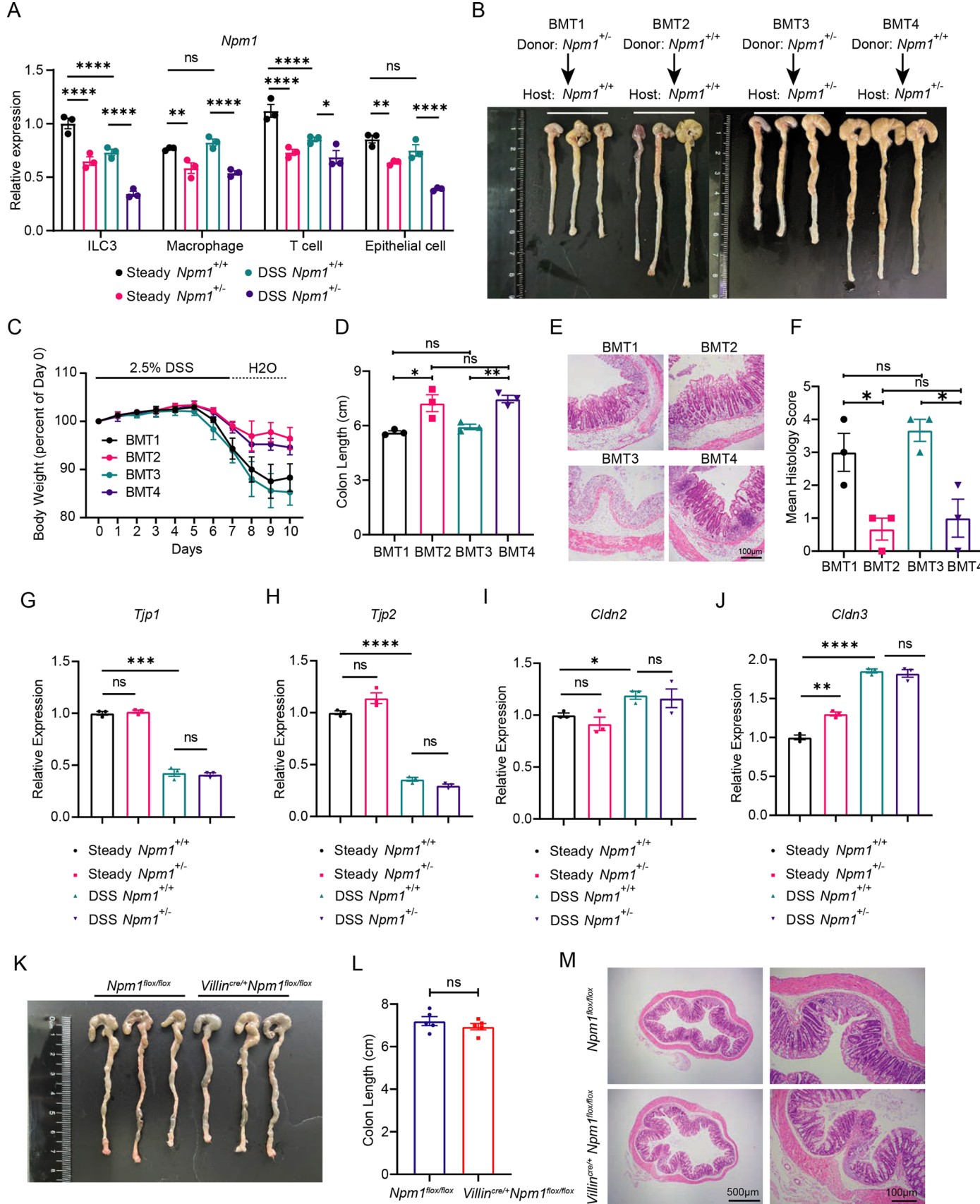

**Extended Data Fig. 2 | See next page for caption.**

**Extended Data Fig. 2 | NPM1 in hematopoietic system is essential to colon against colitis. (a)** RT-PCR analysis of mRNA abundance of *Npm1* in ILC3s, macrophages, T cells and epithelial cells from *Npm1*[+/+] and *Npm1*[+/−] mice exposed to 2.5% DSS or water (steady state) (n = 3 individual mice). **(b–f)** Bone marrow chimeric mice of indicated genotypes were treated with 2.5% DSS water for 7 days, and representative images of the mouse colons on day 10 of the DSS model **(b)**, body weight **(c)**, colon length **(d)** and histopathology **(e,f)** were analyzed (n = 3 individual mice). Scale bars represent 100 μm. **(g–j)** RT-PCR analysis of mRNA expression of the indicated genes in epithelial cells from *Npm1*[+/+] and

*Npm1*[+/−] mice exposed to 2.5% DSS or water (steady state) (n = 3 individual mice). **(k–m)** *Villin*[cre/+]*Npm1*[+/−] and control *Npm1*[flox/flox] mice were treated with 2.5% DSS for 7 days, and representative images **(k)**, colon length **(l)** and histopathology **(m)** were analyzed (n = 5 individual mice). Scale bars: 500 μm (left), 100 μm (right). Data are representative of two independent experiments, shown as the means ± s.e.m., and statistical significance was determined by two-way ANOVA **(a)** and two-tailed unpaired Student's *t*-test **(d,f–j,l)** (*$p < 0.05$, **$p < 0.01$, ***$p < 0.001$, ****$p < 0.0001$).

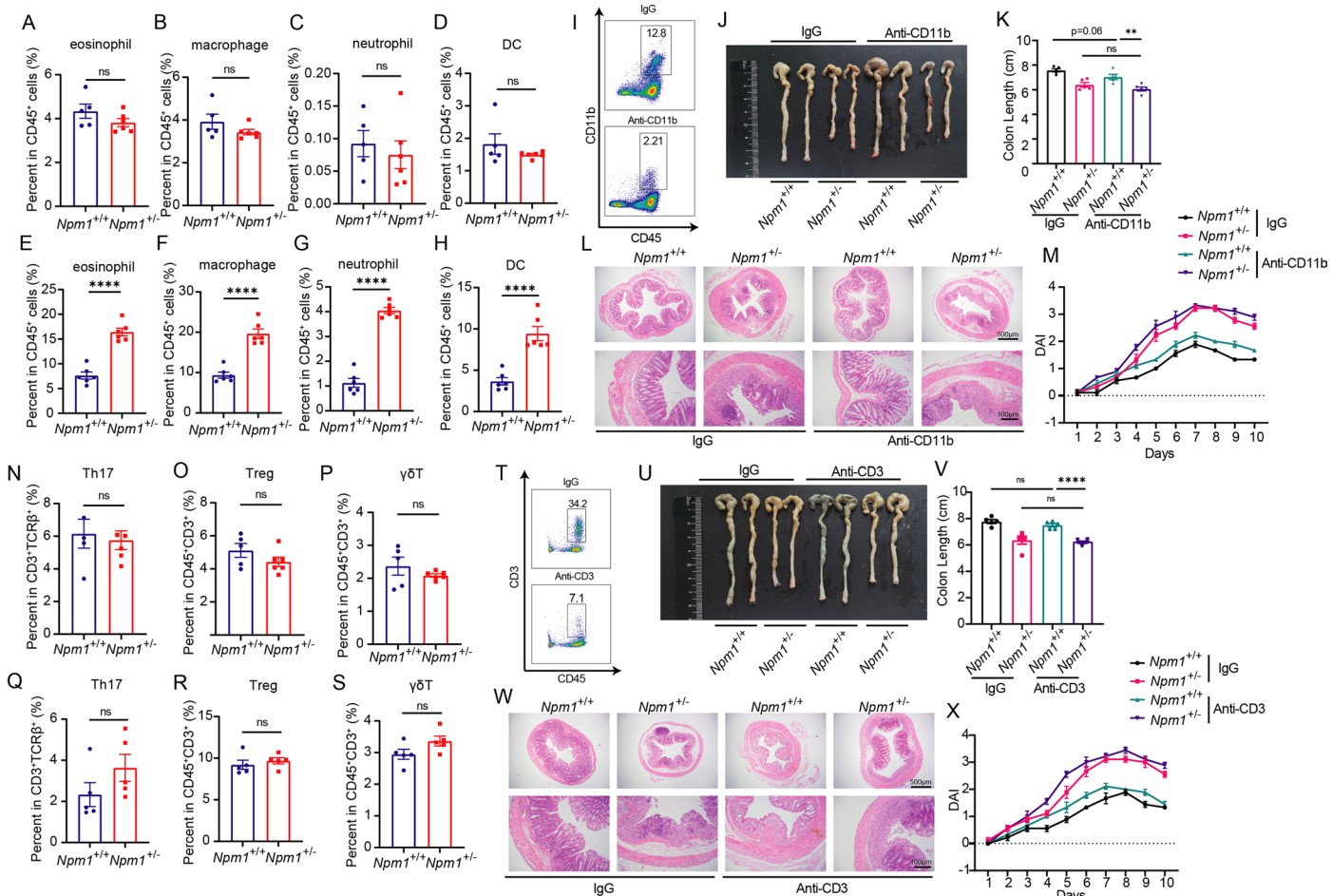

**Extended Data Fig. 3 | Regulation of colitis by NPM1 is independent of myeloid cells and T cells.** (a–h) Proportion of eosinophil, macrophage, neutrophil and dendritic cells (DCs) in lamina propria lymphocytes (LPLs) of *Npm1*[+/+] and *Npm1*[+/−] mice under steady-state conditions (a–d) (*Npm1*[+/+]: n = 5 individual mice; *Npm1*[+/−]: n = 6 individual mice) and during DSS-induced colitis (e–h) (n = 6 individual mice) state are shown. (i–m) Colitis in N*pm1*[+/+] and *Npm1*[+/−] mice was induced by DSS following administration with IgG or anti-CD11b blocking antibody. Deletion of CD11b+ cells by antibody (i). Representative images of colons (j), colon length (k) (n = 5 individual mice), colon histopathology on day 10 (l) and DAI (m) (n = 3 individual mice) are presented. Mice were injected with CD11b antibody (100 μg per mouse) every 2 days (from day −2 to day 6). Scale bars: 500 μm (up), 100 μm (down). (n–s) Proportion

of T_H17, T_reg and γδT in LPLs of *Npm1*[+/+] and *Npm1*[+/−] mice under steady-state conditions (n–p) (*Npm1*[+/+]: n = 5 individual mice; *Npm1*[+/−]: n = 6 individual mice) and during DSS-induced colitis (q–s) (n = 5 individual mice) state are shown. (t–x) DSS-induced colitis in N*pm1*[+/+] and *Npm1*[+/−] mice was established following administration with IgG or anti-CD3 blocking antibody. Deletion of CD3[+] cells by antibody (t). Representative images of colons (u), colon length (v) (n = 5 individual mice), colon histopathology (w) and DAI (x) (n = 3 individual mice) on day 10 are presented. Mice were injected with CD3 antibody (50 μg per mouse) once a day (from day −2 to day 6). Scale bars: 500 μm (up), 100 μm (down). Data in a–h, k, m, n–s, v and x are representative of two independent experiments, shown as the means ± s.e.m., and statistical significance was determined two-tailed unpaired Student's *t*-test (**p < 0.01 and ****p < 0.0001).

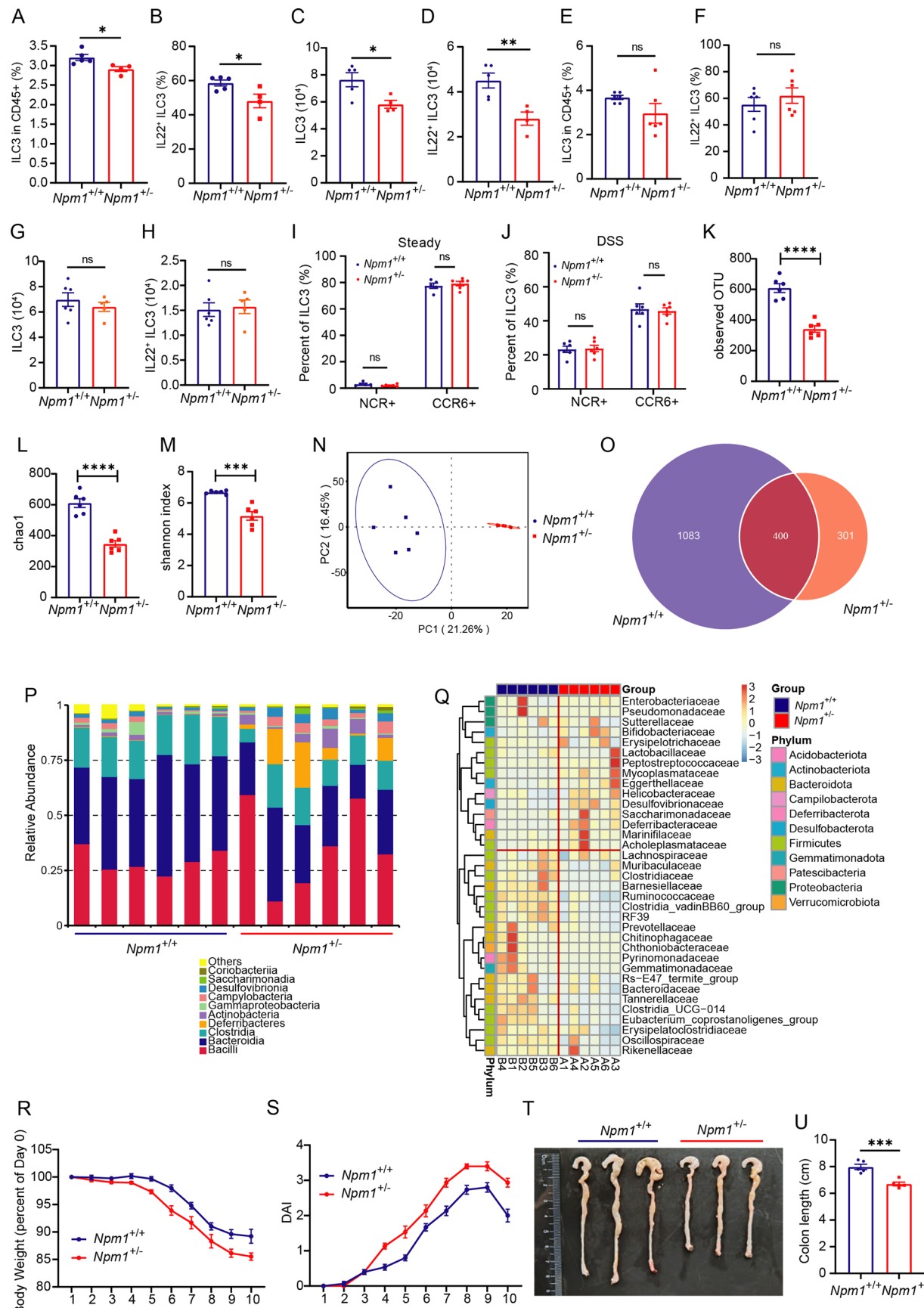

**Extended Data Fig. 4 | See next page for caption.**

**Extended Data Fig. 4 | Changes of ILC3 may contribute to dysbiosis of fecal microflora under pathological conditions.** (**a**–**d**) LPLs were isolated from *Npm1*[+/+] and *Npm1*[+/−] mice on day 5 after administration of TNBS. (**a**) Proportion of ILC3s in Lin[−] cells. (**b**) Proportion of IL-22[+] ILC3s in the total ILC3 population. Number of ILC3s (**c**) and IL-22[+] ILC3s (**d**) are shown (*Npm1*[+/+]: n = 5 individual mice; *Npm1*[+/−]: n = 4 individual mice). (**e**–**h**) ILC3s in LPLs of *Npm1*[+/+] and *Npm1*[+/−] mice under steady-state. (**e**) Proportion of ILC3s in Lin[−] cells. (**f**) Proportion of IL-22[+] ILC3s in the total ILC3 population (n = 6 individual mice). Number of ILC3s (**g**) and IL-22[+] ILC3s (**h**) are depicted (*Npm1*[+/+]: n = 6 individual mice; *Npm1*[+/−]: n = 5 individual mice). (**i**,**j**) Proportion of NCR[+] ILC3s and CCR6[+] ILC3s in total ILC3s from *Npm1*[+/+] and *Npm1*[+/−] mice under steady-state (**i**) (n = 5 individual mice) and during DSS-induced colitis (**j**) (n = 6 individual mice). (**k**–**q**) Feces from *Npm1*[+/+] mice and *Npm1*[+/−] mice under colitis were collected to analyze intestinal microbiota by 16S rRNA sequencing. (**k**) Observed operational taxonomic unit (OTU), (**l**) Chao1 index, (**m**) Shannon–Wiener diversity index (Shannon index) and (**n**) principal coordinates analysis (PCoA). (**o**) Venn diagram of two groups of fecal microbiota. (**p**) The relative abundance of microbiota at phylum level in the fecal samples. (**q**) Heatmap analysis of the relative abundance of microbiota at family level in the fecal samples. (n = 6 individual mice). (**r**–**u**) Co-housed *Npm1*[+/−] and control *Npm1*[+/+] mice were administered 2.5% DSS for 7 days, followed by 3 days of recovery (H$_2$O). Body weight (**r**), DAI (**s**) and colon length on day 10 (**t**,**u**) were analyzed (n = 5 individual mice). Scale bars: 500 μm (left), 100 μm (right). Data are representative of two independent experiments, shown as the means ± s.e.m., and statistical significance was determined by two-tailed unpaired Student's *t*-test (**a**–**h**,**k**–**m**,**u**) and two-way ANOVA (**i**,**j**) (*$p < 0.05$, **$p < 0.01$, ***$p < 0.001$, ****$p < 0.0001$).

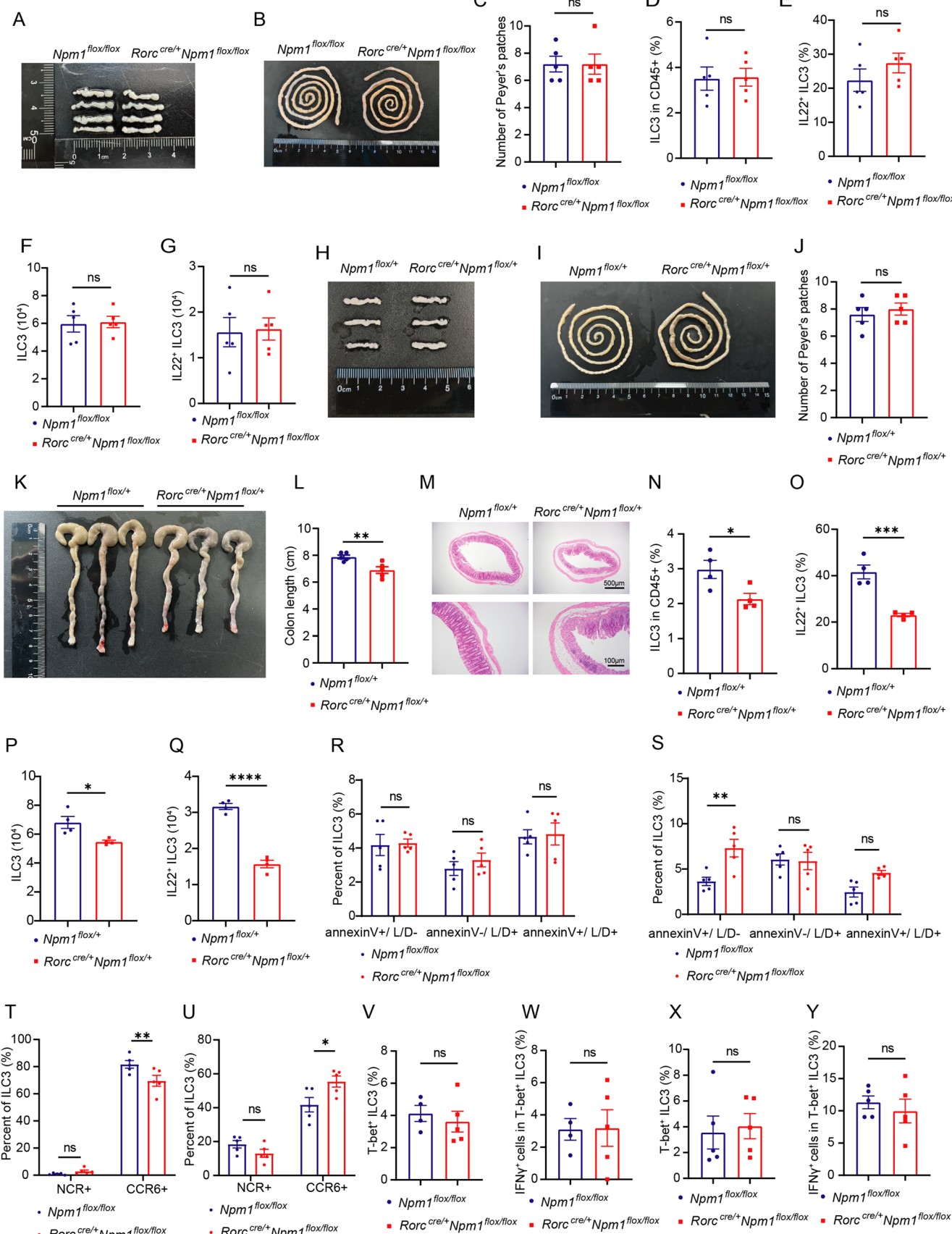

**Extended Data Fig. 5 | See next page for caption.**

**Extended Data Fig. 5 | NPM1 is required for maintaining the frequency and function of colonic ILC3s.** (**a**) Images of mesenteric lymph nodes from *Npm1^{flox/flox}* and *Rorc^{cre/+}Npm1^{flox/flox}* mice (n = 4 individual mice). (**b,c**) Representative images of Peyer's patches from *Npm1^{flox/flox}* and *Rorc^{cre/+}Npm1^{flox/flox}* mice in steady state. The number of Peyer's patches (**c**) was analyzed (n = 5 individual mice). (**d–g**) ILC3s in LPLs of *Npm1^{flox/flox}* and *Rorc^{cre/+}Npm1^{flox/flox}* mice under steady-state. (**d**) Proportion of ILC3s in Lin⁻ cells. (**e**) Proportion of IL-22⁺ ILC3s in the total ILC3 population. Number of ILC3s (**f**) and IL-22⁺ ILC3s (**g**) are depicted (n = 5 individual mice). (**h**) Images of mesenteric lymph nodes from *Npm1^{flox/flox}* and *Rorc^{cre/+}Npm1^{flox/flox}* mice (n = 3 individual mice). (**i,j**) Representative images of Peyer's patches from *Npm1^{flox/+}* and *Rorc^{cre/+}Npm1^{flox/+}* mice in steady state (**i**). The number of Peyer's patches (**j**) was analyzed (n = 5 individual mice). (**k–m**) *Npm1^{flox/+}* and *Rorc^{cre/+}Npm1^{flox/+}* mice were administered 2.5% DSS for 7 days followed by 3 days of recovery. Representative images of colons (**k**), colon length (**l**) and colon histopathology on day 10 (**m**) are presented. (n = 4 individual mice). Scale bars: 500 μm (up), 100 μm (down). (**n–q**) ILC3s in LPLs of *Npm1^{flox/+}* and *Rorc^{cre/+}Npm1^{flox/+}* mice with colitis. (**n**) Proportion of ILC3s in Lin⁻ cells. (**o**) Proportion of IL-22⁺ ILC3s in the total ILC3 population. Number of ILC3s (**p**) and IL-22⁺ ILC3s (**q**) are provided (n = 4 individual mice). (**r,s**) Apoptotic percentage of ILC3s in LPLs from *Npm1^{flox/flox}* and *Rorc^{cre/+}Npm1^{flox/flox}* mice was detected by Annexin V staining under steady-state (**r**) and during DSS-induced colitis (**s**) (n = 5 individual mice). (**t,u**) Proportion of NCR⁺ ILC3s and CCR6⁺ ILC3s in total ILC3s from colon of *Npm1^{flox/flox}* and *Rorc^{cre/+}Npm1^{flox/flox}* mice under steady-state (**t**) and during DSS-induced colitis (**u**) (n = 5 individual mice). (**v–y**) Proportion of T-bet⁺ ILC3s in LPLs and IFNγ⁺ cells in T-bet⁺ ILC3s from *Npm1^{flox/flox}* and *Rorc^{cre/+}Npm1^{flox/flox}* mice under steady-state (**v,w**) (n = 4 individual mice) and during DSS-induced colitis (**x,y**) (n = 5 individual mice). Data are representative of two independent experiments, shown as the means ± s.e.m., and statistical significance was determined by two-tailed unpaired Student's *t*-test (**c–g**, **j**, **l**, **n–q** and **v–y**) and two-way ANOVA (**r–u**) (*$p < 0.05$, **$p < 0.01$, ***$p < 0.001$, ****$p < 0.0001$).

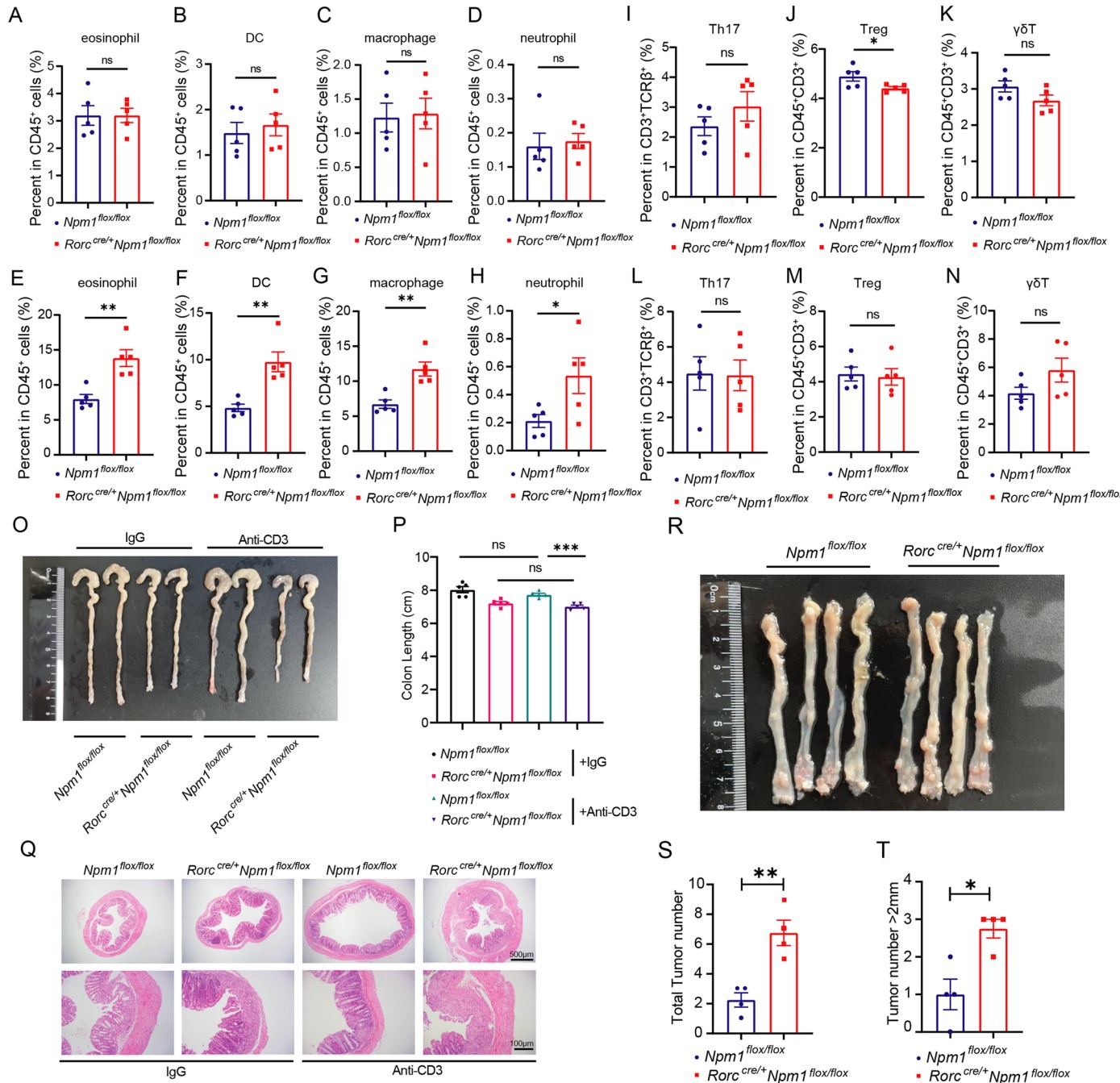

**Extended Data Fig. 6 | Exacerbate enteritis in *Rorc^cre/+^Npm1^flox/flox^* mice under DSS is independent of T cells. (a–n)** Proportions of eosinophil, dendritic cells (DCs), macrophages, neutrophils, $T_H17$, $T_{reg}$ and γδT in LPLs from *Npm1^flox/flox^* and *Rorc^cre/+^Npm1^flox/flox^* mice under steady-state (**a–d,i–k**) and during DSS-induced colitis (**e–h,l–n**) (n = 5 individual mice). (**o–q**) Colitis in *Npm1^flox/flox^* and *Rorc^cre/+^Npm1^flox/flox^* mice was induced by DSS following administration with IgG or anti-CD3 blocking antibody. Representative images of colons (**o**), colon length (**p**) and colon histopathology (**q**) on day 10 are presented (n = 5 individual mice).

Mice were injected with CD3 antibody (50 μg per mouse) once a day (from day −2 to day 6). Scale bars: 500 μm (up), 100 μm (down). (**r**) Representative images of colons with tumors from *Npm1^flox/flox^* and *Rorc^cre/+^Npm1^flox/flox^* on day 65 of the AOM/DSS CAC model. (**s,t**) Total number of tumors (**s**) and number of tumors larger than 2 mm (**t**) in *Npm1^flox/flox^* and *Rorc^cre/+^Npm1^flox/flox^* mice (n = 4 individual mice). Data in **a–n**, **p**, **s** and **t** are representative of two independent experiments, shown as the means ± s.e.m., and statistical significance was determined two-tailed unpaired Student's *t*-test (*$p < 0.05$, **$p < 0.01$ and ***$p < 0.001$).

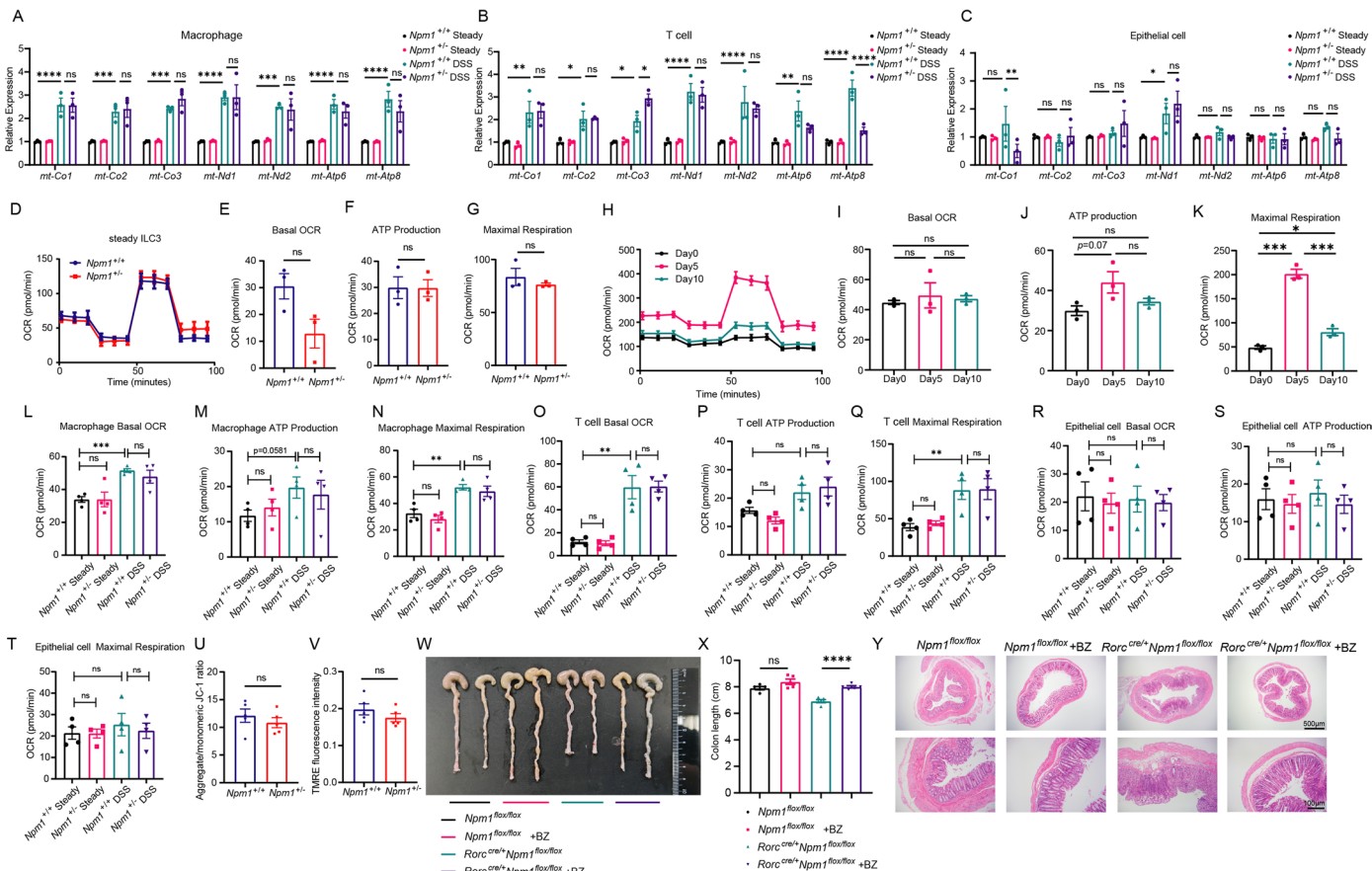

**Extended Data Fig. 7 | NPM1 is essential for mitochondrial function in ILC3s.**
(**a**–**c**) RT-PCR analysis of mRNA abundance of the indicated genes in macrophage
(**a**), T cells (**b**) and epithelial (**c**), sorted from the LPLs of *Npm1*[+/+] and *Npm1*[+/−] mice
in steady state or at day 5 of administration of 2.5% DSS (n = 3 individual mice).
(**d**–**g**) Cell mito stress test was performed with sorted colonic ILC3s from *Npm1*[+/+]
and *Npm1*[+/−] mice under steady-state. Representative oxygen consumption rate
profile (OCR) (**d**), basal OCR (**e**), ATP production (**f**) and maximal respiration (**g**)
are shown (n = 3 individual mice). (**h**–**k**) Cell mito stress test was conducted with
sorted colonic ILC3s from wild-type (WT) mice on days 0, 5 and 10 of DSS-induced
colitis. Representative OCR (**h**), basal OCR (**i**), ATP production (**j**) and maximal
respiration (**k**) are presented (n = 3 individual mice). (**l**–**t**) Cell mito stress test
was performed with sorted colonic macrophages (**l**–**n**), T cells (**o**–**q**), epithelial

cells (**r**–**t**) from LPLs of *Npm1*[+/+] and *Npm1*[+/−] mice at day 5 of administration of
2.5% DSS (n = 4 individual mice). (**u**,**v**) Mitochondrial membrane potential was
assessed with the indicator JC-1 (**u**) and TMRE (**v**) in isolated colonic ILC3s from
*Npm1*[+/+] and *Npm1*[+/−] mice in steady state (n = 5 individual mice). (**w**–**y**) *Npm1*[flox/flox]
and *Rorc*[cre/+]*Npm1*[flox/flox] mice were treated with bezafibrate (i.g., 10 mg/kg) and
administered 2.5% DSS for 7 days followed by 3 days of recovery. Representative
colon images (**w**), colon length (**x**) and colon histopathology (**y**) are shown
(n = 5 individual mice). Scale bars: 500 μm (up), 100 μm (down). Data are
representative of two independent experiments, shown as the means ± s.e.m.,
and statistical significance was determined by two-way ANOVA (**a**–**c**) and two-
tailed unpaired Student's *t*-test (**e**–**g**,**i**–**v**,**x**) (**p* < 0.05, ***p* < 0.01, ****p* < 0.001,
*****p* < 0.0001).

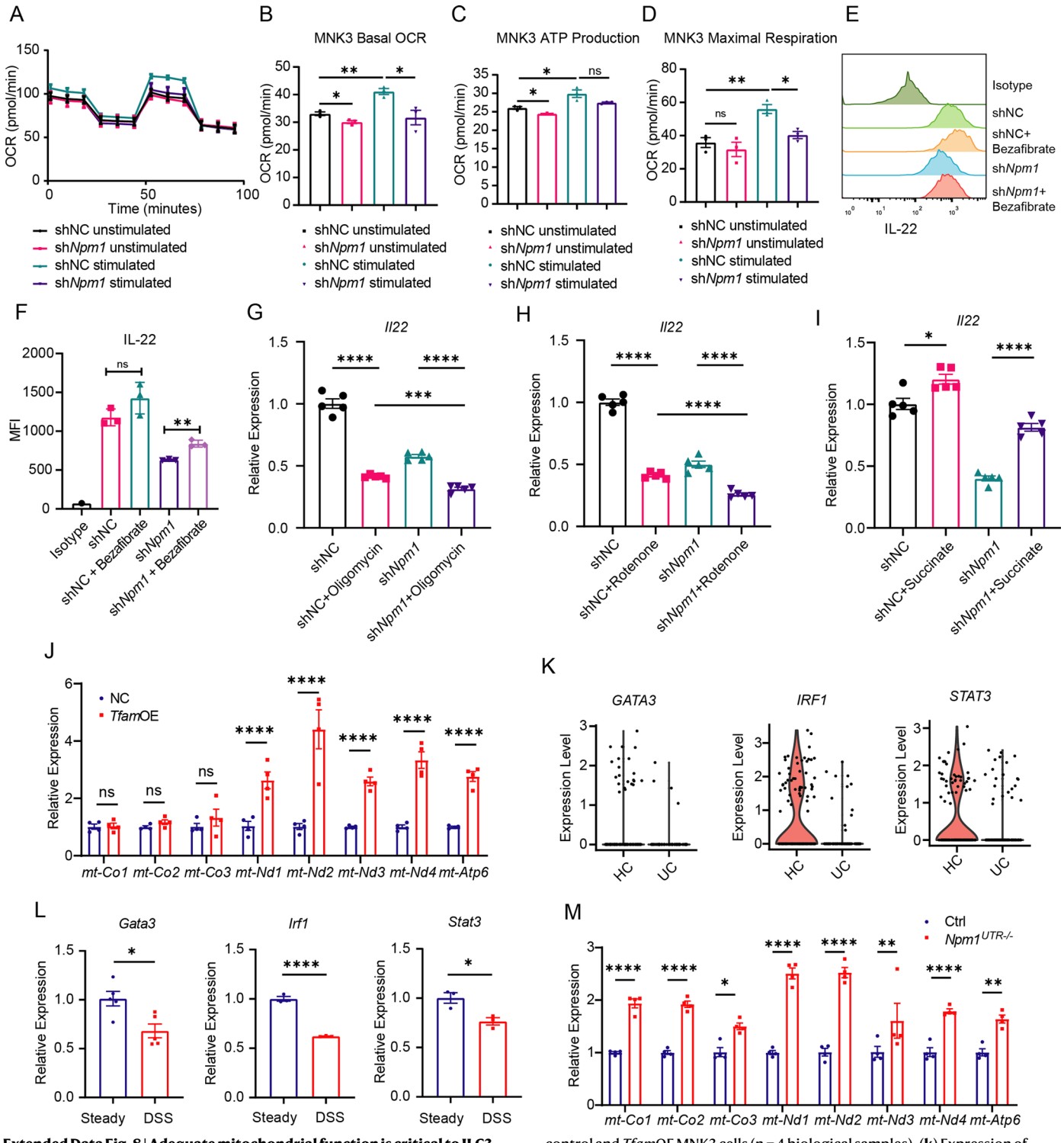

**Extended Data Fig. 8 | Adequate mitochondrial function is critical to ILC3 activation.** (**a**–**d**) Cell mito stress test was performed with unstimulated and stimulated MNK3 cell lines (shNC and sh*Npm1*). Representative oxygen consumption rate profile (OCR) (**a**), basal OCR (**b**), ATP production (**c**) and maximal respiration (**d**) are shown (n = 3 biological samples). (**e**,**f**) MNK3 cells with or without *Npm1*-knockdown were treated with bezafibrate, and the MFI of IL-22 in MNK3 was analyzed by FC (n = 3 biological samples). (**g**–**i**) MNK3 cells with or without *Npm1*-knockdown were treated with oligomycin (**g**), rotenone (**h**) and succinate (**i**), and the expression of *Il22* was analyzed by RT-PCR (n = 5 biological samples). (**j**) RT-PCR analysis of mRNA abundance of the indicated genes in

control and *Tfam*OE MNK3 cells (n = 4 biological samples). (**k**) Expression of indicated genes in ILC3s of UC and HC in GSE182270. (**l**) RT-PCR analysis of mRNA abundance of *Gata3* (n = 5 indicated mice), *Irf1* (n = 3 indicated mice), *Stat3* (n = 3 indicated mice) in ILC3s of mice in steady state or in DSS-induced colitis. (**m**) RT-PCR analysis of mRNA abundance of the indicated genes in wild-type and *Npm1*[UTR-/-] ILC3s at day 5 of administration of 2.5% DSS (n = 4 indicated mice). Data are representative of two independent experiments, shown as the means ± s.e.m., and statistical significance was determined by two-tailed unpaired Student's *t*-test (**b**–**d**, **f**–**i** and **l**) and two-way ANOVA (**j**,**m**) (*$p < 0.05$, **$p < 0.01$, ***$p < 0.001$, ****$p < 0.0001$).

# Reporting Summary

## Statistics

For all statistical analyses, confirm that the following items are present in the figure legend, table legend, main text, or Methods section.

| n/a | Confirmed | |
|---|---|---|
| ☐ | ☒ | The exact sample size (*n*) for each experimental group/condition, given as a discrete number and unit of measurement |
| ☐ | ☒ | A statement on whether measurements were taken from distinct samples or whether the same sample was measured repeatedly |
| ☐ | ☒ | The statistical test(s) used AND whether they are one- or two-sided<br>*Only common tests should be described solely by name; describe more complex techniques in the Methods section.* |
| ☐ | ☒ | A description of all covariates tested |
| ☐ | ☒ | A description of any assumptions or corrections, such as tests of normality and adjustment for multiple comparisons |
| ☐ | ☒ | A full description of the statistical parameters including central tendency (e.g. means) or other basic estimates (e.g. regression coefficient) AND variation (e.g. standard deviation) or associated estimates of uncertainty (e.g. confidence intervals) |
| ☐ | ☒ | For null hypothesis testing, the test statistic (e.g. *F*, *t*, *r*) with confidence intervals, effect sizes, degrees of freedom and *P* value noted<br>*Give P values as exact values whenever suitable.* |
| ☒ | ☐ | For Bayesian analysis, information on the choice of priors and Markov chain Monte Carlo settings |
| ☒ | ☐ | For hierarchical and complex designs, identification of the appropriate level for tests and full reporting of outcomes |
| ☐ | ☒ | Estimates of effect sizes (e.g. Cohen's *d*, Pearson's *r*), indicating how they were calculated |

*Our web collection on statistics for biologists contains articles on many of the points above.*

## Software and code

Policy information about availability of computer code

| | |
|---|---|
| Data collection | Flow cytometric analysis was performed using the BD FACSCelesta Flow Cytometer and Gallios Flow Cytometer.<br>Immunofluorescence: Leica SP5 confocal microscope. |
| Data analysis | Flow Cytometry data were analyzed by FlowJo V10, Immunofluorescence images were analyzed by Image J 64-bit Java 8. Statistical analyses were performed using GraphPad Prism V8. Data were presented as mean±SEM. Statistical significance was assessed by student's t-test (unpaired) or two-way ANOVA analyses. All statistical tests were two-tailed and a p-value of <0.05 was considered statistically significant. |

For manuscripts utilizing custom algorithms or software that are central to the research but not yet described in published literature, software must be made available to editors and reviewers. We strongly encourage code deposition in a community repository (e.g. GitHub). See the Nature Portfolio guidelines for submitting code & software for further information.

## Data

Policy information about <u>availability of data</u>

All manuscripts must include a <u>data availability statement</u>. This statement should provide the following information, where applicable:

- Accession codes, unique identifiers, or web links for publicly available datasets
- A description of any restrictions on data availability
- For clinical datasets or third party data, please ensure that the statement adheres to our <u>policy</u>

Smart-seq analysis of primary colonic ILC3 in Npm1+/+ and Npm1+/- mice can be assessed with GEO number "GSE271455". scRNA-seq data of UC patients and healthy controls was downloaded from GSE182270 in the GEO repository. https://www.ncbi.nlm.nih.gov/geo/query/acc.cgi?acc=GSE182270.

## Research involving human participants, their data, or biological material

Policy information about studies with <u>human participants or human data</u>. See also policy information about <u>sex, gender (identity/presentation), and sexual orientation</u> and <u>race, ethnicity and racism</u>.

| | |
|---|---|
| Reporting on sex and gender | Male and female rectal tissue samples were used in our study, and this information is provided in Supplementary Table 1. |
| Reporting on race, ethnicity, or other socially relevant groupings | We worked to ensure sex and gender balance in the recruitment of participants.<br>We worked to ensure racial and ethnic or other types of diversity in the recruitment of participants. |
| Population characteristics | Information about the human rectal tissue samples is provided in Supplementary Table 1. |
| Recruitment | Human rectal tissue samples were obtained from approved tissue banks. No patients were recruited. |
| Ethics oversight | Pathological sections were obtained from ulcerative colitis patients, Crohn's disease patients and healthy individuals after approval had been obtained from Ethics Committee of Shandong University School of Basic Medicine (ECSBMSSDU2020-1-035). |

Note that full information on the approval of the study protocol must also be provided in the manuscript.

# Field-specific reporting

Please select the one below that is the best fit for your research. If you are not sure, read the appropriate sections before making your selection.

☒ Life sciences　　☐ Behavioural & social sciences　　☐ Ecological, evolutionary & environmental sciences

For a reference copy of the document with all sections, see nature.com/documents/nr-reporting-summary-flat.pdf

# Life sciences study design

All studies must disclose on these points even when the disclosure is negative.

| | |
|---|---|
| Sample size | No statistical method was used to predetermine sample size. All studies using at least 3-5 animals per group, all experiments were performed at least twice to ensure reproducibility, which is commonly accepted in the field of immunology. |
| Data exclusions | No sample were excluded from analysis. |
| Replication | All attempts at replication were successful. All experiments were independently performed at least twice to ensure reproducibility. |
| Randomization | For DSS/TNBS/AOM-DSS animal studies to assess the changes of Npm1-deficiency, no method of randomization was used. Mice were grouped according to genotype and all experiments were performed with sex-matched littermates. For bezafibrate experiment, mice were first grouped by genotype and than randomly assigned to two groups (bezafibrate treated group and control group). For CD11b/CD3 antibody treatment experiments, mice were first grouped by genotype and than randomly assigned to two groups (CD11b/CD3 antibody treated group and IgG antibody treated group). |
| Blinding | Animal studies were not blinded (mice were named with mouse ID and genotyped within 6 weeks of birth). Group allocation was not applicable because mice were grouped based on and compared across different genotypes. Histological analysis were analyzed by two independent investigators, who had limited knowledge of the group of mice and patients. DAI were analyzed by two independent investigators, who had limited knowledge of the group of mice and patients. Data of FACS, qPCR, ELISA, et.al were collected by an investigator with only the knowledge of mouse ID (without grouping information). |

# Reporting for specific materials, systems and methods

We require information from authors about some types of materials, experimental systems and methods used in many studies. Here, indicate whether each material, system or method listed is relevant to your study. If you are not sure if a list item applies to your research, read the appropriate section before selecting a response.

## Materials & experimental systems

| n/a | Involved in the study |
|---|---|
| ☐ | ☒ Antibodies |
| ☐ | ☒ Eukaryotic cell lines |
| ☒ | ☐ Palaeontology and archaeology |
| ☐ | ☒ Animals and other organisms |
| ☒ | ☐ Clinical data |
| ☒ | ☐ Dual use research of concern |
| ☒ | ☐ Plants |

## Methods

| n/a | Involved in the study |
|---|---|
| ☒ | ☐ ChIP-seq |
| ☐ | ☒ Flow cytometry |
| ☒ | ☐ MRI-based neuroimaging |

# Antibodies

| | |
|---|---|
| Antibodies used | Anti-mouse CD45 eFlour 506 (clone 30-F11) eBioscience Cat#69-0451-82; RRID:AB_2637147  dilution 1:100<br>Anti-mouse RORγt PE (clone B2D) eBioscience Cat#12-6981-80; RRID:AB_10807092  dilution 1:50<br>Anti-mouse Ly-6G PE (clone 1A8-Ly6g) eBioscience Cat#12-9668-80; RRID:AB_2572720  dilution 1:100<br>Anti-mouse CD127 Super Bright 645 (clone A7R34) eBioscience Cat#64-1271-80; RRID:AB_2744868  dilution 1:100<br>Anti-mouse F4/80 FITC (clone BM8) eBioscience Cat#11-4801-82; RRID:AB_2637191  dilution 1:100<br>Anti-mouse CD3 Alexa-488 (clone 17A2) eBioscience Cat#53-0032-82; RRID:AB_2848414  dilution 1:100<br>Anti-mouse CD34 FITC (clone RAM34) eBioscience Cat#11-0341-81; RRID:AB_465021  dilution 1:100<br>Anti-mouse CD117 APC (clone ACK2) eBioscience Cat#17-1172-82; RRID:AB_469433  dilution 1:100<br>Anti-mouse CD19 eFlour 450 (clone 1D3) eBioscience Cat#48-0193-82; RRID:AB_2734905  dilution 1:100<br>Anti-mouse IL-22 PE (clone 1H8PWSR) eBioscience Cat#12-7221-80; RRID:AB_10597428  dilution 1:50<br>Anti-mouse CD16/32 (clone 93) eBioscience Cat#14-0161-82; RRID:AB_467133  dilution 1:100<br>Anti-mouse CD4 APC (clone V4) Biolegend Cat#100411; RRID; AB_312696  dilution 1:100<br>Anti-mouse IL-17A BV421 (clone TC11-18H10) BD Cat#566286; RRID; AB_2687547  dilution 1:50<br>Anti-mouse Lineage Percp-cy5.5 Cocktail BD Cat#51-9006964; RRID:AB_10612020  dilution 1:50<br>Anti-mouse T-bet PE (clone 4B10) eBioscience Cat#25-5825-82; RRID:AB_10565980  dilution 1:100<br>Anti-mouse IFNgamma-APC (clone XMG1.2)  eBioscience Cat#17-7311-82; RRID:AB_469504  dilution 1:50<br>Anti-mouse NKp46-PerCPcy5.5 (clone 29A1.4) eBioscience Cat#46-3351-82; RRID:AB_1834441  dilution 1:50<br>Anti-mouse FOXP3-eFlour 450 (clone FJK-16S) eBioscience Cat#48-5773-82; RRID:AB_1518812  dilution 1:100<br>Anti-mouse CCR6-BV421 (clone 29-2L17) BioLegend Cat#129818; RRID:AB_11219003  dilution 1:50<br>Anti-mouse TCR gamma/delta-APC (clone GL3) BioLegend Cat#118116; RRID:AB_1731813  dilution 1:50<br>Anti-mouse CD127-FITC (clone A7R34) BioLegend Cat#135008; RRID:AB_AB_1937232  dilution 1:100<br>Rabbit SDHB Antibody Proteintech  Cat# 10620-1-AP; RRID:AB_2285522  dilution 1:2000<br>Rabbit NDUFB8 Antibody Proteintech Cat# 14794-1-AP; RRID:AB_2150970 dilution 1:2000<br>Rabbit MT-ATP6 Antibody ABclonal Cat# A17960; RRID; AB_2861763 dilution 1:1000<br>Rabbit MT-CO1 Antibody ABclonal Cat# A17889; RRID; AB_2861744 dilution 1:1000<br>Rabbit UQCRC2 Antibody Proteintech Cat# 14742-1-AP; RRID:AB_2241442 dilution 1:1000<br>Rabbit NPM1 Antibody ABclonal  Cat# A17983; RRID; AB_2861784 dilution 1:1000<br>Rabbit TOM20 Antibody Proteintech Cat# 66777-1-Ig; RRID:AB_2882123 dilution 1:200<br>Rabbit NF-kappaB p65 Antibody Cell Signaling Technology Cat# 8242; RRID; AB_10859369 dilution 1:1000<br>Mouse NF-kB p65/RelA Antibody ABclonal Cat# A10609 dilution 1:1000<br>Mouse NPM1 Antibody Proteintech Cat# 60096-1-Ig; RRID:AB_2155162 dilution 1:200<br>ROR gamma (t) Monoclonal Antibody (clone AFKJS-9), eBioscience Cat#14-6988-82; RRID:AB_1834475  dilution 1:200<br>Donkey anti-Mouse IgG (H+L) Highly Cross-Adsorbed Secondary Antibody, Alexa Fluor™ 488 Invitrogen Cat#A21202; RRID:AB_141607 dilution 1:2000<br>Donkey anti-Rat IgG (H+L) Highly Cross-Adsorbed Secondary Antibody, Alexa Fluor™ 555 Invitrogen Cat#A78945; RRID:AB_2910652 dilution 1:2000<br>Donkey anti-Rabbit IgG (H+L) Highly Cross-Adsorbed Secondary Antibody, Alexa Fluor™ 647 Invitrogen Cat#A31573; RRID: AB_2536183 dilution 1:2000<br>IRDye® 680RD Donkey anti-Rabbit IgG Secondary Antibody, LICORbio, Cat#926-68073; RRID: AB_2716687 dilution: 1:10000<br>IRDye® 800CW Donkey anti-Mouse IgG Secondary Antibody, LICORbio, Cat#926-32212; RRID: AB_2716622 dilution: 1:10000 |
| Validation | All antibodies listed above were commercially available and validated by the manufacturer. Validation data are available on the manufacturer's website. All antibodies described here have been further optimised for an appropriate concentration by testing several dilutions. |

# Eukaryotic cell lines

Policy information about cell lines and Sex and Gender in Research

| | |
|---|---|
| Cell line source(s) | Mouse MNK3 cell line; BLUEFBIO Cat# BFN60807579<br>Human HEK293T cell line; Lab preserve |
| Authentication | Identity of the cell lines were frequently checked by their morphological features. |

| | |
|---|---|
| Mycoplasma contamination | All cell lines were tested to be mycoplasma-negative by the standard PCR method. |
| Commonly misidentified lines<br>(See ICLAC register) | No commonly misidentified cell lines are used in this study. |

# Animals and other research organisms

Policy information about studies involving animals; ARRIVE guidelines recommended for reporting animal research, and Sex and Gender in Research

| | |
|---|---|
| Laboratory animals | Npm1+/-, Npm1UTR+/-, Npm1flox/+ mice were generated in this study by CRO company Shanghai Model Organisms Center, Inc. Villin cre/+ mice and Rorc cre/+ mice were gifted by Li Lab from Shandong University.<br>Apc min/+ mice were purchased from The Jackson Laboratory, Cat# 002020<br>All experiments were performed using C57BL/6J mice, which also served as controls for Npm1+/-, Npm1UTR-/- and Apcmin/+ mice.<br>Npm1flox/flox mice were served as controls for Rorccre/+Npm1flox/flox and Villincre/+Npm1flox/flox mice.<br>6-8 week-old male mice were used for experiments.<br>Mice were housed in individually ventilated cages under 12h light-12h dark cycle with normal food and water. |
| Wild animals | No wild animals were included in this study. |
| Reporting on sex | All the mice used in the experiments were male. |
| Field-collected samples | No field-collected samples included in this study |
| Ethics oversight | All animal experiments were approved and are in accordance with the Institutional Animal Care and Use Committee guidelines at Suzhou Institute of Biomedical Engineering and Technology (2021-C058) and Shandong University (ECSBMSSDU2020-2-057). |

Note that full information on the approval of the study protocol must also be provided in the manuscript.

# Flow Cytometry

## Plots

Confirm that:

☒ The axis labels state the marker and fluorochrome used (e.g. CD4-FITC).

☒ The axis scales are clearly visible. Include numbers along axes only for bottom left plot of group (a 'group' is an analysis of identical markers).

☒ All plots are contour plots with outliers or pseudocolor plots.

☒ A numerical value for number of cells or percentage (with statistics) is provided.

## Methodology

| | |
|---|---|
| Sample preparation | To isolate leukocytes from the lamina propria, we incubated intestinal segments of approximately 0.5 cm at 37°C for 1.5 hours in complete RPMI medium, supplemented with DNase I (150 μg/ml, Sigma) and collagenase VIII (300 U/ml, Sigma). The digested fragments were triturated and filtered through a 100 μm cell strainer. The cells were collected from the interface of the 80% and 40% Percoll gradients after centrifugation at 660 xg for 15 minutes at room temperature. Prior to surface staining, Fc receptors were blocked using CD16/32 antibody (eBioscience). Leukocytes isolated from the intestinal lamina propria were then stained with antibodies against the distinct markers. For cytokine staining, cells were stimulated with PMA (50 ng/ml) and ionomycin (500 ng/ml) for 2 hours, along with the addition of brefeldin A (2 μg/ml). Live and dead cells were distinguished using the Live and Dead Violet Viability Kit (BioLegend). |
| Instrument | Flow cytometric analysis was performed using the BD FACSCelesta Flow Cytometer and Gallios Flow Cytometer. Samples for RNAseq and in vivo experiment were sorted on Moflo Astrios EQ and BD FACSMelody Flow Cytometer. |
| Software | FlowJo software v10 |
| Cell population abundance | Purity above 90% |
| Gating strategy | Live single cells were identified based on FSC/SSC as well as live/dead cell staining. The sorted ILC3s were defined as Live+Lin-CD45lowCD90.2high cells. For analysis, macrophages were gated as CD45+CD11b+F4/80+ cells; neutrophils were gated as CD45+CD11b+Ly6G+ cells; eosinophils were gated as CD45+CD11b+SIGLECF+; T cells were gated as CD45+CD3+; B cells were gated as CD45+CD19+; ILC3s were gated as CD45+Lin-RORgt+ cells. |

☒ Tick this box to confirm that a figure exemplifying the gating strategy is provided in the Supplementary Information.

