## [Peer Review File · Nature Immunology]

Nucleophosmin 1 promotes mucosal immunity by supporting mitochondrial oxidative phosphorylation and ILC3 activity

Corresponding Author: Dr Minxuan Sun

Version 0:

Decision Letter:

14th Aug 2023

Dear Dr Sun,

As you know, your Article, "Nucleophosmin 1 Promotes Mucosal Immunity by Supporting Mitochondrial Oxidative Phosphorylation and ILC3 Activity" has now been seen by 3 referees. You will see from their comments copied below that while they find your work of considerable potential interest, they have raised quite substantial concerns that must be addressed. In light of these comments, we cannot accept the manuscript for publication, but would be interested in considering a revised version that addresses these serious concerns.

PLEASE NOTE that we have also looked over your Author Response to the reviewer comments - thanks for sending me that - and we are happy to say that the plan sounds OK to us, but of course if there is further reviewer negativity from reviewer 1 and 2 then the outcome would not be favourable.

We hope you will find the referees' comments useful as you decide how to proceed. If you wish to submit a substantially revised manuscript, please bear in mind that we will be reluctant to approach the referees again in the absence of major revisions.

If you choose to revise your manuscript taking into account all reviewer and editor comments, please highlight all changes in the manuscript text file in Microsoft Word format.

* If you have not done so already please begin to revise your manuscript so that it conforms to our Article format instructions at <http://www.nature.com/ni/authors/index.html>. Refer also to any guidelines provided in this letter.

The Reporting Summary can be found here:

Link Redacted

If you wish to submit a suitably revised manuscript we would hope to receive it within 6 months. If you cannot send it within this time, please let us know. We will be happy to consider your revision so long as nothing similar has been accepted for publication at Nature Immunology or published elsewhere.

Nature Immunology is committed to improving transparency in authorship. As part of our efforts in this direction, we are now requesting that all authors identified as 'corresponding author' on published papers create and link their Open Researcher and Contributor Identifier (ORCID) with their account on the Manuscript Tracking System (MTS), prior to acceptance. ORCID helps the scientific community achieve unambiguous attribution of all scholarly contributions. You can create and link your ORCID from the home page of the MTS by clicking on 'Modify my Springer Nature account'. For more information please visit please visit www.springernature.com/orcid.

Thank you for the opportunity to review your work.

Sincerely,

Nick Bernard, PhD
Senior Editor
Nature Immunology

Reviewers' Comments:

Reviewer #1:

Remarks to the Author:

The authors investigate the role of reduced expression of Nucleophosmin (NPM1) for inflammatory bowel disease and colitis-associated cancer. NPM1 mutations are associated with myelodysplastic syndrome (MDS) and MDS patients often also have IBD. The molecular link between these diseases remains unknown. The authors show that mice with reduced expression of the Npm1 gene (Npm1^{+/-} mice) are more susceptible to DSS-induced colitis. In Npm1^{+/-} mice ILC3 were found to produce less of the epithelia-protective cytokine IL-22. On a mechanistic level, ILC3 from Npm1^{+/-} mice showed defects in mitochondrial biogenesis and reduced OXPHOS. NPM1 could physically interact with p65 (RelA) transcriptionally activating the expression of mitochondrial transcription factor A (Tfam), required for mitochondrial DNA transcription required for mitochondrial fitness and IL-22 production in colonic ILC3. Key data are also observed in human cohorts.

Overall, this is an interesting manuscript linking NPM1 to intestinal protection via IL-22 and ILC3. However, there are currently some limitations and conceptual questions that should be addressed in a revised manuscript. The following issues are raised.

1. NPM1 is expressed quite broadly both in hematopoietic and non-hematopoietic cells. A wider characterization of Npm1^{+/-} mice is required at steady-state and during DSS colitis. In particular, bone marrow and intestinal immune cells as well as epithelial cells should be characterized in some detail (representation, differentiation, function).
2. The authors show that NPM1 controls mitochondrial biogenesis and function in ILC3. Considering that this is quite a basic metabolic program, it is surprising that the authors only detect changes in ILC3. Why is NPM1 so central to ILC3 function? Why is reduced expression of NPM1 in Npm1^{+/-} mice compensated in other cell types that also express NPM1 and also require OXPHOS for their function. While I can see that this is a broad question and a full response would be beyond the scope of the current manuscript, the authors should exclude a role of epithelial NPM1 or of NPM1 in myeloid cells or T cells in the phenotype they observe.
3. The data support a concept in which ILC3 do not require high level expression of NPM1 at steady-state but are dependent on full NPM1 expression during colitis. How can this be explained? Do ILC3 undergo significant metabolic changes during the various phases of DSS colitis (acute tissue damage vs. repair phase)? This needs to be addressed by additional experiments.

In this context, the data with MNK cells and IL-22 is surprising. The authors should add conditions without stimulation with IL-1b and IL-23. Would IL-1b and/or IL-23 change metabolic function of MNK cells?

4. Subsets of ILC3 are needed for prenatal organogenesis of secondary lymphoid tissue (Peyer's patches, lymph nodes) and postnatal formation of solitary intestinal lymphoid tissue. Are these normally represented in Npm1^{+/-} mice?

5. Data during the last years have documented an important effect of the intestinal microbiota for the severity of DSS colitis. Are there differences in the representation of microbiota between Npm1^{+/+} and Npm1^{+/-} mice at steady-state or during colitis? Would crossfostered and/or co-housed littermates show the same differences? The breeding schemes and housing conditions should be explained in the Methods section.

6. The authors state (line 138/39) that "accumulation of populations of macrophages, neutrophils, and eosinophils was similar in Npm1^{+/-} mice compared to WT mice (Extended Data Fig. 5A,5B)". However, the data shows that the representation of macrophages, neutrophils, and eosinophils is significantly increased in Npm1^{+/-} mice. The statement needs to be corrected and a comparable analysis of macrophages, neutrophils, and eosinophils should be performed in mice with a deletion of Npm1 in Rorc-expressing cells (Rorc-Cre; Npm1^{flox/flox}).

7. Rorc-Cre deletes Npm1 in ILC3 and all T cells (including in Rorc⁺ Treg). How can the contribution of T cells be excluded? This should be experimentally expressed.

8. Although DSS colitis is a widely established model of acute colitis, it is not a good correlate of human IBD which is a chronic often relapsing disease. The authors should document a role of NPM1 for other colitis models such as transfer colitis where the role of NPM1 in the various cell population (ILC3, effector T cells, Treg) can be rigorously dissected.

9. As a general note, the authors talk about Npm1-deficient mice (sometimes even about KO mice; Extended Data Figure 3) when referring to heterozygote Npm1^{+/-} mice. The authors should more appropriately refer to Npm1 haploinsufficient mice or cells. In the experiments using Npm1-flox mice, they generate full Npm1-deletion in ILC3 (Rorc-Cre x Npm1^{flox/flox}). Would a heterozygote deletion show the same result? ILC3 should be better characterized in these mice. Is their development and that of lymphoid tissue normal?

10. The decreased abundance of NPM1 in IBD patients (Figure 1A,B) is interesting. Which cell types show reduced NPM1 abundance? How is NPM1 regulated?

11. The data on interaction with p65 is not ultimately convincing. It cannot be excluded that there are other binding partners of NPM1. This should be more openly stated and discussed.

12. The statement (line 306-308) "we ... found that expression of Tfam restored secretion of IL-22 in MNK3 cells in which Npm1 was knocked down (Fig. 6H,6I)" is not fully supported by the data (no complete restoration). These data may argue that Npm1 regulates other factors as well that contribute to the observed phenotype. This needs to be discussed more openly possibly in a section that discusses the limitations of the study.

Reviewer #2:

Remarks to the Author:

This manuscript reports a role for Nucleophosmin 1 (Npm1) in regulating intestinal inflammation in mice, and potentially human disease with a clear clinical rationale for investigation. The authors utilize novel mouse models to dissect the effects of Npm1 deficiency in the context of DSS colitis and perform elegant molecular biology studies to investigate the intracellular interplay of Npm1 with other transcription factors. Moreover, the authors suggest changes in ILC3 and their IL-22 production as the causative driver of worsened colitis in the absence of Npm1, with Npm1 acting to promote mitochondrial metabolism in ILC3 to underpin their effector functions. While an interesting study overall the conclusions and interpretations of the data are often extrapolated from correlations, and in some cases key data sets are not sufficiently linked together to prove causation or connect the cell-intrinsic molecular pathways to phenotypes reported elsewhere in the manuscript. Moreover, my enthusiasm is limited by a number of technical limitations and lack of supporting data to justify interpretations in key places of the manuscript.

Specific points:

The central human data to suggest a role of Npm1 in IBD and COAD patients is overinterpreted.

- Figure 1A demonstrates less Npm1 expressing cells in UC than controls. However, this may simply be explained by the infiltration of non-Npm1 expressing immune cells in IBD which would dilute the number of Npm1⁺ cells detectable by histology. No evidence is provided that Npm1 expression changes at a cellular or genetic level in these disease samples and histological quantification of Npm1⁺ cells is not sufficient to justify the interpretation that "reduced NPM1 function is a common feature in IBD" (Line 80).

- Extended Data 2A shows a range of expression between 0-400 in both level in Stage I/II and III/IV disease. The groups are highly powered and yield statistical significance but the means of the two groups are only slightly different and the data is interpreted to infer correlation between Npm1 expression and disease progression which does not seem justified based on

this data. Extended Data 2B-C split the cohort further to look at survival but the criteria, cutoff and rationale for how the data was split to generate these survival charts is not provided.

- Much of the analysis focuses on DSS only and does not provide naïve controls as important comparators to assess the magnitude of changes shown. This is important as often control animals appear to show surprisingly limited inflammation as measured by weight gain or myeloid cellular infiltrate. While it is understood this is worsened in Npm1 deficient animals, DSS would normally be expected to induce significant inflammation and disease in WT mice too, so comparisons with controls are required to provide full context.

Similarly, the effects on ILC3 frequencies and functions throughout are only shown in DSS colitis, making it unclear to what extent changes seen are present at steady state and thus predisposing to worse outcome in colitis, or only apparent in an inflammatory context.

Technical concerns about the analysis, sorting and attributed roles of ILC3:

- For key experiments ILC3 are identified and sorted as CD45 low CD90 high. This is not sufficiently robust or appropriate to provide the purity required for the studies downstream and falls below the current state of the art in the ILC field. Moreover, the authors provide no evidence that such a strategy indeed yields a pure ILC3 population at steady state or during DSS. Finally the gating strategy for such sorts, shown in Figure 3A appears highly subjective with no clear subpopulation visible. Given the high likelihood of contamination in an inflamed environment significant data is needed to demonstrate this strategy yields pure ILC3, otherwise key assays need to be repeated with an improved gating and sorting strategy in line with key studies in the ILC3 field.

- For analysis ILC3 are identified as Lin neg RORgt+ which is more appropriate, but no further resolution is provided. Given ILC3 in the gut consist of multiple distinct subsets with very different biology and transcriptional signatures it is important to provide further detail on the relative effects of Npm1 manipulation on NCR+ vs. CCR6+ ILC3 subsets, and inclusion of other canonical ILC markers e.g. CD127 should be considered to ensure accuracy of analysis and interpretation.

- RORc Cre will also delete Npm1 in conventional T cells and gdT cells but these cells are not analysed in the Npm1 flox model in any meaningful way. Despite this, the authors attribute all phenotypes to correlative changes in ILC3 and ILC3-intrinsic biology. Detailed analysis of intestinal CD4+ T cells subsets (Th17, Treg) and gdT cells is needed both in the Npm1+/- model and the conditional flox model and other possible immune dysregulation should be considered as potential contributing or causative factors in altered responses of DSS-driven inflammation.

- The reduced ILC3 frequencies in global and conditional deficient animals are interpreted as cell intrinsic, however these changes are only reported in DSS, not steady state. In DSS colitis the authors report marked increases in myeloid cells as would be expected (Extended Data 5) thus it is highly plausible that any perceived reduction in the relative frequency of ILC3 amongst CD45+ cells could be explained by significant increases in infiltrating cell types. Thus, cell numbers must be shown for all quantification to support the conclusion that ILC3 are reduced.

- ILC3 have been extensively shown to be plastic, and “ex-ILC3” with an ILC1-like phenotype may produce IFN-gamma to exacerbate inflammation. What are the frequencies and effector status of these cells in RORc Cre x Npm1 flox mice in DSS colitis?

- A key piece of data presented are the changes in mitochondrial genes in “ILC3” sorted from Npm1+/- mice (Figure 3C+D). However, these genes are not shown to be restored by Npm1 overexpression, or similarly defective in studies targeting Tfam. More needs to be done to connect the different models and link data sets.

- The authors conclude “lack of Npm1 impairs mitochondrial OXPHOS” but Npm1+/- ILC3 actually have increased spare respiratory capacity – suggesting the capacity to engage OXPHOS is not impaired, and possibly even enhanced. Importantly as all assays are done in DSS colitis the cellular environment from which these two cell populations were isolated is dramatically different, and those from Npm1+/- (which are more inflamed) have likely been exposed to significantly different cues in vivo that could alter their mitochondrial profile. A more detailed metabolic analysis of Npm1+/- or conditionally deficient ILC3 isolated from naïve animals is required to determine whether these metabolic changes are truly cell-intrinsic or rather the subject of extrinsic signals during inflammation.

- Bezafibrate treatment was done in Npm1+/- mice. Why was similar not performed in the RORc Cre x Npm1 flox model where the deficiency is at least more restrictive. Such drug treatments in a globally deficient animal are limited when attributing effects to ILC3 in the interpretation of the data.

- Figure 5G; the p65-regulated genes measured have not been shown to be expressed by ILC3 to my knowledge - and no data is provided to validate these genes are truly expressed (i.e. analysis of prior transcriptional data sets etc.). Il12a, Il6, Cxcl2, Ccl4 would all typically be associated with myeloid-cells. Given the potential for contamination in sort-purified cells (see above), significant extra supporting data is required here if utilizing this gene signature as proof of Npm1-p65 interactions in regulating ILC3-intrinsic transcription.

- Moreover, in Figures 5 and 6 the analysis of p65 and Tfam and their links to mitochondrial regulation largely rely on prior data in other systems and assumptions that the same is occurring here without direct evidence to support that in ILC3. Does manipulation of p65 and Tfam similarly change mitochondrial metabolism genes, extracellular flux capacity, and ILC3 cytokine production? What is the evidence these TFs regulate ILC3-associated effector genes or mitochondrial metabolism

programs directly? The links between these data sets and core findings are under developed.

- Given ILC3 appear to have less mitochondria (Figure 4H) and lower cell frequencies (Figure 2A) in absence of Npm1, can the authors rule out cell death/apoptosis as a reason for reduced IL-22 in colitis and other metabolic and transcriptional phenotypes?

Methodology: In a number of key data sets the details of the experiments are insufficiently described and/or methodology used is not described anywhere in the manuscript. Figure legends and methods must be sufficient to allow for a detailed understanding of what was done and how, and this should be checked throughout. Some key specific examples include:

- o Figure 1A; what is indicated by low/medium/high in regards to Npm1 staining. What cutoffs were used and how were they justified?
- o Extended Data 2B-C and 5I, authors mention data was split into “Npm1 hi” and low but no cut-off values on expression are given or rationale for how this was justified. Given data spread in Extended Data 2A in Stage I/II patients is wide it is unclear if this is simply arbitrary.
- o DSS experiments, DAI used throughout as key output but not clearly defined anywhere.
- o Npm1-deficiency in Extended Data Figure 2D was checked at protein level, but not at RNA level. Moreover, in the aforementioned blot no controls are provided (e.g. isotype antibodies, or known negative controls). The methodology used here and in other blots is also not described anywhere in the methods.
- o Figure 3G; single cell data was used to identify DEG in Npm1 “high” and “low” cells. Npm1 is almost uniformly expressed in this cell cluster (Extended Data 6) and single cell RNA data is not quantitative due to dropout – thus lack of mRNA of an individual cell within a given cluster cannot necessarily be indicative of lacking expression. Either way no details or rationale are provided for what was considered high or low or whether this was purely arbitrary.
- o Mitochondrial potential assay in Figure 4E is not described anywhere in methods. Representative data (microscopy or flow cytometry) should be provided and further supported with other measures of mitochondrial mass and potential (e.g. TMRE).
- o Validation of successful flox-mediated deletion of targeted Npm1 exon in ILC3 is not provided in RORc Cre x Npm1 flox mice.

Minor points:

- Extended Data 6 and Figure 3I; t-SNE clusters need to be identified and labelled – numbering of clusters alone is not sufficiently informative and cell types should be clearly indicated.

Reviewer #3:

Remarks to the Author:

In this study, NMP1 is identified as an important factor in the control of intestinal inflammation and mitochondrial respiration. NMP1 levels are observed to be lower in the colon of IBD and mice lacking an Nmp1 allele exhibit exacerbated colonic inflammation and colitis-associated tumor formation. Bone marrow transplant experiments identify the hemopoietic compartment as the origin of this enhanced colitis and the authors identify a reduced frequency of ILC3s as a possible driving factor. Indeed, Rorc-driven deletion of Nmp1 recapitulated much of the macrophenotypes observed in Nmp1^{+/-} mice. Transcriptomic analysis of Nmp1-deficient ILC3s revealed significant perturbations in mitochondrial gene expression and Nmp1 loss reduced mitochondrial load and impaired OXPHOS in ILC3s. Mechanistically, NMP1 was found to interact with p65 to control Tfam expression, a regulator of mitochondrial transcription. This is an elegant study revealing a novel pathway important in the regulation of ILC3 biology, inflammation, and mitochondrial biology. The manuscript is well written and clear to follow

The authors may consider the following comments:

- Lines 138/139: Despite the text claiming the contrary, there are significant differences in the infiltration of neutrophils, macrophages and eosinophils in Nmp1^{+/-}. This also adds to a lack of clarity with regard to the rationale for targeting ILC3s, despite the very convincing subsequent data. What is the contribution of these myeloid cells to exacerbated colitis in NMP1^{+/-} mice? Is the increased flux of myeloid cells into the colon lost in NMP1 flox Rorc Cre mice?
- This is linked to a further question with regard to the specificity of mitochondrial regulation by NMP1, is this an ILC3 specific phenomenon or is this a more widespread mechanism in other cell types?
- Although deciphering exactly how mitochondria control ILC3 activation/maintenance/IL-22 production is perhaps beyond the scope of this study. More work needs to be done to flesh this part of the study out. Specifically:
 - o Does ablation of other key components phenocopy Nmp1 loss on ILC3 activation and metabolism, such as Tfam and p65?
 - o Whilst Tfam targets mitochondrial genes for transcription, its reduction can have wider consequences for mitochondria. To that end, it is important to understand more robustly which NMP1-driven mitochondrial perturbations result in altered ILC3 activation. Is it specifically OXPHOS and the running of the ETC? This can be probed with ETC complex inhibitors or knockdowns in the ILC3 cell line. Is the TCA cycle functioning normally in NMP1-deficient ILC3s? Can NMP1-deficient cells be rescued by targeting mitochondrial metabolism in other ways, such as via succinate, a substrate important for both the

TCA cycle and the ETC?

Minor points

- There is a potential flow cytometry compensation issue in eosinophil plot in extended data figure 4 that if corrected may suggest the population immediately below the eosinophil gate is the true population.
- Some Y-axes not originating at 0 (e.g. Fig 2C and Fig. 4I).
- Figure 1A: The scale bar size in the legend should be corrected to match the figure
- Fig. 1G-HI: the legend should indicate more clearly the source of mRNA, is it whole colon or epithelia etc?

Author Rebuttal letter:

Dear Editors and Reviewers:

Thank you for your critical comments and constructive suggestions on our manuscript entitled "Nucleophosmin 1 Promotes Mucosal Immunity by Supporting Mitochondrial Oxidative Phosphorylation and ILC3 Activity" (NI-A35998A). Those comments are all valuable and very helpful for revising and improving our paper. We have studied comments carefully and have made correction which we hope meet with approval. Revised portion are marked in red in the paper. The main corrections in the paper and the responds to the reviewer's comments are as following:

Reviewer #1

(Remarks to the Author)

The authors investigate the role of reduced expression of Nucleophosmin (NPM1) for inflammatory bowel disease and colitis-associated cancer. NPM1 mutations are associated with myelodysplastic syndrome (MDS) and MDS patients often also have IBD. The molecular link between these diseases remains unknown. The authors show that mice with reduced expression of the Npm1 gene (Npm1^{+/-} mice) are more susceptible to DSS-induced colitis. In Npm1^{+/-} mice ILC3 were found to produce less of the epithelia-protective cytokine IL-22. On a mechanistic level, ILC3 from Npm1^{+/-} mice showed defects in mitochondrial biogenesis and reduced OXPHOS. NPM1 could physically interact with p65 (RelA) transcriptionally activating the expression of mitochondrial transcription factor A (Tfam), required for mitochondrial DNA transcription required for mitochondrial fitness and IL-22 production in colonic ILC3. Key data are also observed in human cohorts.

Overall, this is an interesting manuscript linking NPM1 to intestinal protection via IL-22 and ILC3. However, there are currently some limitations and conceptual questions that should be addressed in a revised manuscript. The following issues are raised.

1. NPM1 is expressed quite broadly both in hematopoietic and non-hematopoietic cells. A wider characterization of Npm1^{+/-} mice is required at steady-state and during DSS colitis. In particular, bone marrow and intestinal immune cells as well as epithelial cells should be characterized in some detail (representation, differentiation, function).

Reply ¶ We appreciate your interest in our manuscript and the constructive comments. We agree with that NPM1 is expressed in various cell types and that a comprehensive characterization of Npm1^{+/-} mice is necessary to elucidate its role in intestinal protection. To address this point, we have performed extensive flow cytometry analysis of multiple intestinal immune cells, including eosinophils, macrophages, neutrophils, DC, Th17, Treg and $\hat{\text{I}}\hat{\text{T}}$ in both steady-state and DSS-induced colitis conditions (Extended Data Fig.5A-5H, Extended Data Fig.5N-5S). The ratio of macrophages, neutrophils, eosinophils and DC infiltrated in colon exhibited few changes between WT and Npm1^{+/-} mice in steady state. However, in DSS-induced colitis, an elevation of these cells was observed in Npm1^{+/-} mice compared to WT mice (Extended Data Fig.5A-5H). In addition, ratio of Th17, Treg and $\hat{\text{I}}\hat{\text{T}}$ cells infiltrated in the colon of WT and Npm1^{+/-} mice were similar under physiological condition and pathological condition (Extended Data Fig.5N-5S). We also detected the expression of tight junction genes (including Zo-1, Zo-2, Claudin-2 and Claudin-3) in epithelial cells, which are pivotal for the maintenance of intestinal barrier function. With the exception of Claudin-3, which is diminished in Npm1-haploinsufficient mice under physiological conditions, the

expression of other tight junction genes remains relatively unchanged between two group of mice in both physiological and pathological conditions (Extended Data Fig.4G-4J). Considering the role of NPM1 in MDS, we further measured the percentage of LK, LSK, CMP, GMP and MEP in bone marrow from NPM1^{+/+} and NPM1^{+/-} mice both in steady-state and DSS-induced colitis conditions. Results indicated that ratio of LK cells, LSK cells and LSlowKlow cells in bone marrow of Npm1^{+/-} mice were elevated compared with that of Npm1^{+/+} mice both in steady state and DSS-induced colitis condition, which is a characteristic phenotype of MDS. Meanwhile, within the LK cell population, ratio of GMP increased in Npm1^{+/-} mice, especially in steady state (Extended Data Fig.2O, 2P). These results have been added to revised figure, and the results section has been modified and highlighted.

The revisions include the following:

âGiven the critical role of NPM1 in MDS and AML, we also examined the change of bone marrow cells in Npm1^{+/-} mice (Extended Data Fig.3A). Results indicated that ratio of LK cells, LSK cells and LSlowKlow cells in bone marrow of Npm1^{+/-} mice were elevated compared with that of Npm1^{+/+} mice both in steady state and DSS-induced colitis condition (Extended Data Fig.2O, 2P), which is a characteristic phenotype of MDS. Meanwhile, within the LK cell population, proportion of granulocyte-macrophage progenitors (GMP) increased in Npm1^{+/-} mice, especially in steady state (Extended Data Fig.2O,2P)â from lines 94 to 100.

âWe also detected the expression of tight junction genes (including Zo-1, Zo-2, Claudin-2 and Claudin-3) in epithelial cells, which are pivotal for the maintenance of intestinal barrier function. With the exception of Claudin-3, which is diminished in Npm1-haploinsufficient mice under physiological conditions, the expression of other tight junction genes remains relatively unchanged between two group of mice in both physiological and pathological conditions (Extended Data Fig.4G-4J)â from lines 147 to 152.

âThe ratio of macrophages, neutrophils, eosinophils and DC infiltrated in colon exhibited few changes between WT and Npm1^{+/-} mice in steady state (Extended Data Fig.5A-5D). However, in DSS-induced colitis, an elevation of these cells was observed in Npm1^{+/-} mice compared to WT mice (Extended Data Fig.5E-5H).....Likewise, evaluation of T cells (Th17, Treg and Î³Î³T cells) coupled with comparable colitis in two genotype mice after deletion of CD3⁺ T cells indicates that exacerbated colitis in Npm1-haploinsufficient mice was not attributed to T cellsâ from lines 164 to 174.

2. The authors show that NPM1 controls mitochondrial biogenesis and function in ILC3. Considering that this is quite a basic metabolic program, it is surprising that the authors only detect changes in ILC3. Why is NPM1 so central to ILC3 function? Why is reduced expression of NPM1 in Npm1^{+/-} mice compensated in other cell types that also express NPM1 and also require OXPHOS for their function. While I can see that this is a broad question and a full response would be beyond the scope of the current manuscript, the authors should exclude a role of epithelial NPM1 or of NPM1 in myeloid cells or T cells in the phenotype they observe.

Reply!4This question is important and we have performed a serial experiments to address it. We had found TFAM as an important regulator of mitochondrial function under NPM1, further investigation showed that remarkable up-expression of Tfam only happened in ILC3s in response to DSS treatment, not in macrophages, T cells, and epithelial cells (revised Figure 6A-6C). However, Tfb1m and Tfb2m were significantly increased in macrophages, T cells, and epithelial cells, but not in ILC3s after DSS treatment (Fig.6A-C). Meanwhile, we analyzed the single cell RNA-seq data (GSE211578) of colon of DSS-treated and control mice from Dr. Zhang lab and found that ILC3s exhibit higher expression of Tfam but lower expression of Tfb1m and Tfb2m under DSS condition, compared to other cell types (see below Fig.1, note that this data set has not been published in authorâs article, so we could not show it in our manuscript). These data suggested that mitochondrial activation in ILC3s is primarily dependent on TFAM, rather than on TFB1M or TFB2M. Contrast to ILC3s, macrophages, T cells, and epithelial cells are primarily depend on TFB1M and/or TFB2M in DSS-treated mice.

Fig.1 Single-cell analysis of Tfam, Tfb1m and Tfb2m in DSS-induced mice and Ctrl mice (GSE211578). Dot size represents the fraction of cells within the cluster that express each gene and colors indicate the z-scaled expression of genes in cells within each cluster

In addition, we sorted macrophage, T cell and epithelial cells from the intestines of mice in physiological and inflammatory conditions and performed mitochondrial stress test, which showed that haploinsufficiency of Npm1 did not have a significant effect on the oxidative phosphorylation of these cells in both steady-state and DSS-induced colitis conditions (Extended Data Fig.9L-9T). Consistent with data of oxidative phosphorylation, expression levels of majority mtDNAs in macrophage, T cell and epithelial cells were not significantly changed by haploinsufficiency of Npm1 (Extended Data Fig.9A-9C).

The above data suggest that Npm1 has a significant effect on mitochondrial function and metabolism in ILC3 after stimulation, while having little effect on other cell types. Therefore, although NPM1 expression is reduced in UC patients (Extended Data Fig.2B) and enteric mice (Extended Data Fig.4A), with no discernible cell-specific characteristics, the up-regulation of TFAM, downstream gene of Npm1, specifically required in ILC3 cells (Fig.6A-C, below Fig.1) explains why the insufficiency of NPM1 preferentially affects function of ILC3. These results have been added to Figure 6, Extended Data Fig.2, Fig.4 and Fig.9. We have modified the description in lines 237-238, lines 268-270 and lines 363-377

The revisions include the following:

âHowever, the universal decrease of mtDNA was not observed in epithelial cells, macrophages or T cells of Npm1^{+/-} mice (Extended Data Fig.9A-9C)â from lines 240 to 242.

âBesides, Epithelial cells, macrophages and T cells in Npm1^{+/-} mice exhibited few differences in OXPHOS compared to that in WT mice in both steady state and DSS-induced colitis condition (Extended Data Fig.9L-9T).â from lines 274 to 276.

âThe expression levels of the three mitochondrial transcription factors in ILC3s showed no differences between Npm1^{+/+} and Npm1^{+/-} mice in steady state. However, under pathological conditions, a remarkable decrease in Tfam expression was only observed in ILC3s, not macrophages, T cells, and epithelial cells, of Npm1 haploinsufficient mice when compared to Npm1^{+/+} mice (Fig.6A-C), suggesting that NPM1 plays an indispensable role in up-regulation of Tfam in ILC3s upon DSS-treatment. However, Tfb1m and Tfb2m were significantly increased in macrophages, T cells, and epithelial cells, but not in ILC3s after DSS treatment (Fig.6A-C). These data suggested that mitochondrial activation in ILC3s is primarily dependent on TFAM, rather than on TFB1M or TFB2M. Overexpression of Tfam in MNK3 markedly enhanced the expression of mtDNAs including mt-Nd1, mt-Nd2, mt-Nd3, mt-Nd4 and mt-Atp6 (Extended Data Fig.10J). Knockdown of Tfam in MNK3 notably impaired its mitochondrial function and attenuated the production of IL22 (Fig.6D-6H). Accordingly, NPM1 is crucial for the heightened demand of TFAM to subsequently increase mitochondrial function in ILC3s, not other cell types, during DSS-induced colitisâ from lines 363 to 376.

3. The data support a concept in which ILC3 do not require high level expression of NPM1 at steady-state but are dependent on full NPM1 expression during colitis. How can this be explained? Do ILC3 undergo significant metabolic changes during the various phases of DSS colitis (acute tissue damage vs. repair phase)? This needs to be addressed by additional experiments.

Reply ¶¼ Thank you for your constructive feedback. To investigate the metabolic changes of ILC3 in various phases of DSS colitis, we sorted ILC3 from mice in different colitis stages (day0, day 5 and day 10) and performed mitochondrial stress tests. The results demonstrated that ILC3 cellular oxidative phosphorylation levels were significantly elevated in the acute phase compared to the physiological state, whereas ILC3 metabolic levels tended to normalize in the repair phase. These results have been added to Extended Data Fig.9H-9K and we have modified the description in lines 270-274 : âIn DSS model, mouse intestinal ILC3s exhibited a dramatically mitochondrial activation in acute tissue damage phase (day 5) and then partially restored to normal state in repair phase (day 10) (Extended Data Fig.9H-9K). The inadequate mitochondrial activation of ILC3 in acute phase caused by heterozygous deletion of Npm1 could lead to exacerbated colitis (Fig.4B-4D)â.

Our explanation is as follows. ILC3 cellular oxidative phosphorylation levels were markedly elevated during the acute phase, compared to physiological conditions (Extended Data Figure 9H-9K). To address the heightened mitochondrial functional demands, the expression of TFAM in ILC3s was significantly upregulated in response to DSS treatment (Figure 6A-C). NPM1 has the capacity to bind to p65 and modulate the transcription of TFAM. However, under steady-state conditions, p65 predominantly resides in the cytoplasm, where NPM1, localized to the nucleus (Figure 5E), exhibits minimal regulatory effects on TFAM expression. In response to DSS induction, p65 translocates to the nucleus, where it interacts with NPM1 to enhance the activation of Tfam transcription (Figure 5E), thereby meeting the increased requirements for TFAM and mitochondrial function. This explains why ILC3 do not require high level expression of NPM1 at steady-state but are dependent on full NPM1 expression during colitis.

In this context, the data with MNK cells and IL-22 is surprising. The authors should add conditions without stimulation with IL-1b and IL-23. Would IL-1b and/or IL-23 change metabolic function of MNK cells?

In this study, all the experiments involving MNK3 cells included unstimulated controls. We have added data of the untreated group, seen in Figure 5N, 6H and Extended Data Fig.10A-10D. We also complemented the metabolic differences between shNC and shNpm1 in MNK3 cells with or without IL-1 β and IL-23 treatment. Control cells underwent a significant enhancement of ATP production, maximal respiration and basal OCR after IL-1 β and IL-23 treatment, whereas metabolic activation of shNpm1 MNK3 cells was inhibited. These results have been added in Extended Data Fig.10A-10D and highlighted in the revised version of the manuscript in lines 308-309.

The revisions include the following:

âMNK3 also exhibited mitochondrial activation after IL-1 β /IL-23 stimulation, which was regulated by NPM1â from lines 308 to 309.

4. Subsets of ILC3 are needed for prenatal organogenesis of secondary lymphoid tissue (Peyer's patches, lymph nodes) and postnatal formation of solitary intestinal lymphoid tissue. Are these normally represented in Npm1 $^{+/-}$ mice?

Reply ¶ We analyzed the changes of Peyer's patches, lymph nodes and SILT in Npm1 $^{+/+}$ vs Npm1 $^{+/-}$ mice, and no significant differences were observed. The results have been added to Extended Data Fig.2K-2N of the revised version of the manuscript. We have added âConcurrently, the organogenesis of secondary lymphoid structures, including Peyer's patches (PP) and mesenteric lymph nodes (MLN), as well as solitary intestinal lymphoid tissue (SILT), was unaffected by Npm1 haploinsufficiencyâ in lines 91 to 94.

5. Data during the last years have documented an important effect of the intestinal microbiota for the severity of DSS colitis. Are there differences in the representation of microbiota between Npm1 $^{+/+}$ and Npm1 $^{+/-}$ mice at steady-state or during colitis? Would crossfostered and/or co-housed littermates show the same differences? The breeding schemes and housing conditions should be explained in the Methods section.

Reply ¶ Indeed, a growing number of studies have confirmed the important role of intestinal flora in inflammatory bowel disease. IL-22-producing ILC3s are critical for the maintenance of microbiota homeostasis (Guo et al., 2015; Hutnick et al., 2012), and microbiota dysbiosis contribute to the pathogenesis of colitis (Belkaid and Hand, 2014). That's one of the things we're very concerned about.

We have done the 16s-rRNA seq to analyze the change of microbiota between Npm1 $^{+/+}$ and Npm1 $^{+/-}$ mice after DSS treatment (Extended Data Fig.6K-6Q). There was a significant decrease in observed operational taxonomic unit (OTU), Chao1 index and shannon index in Npm1 $^{+/-}$ mice (Extended Data Fig.6K-6M), indicating that Npm1 deficiency leads to a repression of microbiota diversity. Additionally, a comparison of bacterial composition between feces from Npm1 $^{+/-}$ mice and WT

mice revealed remarkable differences (Extended Data Fig.6N-6Q). NPM1 deletion creates an intestinal environment that favors the competitive advantage of pro-inflammatory over anti-inflammatory species. The relative abundance of Desulfovibrionaceae, Erysipelotrichaceae, and Helicobacteraceae was significantly increased in Npm1 +/- mice. In contrast, the relative abundance of beneficial microbes, such as Muribaculaceae, Clostridiaceae, Ruminococcaceae, and Tannerellaceae, was decreased in DSS-treated Npm1 +/- mice (Extended Data Fig.6Q). We have added the descriptions to lines 189-194.

We apologize for not describing the breeding schemes of the mice in detail. Different breeding and subcage grouping methods may have an impact on experimental results, especially microbiota-related experiments. We used Npm1 +/+ and Npm1 +/- mice mated for breeding and littermates for experimental and control. Npm1 +/+ and Npm1 +/- mice are generally genotyped and caged at 3 to 4 weeks of age. With regard to caging and grouping, we have used two ways of caging at the beginning of the project: (1) mice of different genotypes were mixed in the same cage, distinguishing only between the sexes, and (2) Npm1 +/+ and Npm1 +/- mice were placed in separate cages. Note that Npm1 +/- mice is much more susceptible to DSS-induced colitis than Npm1 +/+ mice no matter they are co-housed or not, so we used strategy 2 later. Breeding strategies for transgenic mice have been added to the Methods of the revised manuscript in lines 715-716.

The revisions include the following:

16S-rRNA sequence indicated that there was an rapid decrease in observed operational taxonomic unit (OTU), Chao1 index and shannon index in Npm1 +/- mice (Extended Data Fig.6K-6M), indicating that microbiota diversity was repressed caused by Npm1 heterozygote deletion. Moreover, feces from Npm1 +/- mice and WT mice showed a remarkable changes in bacterial composition (Extended Data Fig.6N-6Q) from lines 189 to 194.

Note that littermate mice are generally genotyped at 3 to 4 weeks of age and then placed in separate cages when grouping, according to its genotype from lines 718 to 719.

6. The authors state (line 138/39) that accumulation of populations of macrophages, neutrophils, and eosinophils was similar in Npm1 +/- mice compared to WT mice (Extended Data Fig. 5A,5B). However, the data shows that the representation of macrophages, neutrophils, and eosinophils is significantly increased in Npm1 +/- mice. The statement needs to be corrected and a comparable analysis of macrophages, neutrophils, and eosinophils should be performed in mice with a deletion of Npm1 in Rorc-expressing cells (Rorc-Cre; Npm1 flox/flox).

Reply 1/4 Thank you for your careful reading and your constructive comment. We apologize for the mistake in our statement about the accumulation of macrophages, neutrophils, and eosinophils in Npm1 +/- mice. As showed in Extended Data Fig.5A-5H, Macrophages, neutrophils, and eosinophils are significantly increased in Npm1 +/- mice. We have corrected and highlighted it in lines 164-167 of the revised manuscript.

We further analyzed the change of macrophage, neutrophils, DC and eosinophils by flow cytometry in RorcCre/+Npm1 flox/flox mice in steady and DSS state (Extended Data Fig.8A-8H). Accumulation of macrophage, neutrophils, DC and eosinophils was also significantly increased in the inflammatory intestines of RorcCre/+Npm1 flox/flox mice compared to Npm1 flox/flox mice. These results are consistent with the data obtained in Npm1 +/- mice and have been added to lines 214-216 in revised manuscript.

The revisions include the following:

The ratio of macrophages, neutrophils, eosinophils and DC infiltrated in colon exhibited few changes between WT and Npm1 +/- mice in steady state (Extended Data Fig.5A-5D). However, in DSS-induced colitis, an elevation of these cells was observed in Npm1 +/- mice compared to WT mice (Extended Data Fig.5E-5H) from lines 164 to 167.

Additionally, consistent with changes observed in Npm1^{+/-} mice, the increased infiltration of myeloid cells also existed in Rorc^{cre/+}Npm1^{flox/flox} mice under pathological condition (Extended Data Fig.8A-8H) from lines 214 to 216.

7. Rorc-Cre deletes Npm1 in ILC3 and all T cells (including in Rorc⁺ Treg). How can the contribution of T cells be excluded? This should be experimentally expressed.

Reply: Thank you for this great suggestion. Percentages of T cell subsets (Th17, Treg and $\hat{\rho}$ T) in the colon of Rorc^{cre/+}Npm1^{flox/flox} and Npm1^{flox/flox} mice were similar in physiological state and DSS colitis. Notably there was a slight reduction in the proportion of Treg in the intestines of Rorc^{cre/+}Npm1^{flox/flox} mice in the physiological state. To exclude the contribution of T cells, we have further performed DSS model in Rorc^{cre/+}Npm1^{flox/flox} mice after T cell depletion by anti-CD3 antibody. The results showed that enteritis was still exacerbated in Rorc^{cre/+}Npm1^{flox/flox} mice compared to Npm1^{flox/flox} mice after T cell clearance, suggesting that the regulation of enteritis by Npm1 is mainly dependent on ILC3s, rather than T cells (Extended Data Fig.8I-8Q). The results section has been modified and highlighted in lines 216-219 in revised manuscript.

The revisions include the following:

These results have been added: Since RORc-Cre will also delete Npm1 in conventional T cells and $\hat{\rho}$ T cells, we examined the function of various subsets of T cells and excluded their contributions to exacerbated colitis in Rorc^{cre/+}Npm1^{flox/flox} mice by depleting T cells using a CD3 antibody (Extended Data Fig.8I-8Q) from lines 216-219.

8. Although DSS colitis is a widely established model of acute colitis, it is not a good correlate of human IBD which is a chronic often relapsing disease. The authors should document a role of NPM1 for other colitis models such as transfer colitis where the role of NPM1 in the various cell population (ILC3, effector T cells, Treg) can be rigorously dissected.

Reply: T cell transfer model of colitis is the most widely used model to dissect the initiation, induction, and regulation of immunopathology in chronic colitis mediated by T cells. Since our depletion of CD3 experiments showed that T cells are likely not involved in DSS-induced colitis in Npm1^{+/-} mice, we didn't explore the role of NPM1 in transfer colitis models. Alternatively, we added the TNBS-induced colitis model to enhance the reliability of our conclusions. In this model, colitis was also exacerbated in Npm1^{+/-} mice (Extended Data Fig.2Q-2U), which showed a significant reduction in both the proportion of ILC3 and the ability to secrete IL-22 (Extended Data Fig.6A-6D). Moreover, no significant differences were observed in the proportions of Th17, Treg, and $\hat{\rho}$ T cells in the intestines of Npm1^{+/+} and Npm1^{+/-} mice (see below Fig.2, Data not shown in the manuscript). These data obtained from TNBS-induced colitis model are consistent with the data from DSS-treated Npm1^{+/-} mice, suggesting the important role of NPM1 in various colitis. They were described in lines 111-115, lines 179-180 in revised manuscript.

Fig. 2 Analysis of Th17, Treg, and $\hat{\rho}$ T cells in the intestines of Npm1^{+/+} and Npm1^{+/-} mice in TNBS-induced colitis model

The revisions include the following:

To more broadly explore the role of NPM1 in intestinal inflammation, we also established a trinitrobenzene sulfonic acid (TNBS) induced colitis model and evaluated the progress of colitis in WT and Npm1^{+/-} mice. As anticipated, Npm1^{+/-} mice also exhibited reduced colon length and enhanced inflammation, together with greater body weight loss and increased DAI (Extended Data Fig.2Q-2U) from lines 111 to 115.

Additionally, Npm1^{+/-} ILC3 exhibited similar alterations in TNBS-induced colitis (Extended Data Fig.6A-6D) from lines 179 to 180.

9. As a general note, the authors talk about Npm1-deficient mice (sometimes even about KO mice; Extended Data Figure 3) when referring to heterozygote Npm1^{+/-} mice. The authors should more appropriately refer to Npm1 haploinsufficient mice or cells. In the experiments using Npm1-flox mice, they generate full Npm1-deletion in ILC3 (Rorc-Cre x Npm1flox/flox). Would a heterozygote deletion show the same result? ILC3 should be better characterized in these mice. Is their development and that of lymphoid tissue normal?

Reply: Thanks for your suggestion. We have replaced "Npm1-deficient mice" to "heterozygous Npm1^{+/-} mice" or "Npm1 haploinsufficient mice" in the manuscript.

We generated Npm1 haploinsufficient mice (Npm1^{+/-}) mice, because homozygous knockout was lethal. We further analyzed the change of colitis and ILC3s in Npm1 conditional heterozygous-ablation mice: Rorc^{cre}/+Npm1flox/+ mice. The results from Rorc^{cre}/+Npm1flox/+ mice are similar to those from Rorc^{cre}/+Npm1flox/flox mice. Exacerbation of enteritis is observed in Rorc^{cre}/+Npm1flox/+ mice following DSS induction, accompanied by a reduction in the proportion of ILC3 and a decrease in IL-22 secretion capacity (Extended Data Fig.7K-7Q). However, the degree of reduction in Rorc^{cre}/+Npm1flox/+ mice (30%-50%) is not as great as that in Rorc^{cre}/+Npm1flox/flox (60%-70%), indicating that the quantity of NPM1 expression is vital in ILC3 cells to response to DSS. And no significant differences were observed in the number and size of Peyer's patches and lymph nodes between Npm1flox/+ and Rorc^{cre}/+Npm1flox/+ mice (Extended Data Fig.7H-7J). These data have been added to lines 207-209 in revised manuscript.

The revisions include the following:

"Without development defects, heterozygous deletion of Npm1 in ILC3 also contributed to exacerbated enteritis and reduction of ILC3, appears to be a dose-dependent manner (Extended Data Fig.7H-7Q)." from lines 207 to 209.

10. The decreased abundance of NPM1 in IBD patients (Figure 1A,B) is interesting. Which cell types show reduced NPM1 abundance? How is NPM1 regulated?

Reply: To determine which cell types exhibit reduced NPM1 abundance, we analyzed single-cell sequencing data from normal controls and UC patients' intestinal tissues (GSE182270 reported by Mathieu Uzzan et al., Nature Medicine, 2022). We found that NPM1 expression decreased mainly in ILC3, macrophages, NKT cells, cytotoxic T cells, Tregs and Paneth cells in UC tissues compared with normal intestinal tissues (Extended Data Fig.2B). Moreover, in our DSS-induced colitis model, we also showed that expression of Npm1 exhibited a decrease in ILC3s, macrophages, T cells and epithelial cells (Extended Data Fig.4A). We have modified the description in lines 76-78 and 137-139 in the manuscript.

Furthermore, we focused more on how NPM1 is regulated in ILC3. We checked the expression of predicted transcription factors of NPM1 (predicted in GeneCards database) in GSE182270, and found that the expression of STAT3, IRF1, and GATA3 was decreased in ILC3 of UC patients (Extended Data Fig.10K). Note that GATA3 is a key transcription factor in ILC3, which may answer the question why NPM1 is down-regulated in ILC3 of UC patients. Validation experiments conducted on ILC3 cells from normal and colitis mice confirmed decreased expression of GATA3, STAT3, and IRF1 in colitis mouse ILC3 (Extended Data Fig.10L). We have added the description to lines 399-405 in the manuscript.

The revisions include the following:

"Further scRNA-seq analysis (GSE182270) on colonic biopsies of UC patients and healthy control (HC) indicated that the expression of NPM1 decreased mainly in ILC3s, macrophages, NKT cells, cytotoxic T cells, Tregs and Paneth cells in UC patients (Extended Data Fig.2A, 2B)" from lines 76 to 78.

Given that Npm1 is expressed by many types of cells and decreased under pathological condition (Extended Data Fig.2B, 4A), it is unclear whether the exacerbated colitis in Npm1 +/- mice is due to defects in hematopoietic or non-hematopoietic cells, particularly colonic epithelial cells from lines 137 to 139.

Considering the reduction of NPM1 in human colitis, as well as the exacerbated colitis phenotype in mice with insufficient NPM1 expression across the entire system or specifically in ILC3s, our subsequent investigation aimed to explain how NPM1 is regulated and whether overexpression of NPM1 could ameliorate colitis. GATA3, IRF1, and STAT3, which are predicted transcriptional factors associated with NPM1, demonstrated reduced expression in ILC3s of UC patients and enteritic mice, in comparison to the control groups (Extended Data Fig.10K, 10L). These findings may provide insights into the mechanisms underlying the downregulation of NPM1 in IBD from lines 399 to 405.

11. The data on interaction with p65 is not ultimately convincing. It cannot be excluded that there are other binding partners of NPM1. This should be more openly stated and discussed.

Reply: Thank you very much for your suggestions. Of course there are many other binding partners of NPM1 which have been reported to participate in the mitochondrial functions, such as c-Myc, SP1, p53 and IRF1. We have done the mass spectrometry and found a list of partners of NPM1 (Figure 5C), one of the main interaction partners is P65. Knockdown of p65 in MNK3 cells impaired the mitochondrial function of MNK3 and decreased IL22 secretion under the stimulation of IL-1 β and IL-23. This is similar to the effect of NPM1 defects in ILC3 cells. So these data support that p65 is a critical partner of NPM1 in the regulation of ILC3 mitochondrial function and activation. We have discussed it more in lines 453-459 of the revised manuscript.

The revisions include the following:

Meanwhile, although it cannot be ruled out that NPM1 affects ILC3 mitochondrial function and cell activation through interactions with other molecule participates in the mitochondria functions, such as NPM1's known partner c-Myc, SP1, p53 and IRF1, the effects of p65-knockdown and Tfam-knockdown on ILC3 function in MNK3 cells are similar to those of knocking down Npm1. Therefore, it's believed that the p65-TFAM axis is an important effector for NPM1 to increase the function and metabolism of ILC3 from lines 453 to 459.

12. The statement (line 306-308) "we ... found that expression of Tfam restored secretion of IL-22 in MNK3 cells in which Npm1 was knocked down (Fig. 6H,6I)" is not fully supported by the data (no complete restoration). These data may argue that Npm1 regulates other factors as well that contribute to the observed phenotype. This needs to be discussed more openly possibly in a section that discusses the limitations of the study.

Reply: Thank you for your critical comments. Knockdown of Tfam in MNK3 cells impaired the mitochondrial function of MNK3 and decreased IL22 secretion under the stimulation of IL-1 β and IL-23, which is similar to the effect of NPM1 defects in ILC3s. So we believe that p65-Tfam axis is a critical downstream effector of NPM1 in the regulation of ILC3 mitochondrial function and activation. It could be that other downstream genes of NPM1 also contribute to the colitis, so we have discussed it more in lines 453-459 of the revised manuscript.

The revisions include the following:

Meanwhile, although it cannot be ruled out that NPM1 affects ILC3 mitochondrial function and cell activation through interactions with other molecule participates in the mitochondria functions, such as NPM1's known partner c-Myc, SP1, p53 and IRF1, the effects of p65-knockdown and Tfam-knockdown on ILC3 function in MNK3 cells are similar to those of knocking down Npm1. Therefore, it's believed that the p65-TFAM axis is an important effector for NPM1 to increase the function and metabolism of ILC3 from lines 453 to 459.

Reviewer #2

(Remarks to the Author)

This manuscript reports a role for Nucleophosmin 1 (Npm1) in regulating intestinal inflammation in mice, and potentially human disease with a clear clinical rationale for investigation. The authors utilize novel mouse models to dissect the effects of Npm1 deficiency in the context of DSS colitis and perform elegant molecular biology studies to investigate the intracellular interplay of Npm1 with other transcription factors. Moreover, the authors suggest changes in ILC3 and their IL-22 production as the causative driver of worsened colitis in the absence of Npm1, with Npm1 acting to promote mitochondrial metabolism in ILC3 to underpin their effector functions. While an interesting study overall the conclusions and interpretations of the data are often extrapolated from correlations, and in some cases key data sets are not sufficiently linked together to prove causation or connect the cell-intrinsic molecular pathways to phenotypes reported elsewhere in the manuscript. Moreover, my enthusiasm is limited by a number of technical limitations and lack of supporting data to justify interpretations in key places of the manuscript.

Specific points:

The central human data to suggest a role of Npm1 in IBD and COAD patients is overinterpreted.

1- Figure 1A demonstrates less Npm1 expressing cells in UC than controls. However, this may simply be explained by the infiltration of non-Npm1 expressing immune cells in IBD which would dilute the number of Npm1+ cells detectable by histology. No evidence is provided that Npm1 expression changes at a cellular or genetic level in these disease samples and histological quantification of Npm1+ cells is not sufficient to justify the interpretation that "reduced NPM1 function is a common feature in IBD" (Line 80).

Reply: Thank you for your detailed review and your constructive criticism. We appreciate your interest in our study and your valuable feedback. We agree that our human data on NPM1 expression in IBD and COAD patients is not sufficient to support our interpretation that reduced NPM1 function is a common feature in IBD. To determine which type of cells exhibits reduction in NPM1 abundance, we analyzed single-cell sequencing data from normal controls and UC patients' intestinal tissues (GSE182270) (Mathieu Uzzan et al., Nature Medicine, 2022). We found that NPM1 expression decreased mainly in ILC3, macrophages, NKT cells, cytotoxic T cells, Tregs and Paneth cells in UC tissues compared with normal intestinal tissues (Extended Data Fig.2A,2B). Moreover, in our DSS-induced colitis model, we also showed that expression of Npm1 exhibited a decrease in ILC3s, macrophages, T cells and epithelial cells (Extended Data Fig.4A). We have modified the description in lines 76-78 and 137-139 in the manuscript. The inappropriate descriptions "reduced NPM1 function is a common feature in IBD" have been modified as "NPM1 may be involved in the pathological of IBD, especially UC, and may contribute to tumorigenesis" in lines 83-85, to accurately reflect our updated findings and interpretations. We thank the reviewer for raising this important point and for providing valuable suggestions on how to improve our manuscript.

The revisions include the following:

"Further scRNA-seq analysis (GSE182270) on colonic biopsies of UC patients and healthy control (HC) indicated that expression of NPM1 decreased mainly in ILC3s, macrophages, NKT cells, cytotoxic T cells, Tregs and Paneth cells in UC patients (Extended Data Fig.2A,2B)" from lines 76 to 78.

"These findings suggested that NPM1 may be involved in the pathological of IBD, especially UC, and may contribute to tumorigenesis" from lines 83 to 85.

"Given that Npm1 is expressed by many types of cells and decreased under pathological condition (Extended Data Fig.2B, 4A), it is unclear whether the exacerbated colitis in Npm1 +/- mice is due to defects in hematopoietic or non-hematopoietic cells, particularly colonic epithelial cells" from lines 136 to 139.

2- Extended Data 2A shows a range of expression between 0-400 in both level in Stage I/II and III/IV disease. The groups are highly powered and yield statistical significance but the means of the two groups are only slightly different and the data

is interpreted to infer correlation between Npm1 expression and disease progression which does not seem justified based on this data. Extended Data 2B-C split the cohort further to look at survival but the criteria, cutoff and rationale for how the data was split to generate these survival charts is not provided.

Reply: The analysis of NPM1 expression in Stage I/II and III/IV COAD (Extended Data 2B) was based on data from 425 COAD patients in TCGA, and although the differences were not particularly large, they were statistically significant ($p=0.0246$). It is common that the mRNA expression changes of critical transcription factors are usually small between physiological states and disease conditions, because a small change in critical transcription factors is enough to bring upon several folds change of their downstream genes. Moreover, a small change in mRNA expression levels may lead to a large difference in protein levels due to post-transcriptional regulation. For survival analysis, 270 COAD patients and 92 READ patients in TCGA were used, cutoff by the median expression of NPM1 in groups, 95% Confidence Interval. It is worth noting that, not all COAD patients have included survival data, so the patients in Extended Data 2C and 2D are not exactly the same. We have supplemented and highlighted the details of data analysis in the revised figure legend of Extended Fig.2C-2E.

3- Much of the analysis focuses on DSS only and does not provide naïve controls as important comparators to assess the magnitude of changes shown. This is important as often control animals appear to show surprisingly limited inflammation as measured by weight gain or myeloid cellular infiltrate. While it is understood this is worsened in Npm1 deficient animals, DSS would normally be expected to induce significant inflammation and disease in WT mice too, so comparisons with controls are required to provide full context.

Similarly, the effects on ILC3 frequencies and functions throughout are only shown in DSS colitis, making it unclear to what extent changes seen are present at steady state and thus predisposing to worse outcome in colitis, or only apparent in an inflammatory context.

Reply: Thank you very much for your suggestion. The untreated WT and Npm1^{+/-} mice or Rorc^{cre/+} Npm1^{flox/flox} mice have been added to each experiment as naïve controls. We analyzed the change of macrophage, neutrophils, DC and eosinophils by flow cytometry in Npm1^{+/-} mice and Rorc^{cre/+}Npm1^{flox/flox} mice in steady and DSS state. Under physiological conditions, there were no significant differences between the two groups (controls and genetic modified mice) of mice (Extended Data Fig.5A-5D, Extended Data Fig.8A-8D). However, under inflammatory conditions, there was a significant increase in the infiltration of macrophages, neutrophils, eosinophils, and DCs in the intestinal tissues of Npm1^{+/+} mice, and this infiltration was more pronounced in Npm1^{+/-} mice. (Extended Data Fig.5E-5H). Similar results were obtained in Rorc^{cre/+}Npm1^{flox/flox} mice (Extended Data Fig.8E-8H). We have added the description in the revised manuscript, from lines 164 to 167 and lines 214 to 216.

Intestinal ILC3 frequencies in WT and Npm1^{+/-} mice in steady state were detected by flow cytometry and no significant difference between two genotypic mice was found (Extended Data Fig.6E-6H). The differences in the proportion and function of ILC3 cells were only evident under inflammatory conditions (Fig.2B-2E). Similar results were obtained in Rorc^{cre/+}Npm1^{flox/flox} mice (Fig.2M-2P, Extended Data Fig.7D-7G). We have added the description in the revised manuscript, from lines 180 to 181 and lines 200 to 201.

The revisions include the following:

• The ratio of macrophages, neutrophils, eosinophils and DC infiltrated in colon exhibited few changes between WT and Npm1^{+/-} mice in steady state (Extended Data Fig.5A-5D). However, in DSS-induced colitis, an elevation of these cells was observed in Npm1^{+/-} mice compared to WT mice (Extended Data Fig.5E-5H) from lines 164 to 167.

• Additionally, consistent with changes observed in Npm1^{+/-} mice, the increased infiltration of myeloid cells also existed in Rorc^{cre/+}Npm1^{flox/flox} mice under pathological condition (Extended Data Fig.8A-8H) from lines 214 to 216. However, these changes were not observed under physiological condition (Extended Data Fig.6E-6H) from lines 180 to 181.

• Frequencies of intestinal ILC3 and IL22⁺ ILC3 in Rorc^{cre/+}Npm1^{flox/flox} mice were also

comparable with that of control group (Extended Data Fig.7D-7G) from lines 200 to 201.

Technical concerns about the analysis, sorting and attributed roles of ILC3:
4- For key experiments ILC3 are identified and sorted as CD45 low CD90 high. This is not sufficiently robust or appropriate to provide the purity required for the studies downstream and falls below the current state of the art in the ILC field. Moreover, the authors provide no evidence that such a strategy indeed yields a pure ILC3 population at steady state or during DSS. Finally the gating strategy for such sorts, shown in Figure 3A appears highly subjective with no clear subpopulation visible. Given the high likelihood of contamination in an inflamed environment significant data is needed to demonstrate this strategy yields pure ILC3, otherwise key assays need to be repeated with an improved gating and sorting strategy in line with key studies in the ILC3 field.

Reply: Sorry for our mistake in the description of ILC3 sorting strategy. Indeed, the live+Lin-CD45lowCD90high cells were sorted as ILC3s. We have added the representative sorting strategy of ILC3 in new Figure 3A. We have confirmed a positive ratio of ROR γ T+ cells in sorted live+Lin-CD45lowCD90high cells, with a cell purity reaching 91% (new Figure 3A). The gating strategy picture in Figure 3A with no clear subpopulation visible was caused by size and line thickness. We have corrected the description in the revised manuscript in lines 232-234.

The revisions include the following:

âTo uncover mechanisms by which NPM1 regulates ILC3 expansion and function, we performed RNA sequencing (smart-seq2) of Live+Lin-CD45lowCD90high LPLs from colon of WT and Npm1 +/- mice with colitis induced by DSS treatment (Fig.3A) from lines 232 to 234.

5- For analysis ILC3 are identified as Lin neg ROR γ T+ which is more appropriate, but no further resolution is provided. Given ILC3 in the gut consist of multiple distinct subsets with very different biology and transcriptional signatures it is important to provide further detail on the relative effects of Npm1 manipulation on NCR+ vs. CCR6+ ILC3 subsets, and inclusion of other canonical ILC markers e.g. CD127 should be considered to ensure accuracy of analysis and interpretation.

Reply: Thanks for the suggestion. We further analyzed the NCR+ and CCR6+ subsets of ILC3s in both Npm1 +/- mice and Rorc γ cre/+Npm1 flox/flox under physiological conditions and pathological conditions (Extended Data Fig.6I, 6J, 7T and 7U). There were no significant differences observed in the proportions of these two ILC3 subsets between Npm1 +/- and Npm1 +/- mice under physiological or pathological conditions. Additionally, compared to untreated mice, NCR+ ILC3s were significantly elevated and CCR6+ ILC3s were reduced in colitis mice (Extended Data Fig.6I, 6J). Furthermore, the results slightly differed between Rorc γ cre/+Npm1 flox/flox mice and Npm1 flox/flox mice. Under physiological conditions, the proportion of CCR6+ subsets in Rorc γ cre/+Npm1 flox/flox mice was lower than that in Npm1 flox/flox mice. However, following DSS induction, the proportion of CCR6+ ILC3s in Rorc γ cre/+Npm1 flox/flox mice was higher than that in Npm1 flox/flox mice. There was no difference in the proportion of NCR+ ILC3 cells between the two mouse strains (Extended Data Fig.7T, 7U). The description of these results have been added to lines 181-183 and lines 210-212. Note that CD127 have been considered in the analysis (Extended Data Fig.3D) to ensure the accuracy of analysis and interpretation (Almost all the CD45+Lin-ROR γ T+ cells are CD127 positive).

The revisions include the following:

âFurther analysis revealed that there were no evident alterations in proportions of NCR+ ILC3 and CCR6+ ILC3 between WT and Npm1 +/- mice under physiological or pathological conditions (Extended Data Fig.6I, 6J) from lines 181 to 183.

âProportion of CCR6+ ILC3 in total ILC3s was higher in Rorc γ cre/+Npm1 flox/flox mice than that in Npm1 flox/flox mice under pathological condition, which was opposite in steady state (Extended Data Fig.7T,7U) from lines 210 to 212.

6- RORc Cre will also delete Npm1 in conventional T cells and gdT cells but these cells are not analysed in the Npm1 flox model in any meaningful way. Despite this, the authors attribute all phenotypes to correlative changes in ILC3 and ILC3-intrinsic biology. Detailed analysis of intestinal CD4+ T cells subsets (Th17, Treg) and gdT cells is needed both in the Npm1 +/- model and the conditional flox model and other possible immune dysregulation should be considered as potential contributing or causative factors in altered responses of DSS-driven inflammation.

Reply: Thank you for your suggestions. We analyzed the change of Th17, Treg and $\gamma\delta$ T cells in Npm1 +/- and Rorc^{cre}/+Npm1flox/flox mice using flow cytometry. The proportions of these cell subsets showed no significant differences between Npm1 +/- and Npm1 +/+ mice under physiological or pathological conditions (Extended Data Fig.5N-5S). Deletion of CD3+ T cells indicated that exacerbated colitis in Npm1 +/- mice was unlikely related to T cells (Extended Data Fig.5T-5X). Similarly, the percentages of Th17, Treg and $\gamma\delta$ T in the colon of Rorc^{cre}/+Npm1flox/flox and Npm1flox/flox mice were similar in physiological state and DSS colitis. Notably there was a slight reduction in the proportion of Treg in the intestines of Rorc^{cre}/+Npm1flox/flox mice in the physiological state. To exclude the contribution of T cells, we performed DSS model in Rorc^{cre}/+Npm1flox/flox mice after T cell depletion by anti-CD3 antibody. The results showed that enteritis was also exacerbated in Rorc^{cre}/+Npm1flox/flox mice compared to Npm1flox/flox mice after T cell clearance, suggesting that the regulation of enteritis by Npm1 is mainly dependent on ILC3s, rather than T cells (Extended Data Fig.8I-8Q). We have modified the description in lines 171-174 and line 216-219 in revised manuscript.

The revisions include the following:

âLikewise, evaluation of T cells (Th17, Treg and $\gamma\delta$ T cells) coupled with comparable colitis in two genotype mice after deletion of CD3+ T cells indicated that exacerbated colitis in Npm1-haploinsufficient mice was unlikely related to T cells (Extended Data Fig.3C, 5N-5X)â from lines 171 to 174.

âSince RORc-Cre will also delete Npm1 in conventional T cells and $\gamma\delta$ T cells, we examined the function of various T cell subsets and excluded their contributions to exacerbated colitis in Rorc^{cre}/+Npm1flox/flox mice by depleting T cells using a CD3 antibody (Extended Data Fig.8I-8Q)â from lines 216 to 219.

7- The reduced ILC3 frequencies in global and conditional deficient animals are interpreted as cell intrinsic, however these changes are only reported in DSS, not steady state. In DSS colitis the authors report marked increases in myeloid cells as would be expected (Extended Data 5) thus it is highly plausible that any perceived reduction in the relative frequency of ILC3 amongst CD45+ cells could be explained by significant increases in infiltrating cell types. Thus, cell numbers must be shown for all quantification to support the conclusion that ILC3 are reduced.

Reply: Thank you for the constructive suggestion. We recorded data on the number of cells in our experiments, which have been provided in the revised version of the manuscript (Figure 2D, 2E, 2O, 2P, 7I, 7J and Extended Data Fig.6C, 6D, 6G, 6H, 7F, 7G, 7P, 7Q). The results of absolute cell numbers and cell proportions were consistent. In brief, Npm1 +/- and Rorc^{cre}/+Npm1flox/flox mice both exhibited decreased number of ILC3 and IL22+ILC3s in DSS-induced colitis, compared to control group. We appreciate your suggestions, which helped us present our data in a more rigor way.

8- ILC3 have been extensively shown to be plastic, and $\alpha\text{ex-ILC3}$ with an ILC1-like phenotype may produce IFN- γ to exacerbate inflammation. What are the frequencies and effector status of these cells in RORc Cre x Npm1 flox mice in DSS colitis?

Reply: Thank you for your suggestions. We further analyzed the ratio of T-bet+ ILC3

subset and T-bet+IFN γ + ILC3 subset in total ILC3s in Rorc $^{cre/+}$ Npm1 $^{flox/flox}$ under physiological or pathological conditions. Results indicated that there were no significant differences in the proportion of IFN- γ producing ex-ILC3 between these two groups of mice with or without DSS administration (Extended Data Fig.7V-7Y). We have added the description in the revised manuscript, from lines 212 to 214.

The revisions include the following:

âThe proportion of IFN- γ producing ex-ILC3 was also unchanged between these two group of mice with or without DSS administration either (Extended Data Fig.7V-7Y)â from lines 212 to 214

9- A key piece of data presented are the changes in mitochondrial genes in âILC3â sorted from Npm1 $^{+/-}$ mice (Figure 3C+D). However, these genes are not shown to be restored by Npm1 overexpression, or similarly defective in studies targeting Tfam. More needs to be done to connect the different models and link data sets.

Reply: Thank you for your constructive suggestions. We detected the expression of mitochondrial genes in ILC3s sorted from Npm1 overexpressed mice (Npm1UTR $^{-/-}$). As anticipated, mitochondrial genes exhibited increased expression in ILC3, along with up-regulation of Npm1 expression (Extended Data Fig.10M). Furthermore, overexpression of Tfam in MNK3 markedly enhanced the expression of mtDNA including mt-Nd1, mt-Nd2, mt-Nd3, mt-Nd4 and mt-Atp6 (Extended Data Fig.10J). We have added the description in the revised manuscript, from lines 371 to 372, lines 414 to 415.

The revisions include the following:

âOverexpression of Tfam in MNK3 markedly enhanced the expression of mtDNA including mt-Nd1, mt-Nd2, mt-Nd3, mt-Nd4 and mt-Atp6 (Extended Data Fig.10J)â from lines 371 to 372.

âExpression of various mtDNA were upregulated in Npm1UTR $^{-/-}$ ILC3 compared to control group (Extended Data Fig.10M)âfrom lines 414 to 415.

10- The authors conclude âlack of Npm1 impairs mitochondrial OXPHOSâ but Npm1 $^{+/-}$ ILC3 actually have increased spare respiratory capacity â suggesting the capacity to engage OXPHOS is not impaired, and possibly even enhanced. Importantly as all assays are done in DSS colitis the cellular environment from which these two cell populations were isolated is dramatically different, and those from Npm1 $^{+/-}$ (which are more inflamed) have likely been exposed to significantly different cues in vivo that could alter their mitochondrial profile. A more detailed metabolic analysis of Npm1 $^{+/-}$ or conditionally deficient ILC3 isolated from naïve animals is required to determine whether these metabolic changes are truly cell-intrinsic or rather the subject of extrinsic signals during inflammation.

Reply: Thank you for your constructive suggestions. The spare respiratory capacity increased in Npm1 $^{+/-}$ ILC3, however it may be infected by the compensatory alterations in Npm1 $^{+/-}$ ILC3 in process of seahorse mitochondrial stress test. As mt-Atp6 and mt-Atp8 was down-regulated in Npm1 $^{+/-}$ ILC3s (Figure 3C-3E), which indicated that the function of mitochondrial complex V was impaired. In the first step of Seahorse test, oligomycin was used to inhibit the function of mitochondrial complex V, which resulted in the dual inhibition of complex V in Npm1 $^{+/-}$ ILC3s and probably led to a compensatory change in cells when FCCP was administered, and thus exhibited an increase in spare respiratory capacity. Nonetheless, additional evidence, including mtDNA level, membrane potential, basal OCR, maximal respiration and ATP production, indicated that Npm1 $^{+/-}$ mitochondria exhibited impaired function. Furthermore, we performed mitochondrial stress tests on sorted ILC3s from Npm1 $^{+/-}$ mice and WT mice under steady state (Extended Data Fig.9D-9G). However, the mitochondrial function in Npm1 $^{+/-}$ ILC3s was not impaired under physiological condition, suggesting that haploinsufficient NPM1 is enough to maintain mitochondrial function in ILC3s in steady status. These results have been added and highlighted in the revised version of the manuscript in lines 268-270. We also complemented the metabolic differences between shNC and shNpm1 in MNK3 cells with or without IL-1 β and IL-23 treatment. Control cells underwent a significant enhancement of ATP

production, maximal respiration and basal OCR after IL-1 β and IL-23 treatment, whereas metabolic activation of shNpm1 MNK3 cells was inhibited. These new results support that lack of Npm1 impairs mitochondrial OXPHOS, and have been added in Extended Data Fig.10A-10D and highlighted in the revised version of the manuscript in lines 308-309. Moreover, we found that ILC3 cellular oxidative phosphorylation levels were markedly elevated during the acute phase, compared to physiological conditions (Extended Data Figure 9H-9K). However, due to the internal decrease of Npm1, the metabolic enhancement in ILC3 is inhibited (Figure 4A-4D).

The revisions include the following:

âHowever, such impaired mitochondrial function in Npm1 \pm ILC3s was not observed under physiological condition (Extended Data Fig.9D-9G)â from lines 268 to 270.

âMNK3 also exhibited mitochondrial activation after IL-1 β /IL-23 stimulation, which was regulated by NPM1 (Extended Data Fig.10A-10D)â from lines 308 to 309.
11- Bezafibrate treatment was done in Npm1 \pm mice. Why was similar not performed in the RORc Cre x Npm1 flox model where the deficiency is at least more restrictive. Such drug treatments in a globally deficient animal are limited when attributing effects to ILC3 in the interpretation of the data.

Reply: Thank you for your feedback. The bezafibrate treatment experiment was conducted in Rorc Δ cre/+Npm1 flox/flox mice. The results were similar to those obtained in Npm1 \pm mice, indicating that bezafibrate can alleviate colitis in Rorc Δ cre/+Npm1 flox/flox mice too (Extended Data Fig.9W-9Y). The results of the animal experiments have been added to the revised version of the manuscript in lines 301-303.

The revisions include the following:

âSimilarly, bezafibrate succeeded in reversing the colitis in Rorc Δ cre/+Npm1 flox/flox mice (Extended Data Fig.9W-9Y)â from lines 301 to 303.

12- Figure 5G; the p65-regulated genes measured have not been shown to be expressed by ILC3 to my knowledge - and no data is provided to validate these genes are truly expressed (i.e. analysis of prior transcriptional data sets etc.). Il12a, Il6, Cxcl2, Ccl4 would all typically be associated with myeloid-cells. Given the potential for contamination in sort-purified cells (see above), significant extra supporting data is required here if utilizing this gene signature as proof of Npm1-p65 interactions in regulating ILC3-intrinsic transcription.

Reply: Thanks for your question. We checked the expression level of CCL4, CXCL2, IL12A and IL6 in human colonic scRNA-seq data (GSE182270) and found that ILC3 typically do not express IL12A and IL6. We also examined their expression in mouse ILC3 (macrophages were used as positive control). Results indicated that the expression level of Il6 and Il12a in ILC3 is 5%-10% compared with that in macrophages (See below Fig. 3). Given the positive ratio of ROR γ T γ cells in sorted Live+Lin-CD45 $^{\text{low}}$ CD90 $^{\text{high}}$ cells was about 90%, we are not sure if ILC3 expressed a lower level of Il6 and Il12a or not. To avoid misinformation, we have removed Il6 and Il12a data from Figure 5G and manuscript.

Fig. 3 Expression analysis of indicated genes in human colon infiltrated cells in GSE182270 (A) and mice macrophages and ILC3s (B).

13- Moreover, in Figures 5 and 6 the analysis of p65 and Tfam and their links to mitochondrial regulation largely rely on prior data in other systems and assumptions that the same is occurring here without direct evidence to support that in ILC3. Does manipulation of p65 and Tfam similarly change mitochondrial metabolism genes, extracellular flux capacity, and ILC3 cytokine production? What is the evidence these TFs regulate ILC3-associated effector genes or mitochondrial

metabolism programs directly? The links between these data sets and core findings are under developed.

Reply: In order to establish a direct relationship between Tfam and p65 and mitochondrial metabolism regulation in ILC3 cells, we generated MNK3 cell lines with knocked-down Tfam or p65 using shRNA-expressing lentivirus. We then analyzed oxidative phosphorylation and cytokine expression levels. Our results showed that compared to the control group, Knock-down of Tfam significantly repressed basal OCR, maximal respiration, and ATP production in MNK3 cells (Figure.6D-6G). Similarly, MNK3 cells with p65 knockdown showed a decrease in basal OCR (Figure.5J-5M). Following stimulation with IL-1 β and IL-23, IL-22 expression was upregulated in MNK3 cells. However, knockdown of Tfam and p65 both resulted in downregulation of IL-22 expression (Figure.5N, 6H). These data have been added to Figure 5J-5N, 6D-6H. We have modified the description in lines 352-355 and 373-374.

The revisions include the following:

âIn vitro tests of MNK3 cells with p65-knockdown exhibited a decrease in OCR, especially basal OCR (Fig.5J-5M). More importantly, knockdown of p65 resulted in downregulation of IL22 expression after stimulation (Fig.5N)â from lines 352 to 355.

âKnockdown of Tfam in MNK3 notably impaired its mitochondrial function and attenuated the production of IL22 (Fig.6D-6H)â from lines 373 to 374.

14- Given ILC3 appear to have less mitochondria (Figure 4H) and lower cell frequencies (Figure 2A) in absence of Npm1, can the authors rule out cell death/apoptosis as a reason for reduced IL-22 in colitis and other metabolic and transcriptional phenotypes?

Reply: We analyzed the apoptosis of ILC3 in Rorc^{cre}/+Npm1^{flox/flox} mice under physiological or pathological conditions. Results indicated that there were no significant difference in the ratio of apoptotic ILC3 with or without NPM1 deficiency under physiological condition (Extended Data Fig. 7R). However, after DSS-induced colitis, the percentage of apoptotic ILC3s were increased in Rorc^{cre}/+Npm1^{flox/flox} mice compared to control group (Extended Data Fig. 7S). However, the proportion of decrease in IL22+ILC3 after Npm1 deletion is higher than that of increase in apoptotic cells, indicating that apoptosis of ILC3 and decrease of IL-22 expression are reasons for decrease of IL-22 in colitis. We have modified the description in lines 209-210.

The revisions include the following:

âFurthermore, the percentage of apoptotic ILC3s were increased in Rorc^{cre}/+Npm1^{flox/flox} mice in DSS-induced colitis (Extended Data Fig.7R,7S)â from lines 209 to 210.

15 Methodology: In a number of key data sets the details of the experiments are insufficiently described and/or methodology used is not described anywhere in the manuscript. Figure legends and methods must be sufficient to allow for a detailed understanding of what was done and how, and this should be checked throughout. Some key specific examples include:

o Figure 1A; what is indicated by low/medium/high in regards to Npm1 staining. What cutoffs were used and how were they justified?

Reply: The semi quantitative immunohistochemistry is generally divided into three levels: Low (+), medium (++), and high (+++). Low (+)=1, medium (++)=2, high (+++)=3. Then calculate the value based on (+)%*1+(++)%*2+(+++)%*3; The final score is (+) for value less than 1.0, (++) for value between 1.0 and 1.5, and (+++) for value greater than 1.5. We have added it in the method part from lines 877 to 880.

The revisions include the following:

The semi quantitative immunohistochemistry is generally divided into three levels: Low (+), medium (++), and high (+++). Low (+)=1, medium (++)=2, high (+++)=3. Then calculate the value based on (+)%*1+(++)%*2+(+++)%*3; The final score is (+) for value less than 1.0, (++) for value between 1.0 and 1.5, and (+++) for value greater than 1.5. In lines 877 to 880.

Extended Data 2B-C and 5I, authors mention data was split into Npm1 high and low but no cut-off values on expression are given or rationale for how this was justified. Given data spread in Extended Data 2A in Stage I/II patients is wide it is unclear if this is simply arbitrary.

Reply: To validate the prognosis value of NPM1 in COAD and READ patients, NPM1 high or low was cutoff by median expression of NPM1 in groups (Extended Data Fig. 2D, 2E). We have modified the descriptions in figure legend in lines 1152-1155. Meanwhile, in the revised figure 5I, NPM1low ILC3s and NPM1high ILC3s were identified using a median expression cutoff for NPM1 in ILC3s of UC patients. We have modified the descriptions in figure legend from lines 615-617.

DSS experiments, DAI used throughout as key output but not clearly defined anywhere.

Reply: The Disease Activity Index (DAI) is calculated by combining three parameters: the percentage weight loss of the mice, the consistency of stool, and the presence of stool blood. The scoring for each parameter is as follows: (1) Weight loss: 0 points if weight remains stable, 1 point for a 1-5% weight loss, 2 points for a 5-10% weight loss, 3 points for a 10-15% weight loss, and 4 points for a weight loss greater than 15%. (2) Stool consistency: 0 points for normal stool, 2 points for loose stool, and 4 points for diarrhea. (3) Stool blood: 0 points for no blood, 2 points for occult blood positivity, and 4 points for overt bleeding. $DAI = (\text{Weight loss index} + \text{Stool consistency} + \text{Blood in stool}) / 3$. We have added it in the method part from lines 720 to 728.

The revisions include the following:

The Disease Activity Index (DAI) is calculated by combining three parameters: the percentage weight loss of the mice, the consistency of stool, and the presence of stool blood. The scoring for each parameter is as follows: (1) Weight loss: 0 points if weight remains stable, 1 point for a 1-5% weight loss, 2 points for a 5-10% weight loss, 3 points for a 10-15% weight loss, and 4 points for a weight loss greater than 15%. (2) Stool consistency: 0 points for normal stool, 2 points for loose stool, and 4 points for diarrhea. (3) Stool blood: 0 points for no blood, 2 points for occult blood positivity, and 4 points for overt bleeding. $DAI = (\text{Weight loss index} + \text{Stool consistency} + \text{Blood in stool}) / 3$. Note that mice are generally genotyped and caged at 3 to 4 weeks of age, and then placed in separate cages when grouping, according to its genotype. In lines 720-728.

Npm1-deficiency in Extended Data Figure 2D was checked at protein level, but not at RNA level. Moreover, in the aforementioned blot no controls are provided (e.g. isotype antibodies, or known negative controls). The methodology used here and in other blots is also not described anywhere in the methods.

Reply: Thank you for your suggestions. In revised extended data Fig. 2H, we showed the decreased expression of Npm1 in whole colon of Npm1-haploinsufficient mice at RNA level compared to wildtype mice. We also added the methodology used in IP and Western Blot in the method part from lines 795 to 808.

Immunoprecipitation and Western Blot Analysis: To perform immunoprecipitation, cells were lysed in a IP lysis buffer containing 20 mmol/L Tris (pH 7.5), 150 mmol/L NaCl, and 1% Triton X-100, supplemented with a cocktail of protease and phosphatase inhibitors. Following lysis, the supernatants were collected after centrifugation and incubated overnight at 4°C with constant rotation with the indicated antibodies. The antibody-antigen complexes were then precipitated using

protein A/G magnetic beads (Millipore) and washed with PBS. For western blot analysis, cell lysates were prepared using RIPA lysis buffer (CoWin Biosciences, China) containing protease inhibitors and phosphatase inhibitors (CoWin Biosciences, China). Equal amounts of protein were loaded onto SDS-PAGE gels and transferred to nitrocellulose membranes. The membranes were blocked with 5% non-fat dried milk for 1 hour at room temperature before being incubated with primary antibodies overnight at 4°C. After washing, the membranes were incubated with IRDye 800cw or 680cw conjugated secondary antibodies (1:10,000 dilution) for 1 hour. The membranes were then imaged using an Odyssey® CLx Infrared Imaging System from lines 795 to 808.

o Figure 3G; single cell data was used to identify DEG in Npm1^{high} and Npm1^{low} cells. Npm1 is almost uniformly expressed in this cell cluster (Extended Data 6) and single cell RNA data is not quantitative due to dropout – thus lack of mRNA of an individual cell within a given cluster cannot necessarily be indicative of lacking expression. Either way no details or rationale are provided for what was considered high or low or whether this was purely arbitrary.

Reply: We are sorry for not describing the process of scRNA-seq data analysis. scRNA-seq datasets GSE182270 was downloaded from the GEO database, which was performed on cells extracted from colonic biopsies of inflamed mucosa (ulcerative colitis patients, n=5) and normal colonic mucosa (healthy controls, n=4). Count tables were analyzed used Seurat 4.0 package following the standard workflow with default settings. The number of principal components (PCs) was determined based on Elbow plots, PCs= 13. Next, FindNeighbors and FindClusters functions were used for cell clustering, UMAP method was used for visualization. cell type-specific markers found by FindMarkers function; cell type identities manually annotated by matching cluster-specific upregulated marker genes with cell-type markers in CellMarker 2.0 database. NPM1^{low} ILC3s and NPM1^{high} ILC3s were identified using a median expression cutoff for NPM1 in ILC3s. Note that cells dropout of NPM1 was not included in analysis. FindMarkers function was used to identify significantly regulated genes in NPM1^{high} ILC3. ClusterProfiler package was applied for functional annotation.

In our scRNA-seq analysis, cells dropout of NPM1 was not included in analysis. Thus these cells could not influence the identification of Npm1^{high} and Npm1^{low} cells. Moreover, our RNA-seq analysis of isolated primary ILC3s showed that OXPHOS was down-regulated in Npm1[±] ILC3s. The scRNA-seq analysis in figure 5I was just for verification.

The revisions include the following:

• scRNA-seq datasets GSE182270 was downloaded from the GEO database, which was performed on cells extracted from colonic biopsies of inflamed mucosa (ulcerative colitis patients, n=5) and normal colonic mucosa (healthy controls, n=4). Count tables were analyzed used Seurat 4.0 package following the standard workflow with default settings. The number of principal components (PCs) was determined based on Elbow plots, PCs= 13. Next, FindNeighbors and FindClusters functions were used for cell clustering, UMAP method was used for visualization. cell type-specific markers found by FindMarkers function; cell type identities manually annotated by matching cluster-specific upregulated marker genes with cell-type markers in CellMarker 2.0 database. NPM1^{low} ILC3s and NPM1^{high} ILC3s were identified using a median expression cutoff for NPM1 in ILC3s. Note that cells dropout of NPM1 was not included in analysis. FindMarkers function was used to identify significantly regulated genes in NPM1^{high} ILC3. ClusterProfiler package was applied for functional annotation from lines 781 to 793.

o Mitochondrial potential assay in Figure 4E is not described anywhere in methods. Representative data (microscopy or flow cytometry) should be provided and further supported with other measures of mitochondrial mass and potential (e.g. TMRE).

Reply: Thank you for your suggestions. We have detected the mitochondrial membrane potential of ILC3s by TMRE (Figure 4F, Extended Data Fig.9V) and got similar results with JC-1 assay (Figure 4E, Extended Data Fig.9U): the mitochondrial membrane potential of Npm1[±] ILC3s was reduced significantly compared with that of WT ILC3s only under pathological condition.

We are sorry for not describing the method of mitochondrial potential assay. We

have added Primary ILC3s were loaded with the JC-1 primer (Beyotime, C2006) and potentiometric dye TMRE (Beyotime, C2001S) at 37°C for 20 min and washed with buffer or cell medium for 3 times. Fluorescence were measured using microplate reader. When detecting JC-1 monomers, the excitation light can be set to 490nm and the emission light can be set to 530nm; When detecting JC-1 polymer, the excitation light can be set to 525nm and the emission light can be set to 590nm. The maximum excitation wavelength of TMRE is 550nm, and the maximum emission wavelength is 575nm. To revised manuscript from lines 861 to 868.

o Validation of successful flox-mediated deletion of targeted Npm1 exon in ILC3 is not provided in RORc Cre x Npm1 flox mice.

Reply: We have validated the deletion of exon 2 to exon 6 of Npm1 (2663bp deletion) in ILC3s of Rorccre/+Npm1flox/flox mice and added it to extended data fig.1F of revised manuscript.

Minor points:

16- Extended Data 6 and Figure 3I; t-SNE clusters need to be identified and labelled. Numbering of clusters alone is not sufficiently informative and cell types should be clearly indicated.

Reply: Thank you for your suggestions. The clusters have been identified and labelled in the revised figure 3F and extended data fig.2A

Reviewer #3

(Remarks to the Author)

In this study, NMP1 is identified as an important factor in the control of intestinal inflammation and mitochondrial respiration. NMP1 levels are observed to be lower in the colon of IBD and mice lacking an Nmp1 allele exhibit exacerbated colonic inflammation and colitis-associated tumor formation. Bone marrow transplant experiments identify the hemopoietic compartment as the origin of this enhanced colitis and the authors identify a reduced frequency of ILC3s as a possible driving factor. Indeed, Rorc-driven deletion of Nmp1 recapitulated much of the macrophenotypes observed in Nmp1^{+/-} mice. Transcriptomic analysis of Nmp1-deficient ILC3s revealed significant perturbations in mitochondrial gene expression and Nmp1 loss reduced mitochondrial load and impaired OXPHOS in ILC3s. Mechanistically, NMP1 was found to interact with p65 to control Tfam expression, a regulator of mitochondrial transcription. This is an elegant study revealing a novel pathway important in the regulation of ILC3 biology, inflammation, and mitochondrial biology. The manuscript is well written and clear to follow

The authors may consider the following comments:

1. Lines 138/139: Despite the text claiming the contrary, there are significant differences in the infiltration of neutrophils, macrophages and eosinophils in Nmp1^{+/-}. This also adds to a lack of clarity with regard to the rationale for targeting ILC3s, despite the very convincing subsequent data. What is the contribution of these myeloid cells to exacerbated colitis in NMP1^{+/-} mice? Is the increased flux of myeloid cells into the colon lost in NPM1 flox Rorc Cre mice?

Reply: Thank you for your appreciation of our work and constructive feedback. The ratio of macrophages, neutrophils, eosinophils and DC infiltrated in colon exhibited few changes between WT and Nmp1^{+/-} mice in steady state. However, in DSS-induced colitis, an elevation of these cells was observed in Nmp1^{+/-} mice compared to WT mice (Extended Data Fig.5A-5H). Furthermore, clearance of CD11b⁺ myeloid cells failed to rescue the exacerbated enteritis in Nmp1^{+/-} mice, suggesting that NPM1 in myeloid cells was insufficient to regulate intestinal inflammation (Extended Data Fig.5I-5M). Thus, our data suggest that the infiltration of large numbers of myeloid cells may be a consequence rather than a cause of the exacerbation of colitis. We have modified the description in lines 164-171. In addition, increased infiltration of myeloid cells into the colon were also observed in Rorccre/+Npm1flox/flox mice after DSS-induced colitis (Extended Data Fig.8A-8H). We have added the description in lines 214-216.

The revisions include the following:

âThe ratio of macrophages, neutrophils, eosinophils and DC infiltrated in colon exhibited few changes between WT and Npm1 +/- mice in steady state (Extended Data Fig.5A-5D). However, in DSS-induced colitis, an elevation of these cells was observed in Npm1 +/- mice compared to WT mice (Extended Data Fig.5E-5H). It's known infiltration of myeloid cells into the intestinal lamina propria is considered a common cause of progressive colitis³⁸. Furthermore, clearance of CD11b+ myeloid cells failed to rescue the exacerbated enteritis in Npm1 +/- mice, suggesting that NPM1 in myeloid cells was insufficient to regulate intestinal inflammation (Extended Data Fig.5I-5M)â from lines 164 to 171.

âAdditionally, consistent with changes observed in Npm1 +/- mice, the increased infiltration of myeloid cells also existed in Rorc^{cre}+/Npm1^{flox/flox} mice under pathological condition (Extended Data Fig.8A-8H)â from lines 214 to 216.

2âç This is linked to a further question with regard to the specificity of mitochondrial regulation by NPM1, is this an ILC3 specific phenomenon or is this a more widespread mechanism in other cell types?

Reply: To further clarify the specificity of mitochondrial regulation by NPM1, we sorted macrophage, T cell and epithelial cells from the intestines of mice in physiological and inflammatory conditions and performed mitochondrial stress tests. The results showed that deletion of Npm1 did not have a significant effect on the oxidative phosphorylation of these cells (Extended Data Fig.9L-9T). mtDNA levels in macrophage, T cell and epithelial cells were also detected by qPCR. The vast majority of these mtDNAs showed no significant changes (Extended Data Fig.9A-9C). The above data suggest that Npm1 has a significant effect on ILC3 on mitochondrial function and metabolism, while having little effect on other cell types. We have added the description in lines 240-242 and lines 268-270.

Our explanation is as follows: In our manuscript, we found TFAM as an important regulator of mitochondrial function under NPM1, further investigation showed that remarkable up-expression of Tfam only happened in ILC3s in response to DSS treatment, not in macrophages, T cells, and epithelial cells (revised Figure 6A-6C). However, Tfb1m and Tfb2m were significantly increased in macrophages, T cells, and epithelial cells, but not in ILC3s after DSS treatment (Fig.6A-C). Meanwhile, we analyzed the single cell RNA-seq data (GSE211578) of colon of DSS-treated and control mice from Dr. Zhang lab and found that ILC3s exhibit higher expression of Tfam but lower expression of Tfb1m and Tfb2m under DSS condition, compared to other cell types (see below Fig.1, note that this data set has not been published in author's article, so we could not show it in our manuscript). These data suggested that mitochondrial activation in ILC3s is primarily dependent on TFAM, rather than on TFB1M or TFB2M. Contrast to ILC3s, macrophages, T cells, and epithelial cells are primarily depend on TFB1M and/or TFB2M in DSS-treated mice. Therefore, although NPM1 expression is reduced in UC patients (Extended Data Fig.2B) and enteric mice (Extended Data Fig.4A), with no discernible cell-specific characteristics, the up-regulation of TFAM, downstream gene of Npm1, specifically required in ILC3 cells (Fig.6A-C, below Fig.1) explains why the insufficiency of NPM1 preferentially affects function of ILC3. Accordingly, NPM1 is crucial for the heightened demand of TFAM to subsequently increase mitochondrial function in ILC3s, not other cell types, during DSS-induced colitisâ

Fig.4 Single-cell analysis of Tfam, Tfb1m and Tfb2m in DSS-induced mice and Ctrl mice (GSE211578). Dot size represents the fraction of cells within the cluster that express each gene and colors indicate the z-scaled expression of genes in cells within each cluster

The revisions include the following:

âHowever, the universal decrease of mtDNA was not observed in epithelial cells, macrophages or T cells of Npm1 +/- mice (Extended Data Fig.9A-9C)â from line 240 to 242.

âHowever, such impaired mitochondrial function in Npm1 +/- ILC3s was not observed

under physiological condition (Extended Data Fig.9D-9G) from line 268 to 270.

The expression levels of the three mitochondrial transcription factors in ILC3s showed no differences between Npm1^{+/+} and Npm1^{+/-} mice in steady state. However, under pathological conditions, a remarkable decrease in Tfam expression was only observed in ILC3s, not macrophages, T cells, and epithelial cells, of Npm1 haploinsufficient mice when compared to Npm1^{+/+} mice (Fig.6A-C), suggesting that NPM1 plays an indispensable role in up-regulation of Tfam in ILC3s upon DSS-treatment. However, Tfb1m and Tfb2m were significantly increased in macrophages, T cells, and epithelial cells, but not in ILC3s after DSS treatment (Fig.6A-C). These data suggested that mitochondrial activation in ILC3s is primarily dependent on TFAM, rather than on TFB1M or TFB2M. Overexpression of Tfam in MNK3 markedly enhanced the expression of mtDNAs including mt-Nd1, mt-Nd2, mt-Nd3, mt-Nd4 and mt-Atp6 (Extended Data Fig.10J). Knockdown of Tfam in MNK3 notably impaired its mitochondrial function and attenuated the production of IL22 (Fig.6D-6H). Accordingly, NPM1 is crucial for the heightened demand of TFAM to subsequently increase mitochondrial function in ILC3s, not other cell types, during DSS-induced colitis from line 363 to 376.

3 Although deciphering exactly how mitochondria control ILC3 activation/maintenance/IL-22 production is perhaps beyond the scope of this study. More work needs to be done to flesh this part of the study out. Specifically:
3.1 Does ablation of other key components phenocopy Nmp1 loss on ILC3 activation and metabolism, such as Tfam and p65?

Reply: This is a good suggestion. We generated MNK3 cell lines with knocked-down Tfam or p65 using shRNA-expressing lentivirus. We then analyzed oxidative phosphorylation and cytokine expression levels. Our results showed that compared to the control group, MNK3 cells with shTfam exhibited significant suppression in basal OCR, maximal respiration, and ATP production (Figure.6D-6G). On the other hand, MNK3 cells with p65 knockdown showed a decrease in basal OCR (Figure.5J-5M). Following stimulation with IL-1 β and IL-23, IL-22 expression is upregulated in MNK3 cells. However, knockdown of Tfam and p65 both resulted in downregulation of IL-22 expression (Figure.5N, 6H). These data have been added to Figure 5J-5N, 6D-6H. We have modified the description in lines 352-355 and 373-374.

The revisions include the following:

In vitro tests of MNK3 cells with p65-knockdown exhibited a decrease in OCR, especially basal OCR (Fig.5J-5M). More importantly, knockdown of p65 resulted in downregulation of IL22 expression after stimulation (Extended Data Fig.5N) from line 352 to 355.

Knockdown of Tfam in MNK3 notably impaired its mitochondrial function and attenuated the production of IL22 (Fig.6D-6H) from line 373 to 374.

3.2 Whilst Tfam targets mitochondrial genes for transcription, its reduction can have wider consequences for mitochondria. To that end, it is important to understand more robustly which NMP1-driven mitochondrial perturbations result in altered ILC3 activation. Is it specifically OXPHOS and the running of the ETC? This can be probed with ETC complex inhibitors or knockdowns in the ILC3 cell line. Is the TCA cycle functioning normally in NMP1-deficient ILC3s? Can NMP1-deficient cells be rescued by targeting mitochondrial metabolism in other ways, such as via succinate, a substrate important for both the TCA cycle and the ETC?

Reply: We thank you for your valuable suggestions. In response, we conducted additional experiments on MNK3 cells. We found that treatment with ETC inhibitors (Oligomycin and Rotenone) significantly inhibits IL-22 expression in MNK3 cells. However, the difference in IL-22 expression between shNC and shNpm1 MNK3 after OXPHOS inhibitor administration indicated that NPM1 may participate in other biological process to sustain ILC3 activation (Extended Data Fig.10G,10H). Succinate supplementation partially rescued the IL-22 expression in

Npm1-knockdown MNK3 cells, which means TCA cycle is also impaired because of Npm1 deficiency (Extended Data Fig.10I). Based on these results, it appears that the impaired activation of Npm1-haploinsufficient ILC3 can be rescued by targeting mitochondrial metabolism. We have added the description of these additional results in lines 313-320 of revised manuscript.

The revisions include the following:

In contrast, OXPHOS inhibitor oligomycin and rotenone suppressed the activation of MNK3 (Extended Data Fig.10G,10H). However, the difference in Il22 expression between shNC and shNpm1 MNK3 after OXPHOS inhibitor administration indicated that NPM1 may participate in other biological process to sustain ILC3 activation (Extended Data Fig.10G,10H). The tricarboxylic acid (TCA) cycle, a crucial component of mitochondrial metabolism, is known to participate in the activation of immune cells. Since succinate is a substrate for the TCA cycle, its addition partially rescued the impaired ILC3 activation resulting from Npm1 heterozygous deletion (Extended Data Fig.10I) from line 313 to 320.

Minor points

There is a potential flow cytometry compensation issue in eosinophil plot in extended data figure 4 that if corrected may suggest the population immediately below the eosinophil gate is the true population.

Reply: Thank you for your suggestions. We have re-analyzed our flow cytometry and correct the compensation. The revised gating strategy was shown in Extended Data Fig.3B.

Some Y-axes not originating at 0 (e.g. Fig 2C and Fig. 4I).

Reply: Thank you for this suggestion. We have made the modification and all the Y-axes of figures were now originated at 0, except for the body weight.

Figure 1A: The scale bar size in the legend should be corrected to match the figure

Reply: Thank you for your suggestions. The scale bar size in the legend of figure 1A has been corrected.

Fig. 1G-HI: the legend should indicate more clearly the source of mRNA, is it whole colon or epithelia etc?

Reply: Thank you for your suggestions. The source of mRNA in Figure 1G-1H was from whole colon of mice. We have modified it the revised manuscript.

The revisions include the following:

RT-PCR analysis of mRNA abundance of Reg3b, Reg3g (G) and S100a8, S100a9 (H) in whole colon of mice at day 5 of administration of 2.5% DSS (n=4) in line 503.

Version 1:

Decision Letter:

Our ref: NI-A35998A

12th Jun 2024

Dear Dr. Sun,

Thank you for submitting your revised manuscript "Nucleophosmin 1 Promotes Mucosal Immunity by Supporting Mitochondrial Oxidative Phosphorylation and ILC3 Activity" (NI-A35998A). It has now been seen by the original referees and their comments are below. The reviewers find that the paper has improved in revision, and therefore we'll be happy in principle to publish it in Nature Immunology, pending minor revisions to satisfy the referees' final requests and to comply with our editorial and formatting guidelines.

We will now perform detailed checks on your paper and will send you a checklist detailing our editorial and formatting requirements in about a week. Please do not upload the final materials and make any revisions until you receive this additional information from us.

If you had not uploaded a Word file for the current version of the manuscript, we will need one before beginning the editing process; please email that to immunology@us.nature.com at your earliest convenience.

Thank you again for your interest in Nature Immunology Please do not hesitate to contact me if you have any questions.

Sincerely,

Nick Bernard, PhD
Senior Editor
Nature Immunology

Reviewer #1 (Remarks to the Author):

This is a revised version of this manuscript that is considerably improved. The authors have addressed virtually all of my major concerns, in particular regarding the contribution of T cells in the conditional deletion experiments with Rorc-Cre, the role of the microbiota, potential gene dosage effects and the exploration of additional colitis models. The new data considerably strengthen the manuscript and I have no major concerns. The authors should comment on/address one issue. They show changes in the microbiota in Npm1 +/- mice. Are these a consequence of the increased severity of inflammation or they an a priori driver of disease (ILC3s produce IL-22 also in steady-state). This could be clarified by microbiota transfer or rigorous crossfostering and cohousing of littermates.

Reviewer #2 (Remarks to the Author):

The authors have comprehensively addressed my concerns and comments as well as those from other reviewers. The degree of mechanistic depth and complementary data is extensive and I have no further comments.

Reviewer #3 (Remarks to the Author):

The authors have adequately addressed the points raised, I have no further comments

Version 2:

Decision Letter:

In reply please quote: NI-A35998B

Dear Dr. Sun,

I am delighted to accept your manuscript entitled "Nucleophosmin 1 promotes mucosal immunity by supporting mitochondrial oxidative phosphorylation and ILC3 activity" for publication in an upcoming issue of Nature Immunology.

Over the next few weeks, your paper will be copyedited to ensure that it conforms to Nature Immunology style. Once your paper is typeset, you will receive an email with a link to choose the appropriate publishing options for your paper and our Author Services team will be in touch regarding any additional information that may be required.

Acceptance is conditional on the data in the manuscript not being published elsewhere, or announced in the print or electronic media, until the embargo/publication date. These restrictions are not intended to deter you from presenting your

data at academic meetings and conferences, but any enquiries from the media about papers not yet scheduled for publication should be referred to us.

Please note that *Nature Immunology* is a Transformative Journal (TJ). Authors may publish their research with us through the traditional subscription access route or make their paper immediately open access through payment of an article-processing charge (APC). Authors will not be required to make a final decision about access to their article until it has been accepted. Find out more about Transformative Journals.

Authors may need to take specific actions to achieve compliance with funder and institutional open access mandates. If your research is supported by a funder that requires immediate open access (e.g. according to Plan S principles) then you should select the gold OA route, and we will direct you to the compliant route where possible. For authors selecting the subscription publication route, the journal's standard licensing terms will need to be accepted, including self-archiving policies. Those licensing terms will supersede any other terms that the author or any third party may assert apply to any version of the manuscript.

Your paper will be published online soon after we receive your corrections and will appear in print in the next available issue.

Also, if you have any spectacular or outstanding figures or graphics associated with your manuscript - though not necessarily included with your submission - we'd be delighted to consider them as candidates for our cover. Simply send an electronic version (accompanied by a hard copy) to us with a possible cover caption enclosed.

If you have not already done so, we strongly recommend that you upload the step-by-step protocols used in this manuscript to protocols.io. protocols.io is an open online resource that allows researchers to share their detailed experimental know-how. All uploaded protocols are made freely available and are assigned DOIs for ease of citation. Protocols can be linked to any publications in which they are used and will be linked to from your article. You can also establish a dedicated workspace to collect all your lab Protocols. By uploading your Protocols to protocols.io, you are enabling researchers to more readily reproduce or adapt the methodology you use, as well as increasing the visibility of your protocols and papers. Upload your Protocols at <https://protocols.io>. Further information can be found at <https://www.protocols.io/help/publish-articles>.

Please note that we encourage the authors to self-archive their manuscript (the accepted version before copy editing) in their institutional repository, and in their funders' archives, six months after publication. Nature Portfolio recognizes the efforts of funding bodies to increase access of the research they fund, and strongly encourages authors to participate in such efforts. For information about our editorial policy, including license agreement and author copyright, please visit www.nature.com/ni/about/ed_policies/index.html

An online order form for reprints of your paper is available at https://www.nature.com/reprints/author-reprints.html. Please let your coauthors and your institutions' public affairs office know that they are also welcome to order reprints by this method.

Sincerely,

Nick Bernard, PhD
Senior Editor
Nature Immunology

Click here if you would like to recommend Nature Immunology to your librarian
<http://www.nature.com/subscriptions/recommend.html#forms>

** Visit the Springer Nature Editorial and Publishing website at http://editorial-jobs.springernature.com?utm_source=ejp_NImm_email&utm_medium=ejp_NImm_email&utm_campaign=ejp_NImm for more information about our career opportunities. If you have any questions please click [here](mailto:editorial.publishing.jobs@springernature.com).
